# Nuclear and cytoplasmic huntingtin inclusions exhibit distinct biochemical composition, interactome and ultrastructural properties

Nathan Riguet [1], Anne-Laure Mahul-Mellier[1], Niran Maharjan[1], Johannes Burtscher [1], Marie Croisier[2], Graham Knott [2], Janna Hastings[1,3], Alice Patin[1], Veronika Reiterer[4], Hesso Farhan[4], Sergey Nasarov[1] & Hilal A. Lashuel [1✉]

Despite the strong evidence linking the aggregation of the Huntingtin protein (Htt) to the pathogenesis of Huntington's disease (HD), the mechanisms underlying Htt aggregation and neurodegeneration remain poorly understood. Herein, we investigated the ultrastructural properties and protein composition of Htt cytoplasmic and nuclear inclusions in mammalian cells and primary neurons overexpressing mutant exon1 of the Htt protein. Our findings provide unique insight into the ultrastructural properties of cytoplasmic and nuclear Htt inclusions and their mechanisms of formation. We show that Htt inclusion formation and maturation are complex processes that, although initially driven by polyQ-dependent Htt aggregation, also involve the polyQ and PRD domain-dependent sequestration of lipids and cytoplasmic and cytoskeletal proteins related to HD dysregulated pathways; the recruitment and accumulation of remodeled or dysfunctional membranous organelles, and the impairment of the protein quality control and degradation machinery. We also show that nuclear and cytoplasmic Htt inclusions exhibit distinct biochemical compositions and ultrastructural properties, suggesting different mechanisms of aggregation and toxicity.

[1] Laboratory of Molecular and Chemical Biology of Neurodegeneration, Brain Mind Institute, Ecole Polytechnique Fédérale de Lausanne (EPFL), 1015 Lausanne, Switzerland. [2] BIO EM facility (BIOEM), EPFL, 1015 Lausanne, Switzerland. [3] Bioinformatics Competence Centre (BICC), EPFL, 1015 Lausanne, Switzerland. [4] Institute of Pathophysiology, Medical University of Innsbruck, Innsbruck, Austria. ✉email: hilal.lashuel@epfl.ch

Huntington's disease (HD) is an autosomal dominant genetic and progressive neurodegenerative disorder caused by the abnormal expansion of CAG trinucleotide repeats within exon 1 of the huntingtin gene (HTT)[1–4]. The expanded repeats are translated into a long pathogenic polyglutamine (polyQ) tract (> 36 repeats) that renders the Huntingtin protein (Htt) more prone to aggregate[5]. The length of the CAG repeat is inversely correlated to the age of onset, with the juvenile form associated with a polyQ repeat length of 75 or more[6,7]. The accumulation of Htt-positive intraneuronal aggregates and inclusions in the cortex and the striatum of postmortem brain from HD patients and in animal and cellular models of HD[2] has led to the hypothesis that Htt aggregation plays a central role in the pathogenesis of HD. However, our understanding of the ultrastructural and biochemical composition of Htt inclusions and the molecular mechanisms that drive their formation, clearance, and toxicity remains incomplete. Addressing this knowledge gap is crucial to understand the molecular and cellular mechanisms underpinning HD and to enable the development of effective therapies and disease-modifying strategies for this disease.

Although several studies have investigated specific cellular mechanisms linked to Htt aggregation, the current models often lack detailed characterization of the inclusions at both the biochemical and structural levels. Furthermore, while some of the previous studies explored the effect of polyQ repeat length on the ultrastructure properties of mutant Htt, there are no reports on the role of the first N-terminal 17 amino acids (Nt17) of Htt – which regulates many aspects of the aggregation and cellular properties of Htt proteins[8–11] – on the organization and ultrastructure properties of cytoplasmic and nuclear Htt inclusions. Finally, the vast majority of the previous studies used mutant Htt constructs fused to either peptide-based tags or large fluorescent proteins like GFP that could interfere with the cellular and aggregation properties of Htt[12–15].

Here, we employed cellular and neuronal models of Htt cytoplasmic and nuclear inclusion formation to gain insight into how sequence modifications influence the final ultrastructural and biochemical properties of cytoplasmic and nuclear Htt inclusions and their impact on cellular organelles and functions. The cellular models used in this study are based on the overexpression of N-terminal fragments of mutant Htt comprising the Exon 1 region (Httex1), which contains the polyQ expansion. Incomplete splicing of HTT leading to Httex1 protein expression has been shown to occur in HD patient's brains[16], and Httex1 protein was previously described as a key component of the intracellular inclusions found in HD post-mortem brains[16–18]. Moreover, the expression of pathogenic Httex1 (polyQ tract > 43Q) is sufficient to induce HD-like features, including aggregates formation and toxicity in mice[19–21], Drosophila[22], C. elegans[23] and cell culture models[11,12,15,24–31]. In vitro and cellular studies also showed that Httex1 aggregates in a polyQ repeat length and concentration-dependent manner[32–34]. Therefore the mutant Httex1-based models are useful to study the pathogenesis of HD as they reproduce different aspects of Htt aggregation and have been instrumental in advancing our understanding of the sequence, molecular, and structural determinants of Htt aggregation and inclusion formation[10,26–29,35,36].

We applied a combination of Correlative Light and Electron Microscopy (CLEM) and proteomics-based approaches to investigate the structural and biochemical properties of the cytoplasmic and nuclear Httex1 inclusions in HEK 293 cells and primary cortical neurons. We also investigated the role of the Nt17 domain and the polyQ tract length in modulating the composition and the structural and toxic properties of mutant Httex1 inclusions. Finally, given that a large body of published cellular studies on the mechanisms of Httex1 aggregation and inclusion formation is based on constructs in which Httex1 is fused to the GFP[9,11,12,37–40], we also compared, for the first time, the composition, ultrastructural properties, and toxicity of inclusions formed by native (tag-free) and GFP-tagged mutant Httex1 proteins.

Our results demonstrate that Htt inclusions are composed of a complex mixture of aggregated mutant Httex1, different cellular proteins and membranous organelles, including the endosomal system. We show that functional and ultrastructural properties of Httex1 inclusions are differentially altered by sequence modifications and interactions with lipids and cellular organelles. We demonstrate that Htt cytoplasmic and nuclear inclusions exhibit distinct composition and ultrastructural properties, suggesting different mechanisms of aggregation, inclusion formation, and toxicity. These observations revealed that the differences in the cellular environment and interactome could influence the mechanisms of aggregation, the structural and biochemical properties of the inclusions, and their relative contribution to mutant Htt-induced toxicity. Finally, we show that inclusions produced by mutant Httex1 72Q-GFP exhibit striking differences in terms of organization, ultrastructural properties, protein composition, and their impact on mitochondrial functions compared to the inclusions formed by the tag-free mutant Httex1 72Q. Overall, our findings contribute to a better understanding of the sequence, molecular, and cellular determinants of mutant Httex1 aggregation, inclusion formation, and toxicity. They also underscore the critical importance of further studies to investigate the role that lipids, organelles, and the Htt interactome play in Htt pathology formation and maturation, and Htt-induced toxicity in HD.

## Results

**Httex1 cytoplasmic inclusions exhibit a distinctive core and shell morphology and are composed of highly organized fibrils, cytoplasmic proteins, and membranous structures**. To investigate the ultrastructural properties of Htt inclusions, we first used a mammalian cell model system of HD, in which overexpression of mutant Httex1 with polyQ repeats > 39 has been shown to result in a robust and reproducible formation of cytoplasmic Htt inclusions[15,29]. In this model, nuclear inclusions are also observed in ~15% of transfected cells by Httex1 72Q[41], thus providing an opportunity to investigate and compare the ultrastructural features of cytoplasmic and nuclear Htt inclusions under identical conditions. This model system is widely used to investigate molecular, cellular, and pharmacological modulators of Htt aggregation and inclusion formation[11,12,15,24–31,42,43], though very often using Htt constructs with extra non-natural sequences such as Myc, FLAG, and GFP tags. Previous studies have shown that the presence of such tags significantly alters the aggregation properties of N-terminal fragments of mutant Htt[44–46]. Therefore, in this work, we opted to maintain the native sequence of the protein and investigated mutant Httex1 aggregation and inclusion formation in the absence of additional sequences or tags. Tag-free Httex1 72Q was overexpressed in HEK 293 cells (HEK cells), and the morphology and structural properties of Httex1 inclusions were assessed. Despite the formation and abundance of inclusions formed by Httex1 72Q, we did not observe overt toxicity in HEK cells. The initiation of apoptotic events was apparent only after 96 h, as indicated by Caspase 3 activation without loss of plasma membrane integrity[41].

Next, we assessed the morphology of Httex1 72Q inclusions by immunocytochemistry (ICC) using several antibodies against different epitopes along the sequence of Httex1 (Fig. 1a and S1). Interestingly, all the antibodies presented a strong

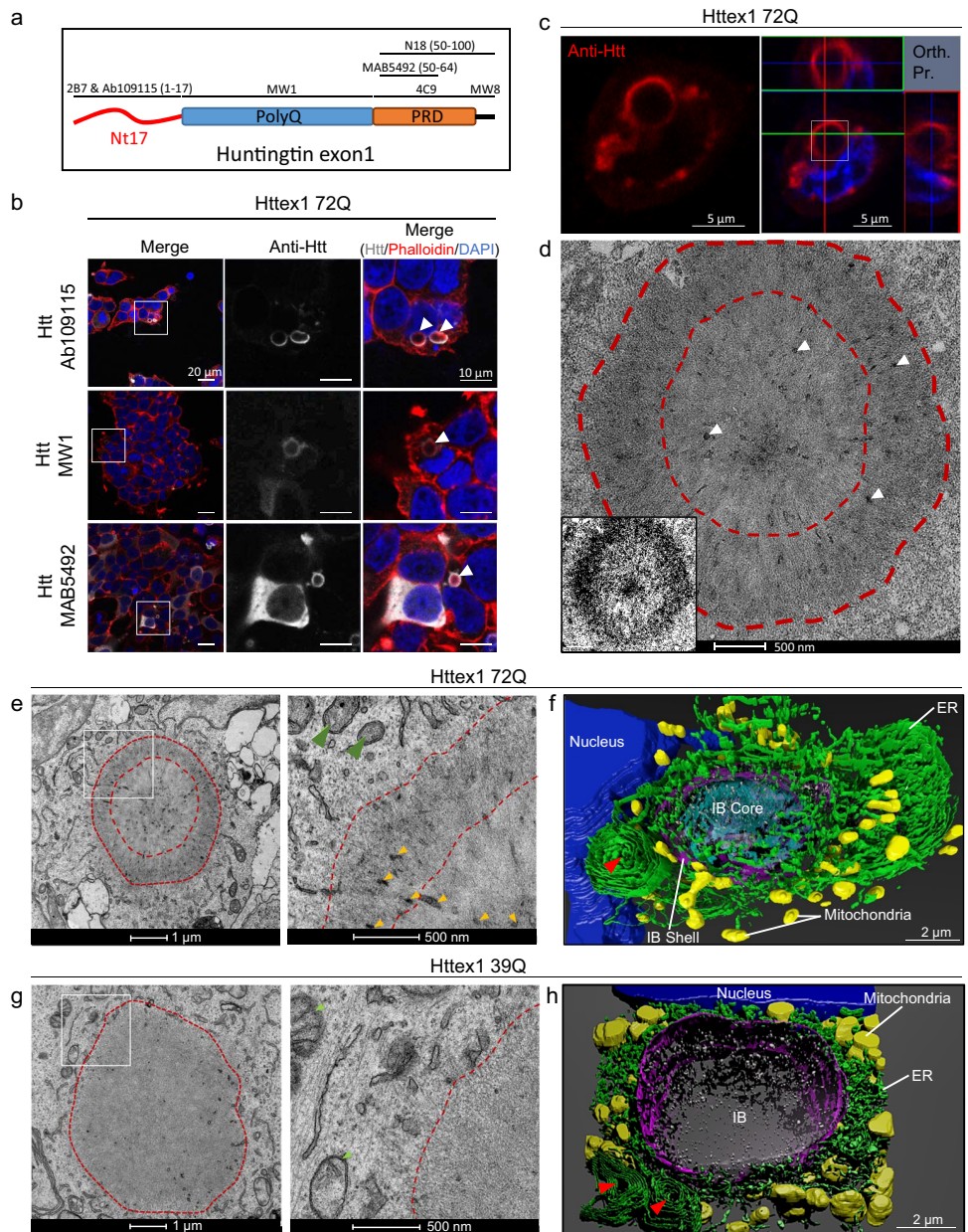

**Fig. 1 Confocal microscopy and CLEM revealed the ring-like structure of the Httex1 72Q inclusions in HEK cells. a** Epitope mapping of the Httex1 antibodies used in this study. **b** Confocal imaging of Httex1 72Q inclusions formed 48 h after transfection in HEK cells. All the Htt antibodies showed strong immunoreactivity to the periphery of the inclusions. The nucleus was counterstained with DAPI and the F-actin with phalloidin. White arrows indicate the colocalization of the F-actin with the ring-like structure of Httex1 inclusions. Scale bar = 20 μm and 10 μm. **c**, **d** CLEM of Httex1 72Q inclusions. **c** Confocal imaging of Httex1 72Q inclusions formed 48 h after transfection in HEK cells. The selected area (white square) was examined by EM (**d**). Orthogonal projection (Orth. Pr.), Scale bars = 5 μm. **d** The presence of membrane fragments and vesicles are indicated by the white arrowheads. The binary image (inset) shows the core and the shell ultrastructure of the Httex1 72Q inclusion. Scale bar = 500 nm. **e** Representative EM images of Httex1 72Q inclusion. Higher magnification (white square) are shown in the right panel. Dashed lines delimit the aggregate and the core of the inclusion. Internalized membranous structures and the mitochondria are indicated by the orange and the green arrowheads, respectively. Scale bar = 1 μm and 500 nm. **f** 3D model of Httex1 72Q cellular inclusion and surrounding organelles (top view). The Httex1 inclusion body (IB) shell is represented in purple, the core in cyan. ER membranes are shown in green, intra-inclusion membranous structures in white, the nucleus in blue, mitochondria in yellow, and the stacked ER cisternae are indicated by a red arrowhead. Scale bar = 2 μm. **g** Representative EM images of Httex1 39Q inclusion formed 48 h after transfection in HEK cells. The white square indicates the area shown in the right panel at higher magnification. Dashed lines delimit the inclusion. Scale bar = 1 μm and 500 nm. **h** 3D model of the Httex1 39Q IB is shown in purple, surrounded by mitochondria (yellow), ER structures (green), stacked ER cisternae (red arrowheads), intra-inclusion membranous structures are shown in white and the nucleus in blue. Scale bar = 2 μm.

immunoreactivity to the periphery of the Httex1 72Q inclusions, and none of these antibodies labeled the core of these inclusions (Fig. 1b and S2a). This observation suggests either an absence of Htt in the center of the inclusions or poor accessibility of the used antibodies to the core of the inclusions, possibly due to the high compactness of the Htt aggregates in the core compared to the periphery of the inclusions. In addition, we observed high colocalization of the filamentous actin (visualized by phalloidin) with Httex1 inclusions (Fig. 1b, white arrowheads), indicating a possible involvement of the cytoskeleton proteins in Htt inclusion formation.

To gain more insight into the structural and organizational features of Httex1 inclusions formed in cells, we turned to electron microscopy (EM). We first employed a correlative approach to analyze the ultrastructure of the inclusions by EM and their subcellular environment seen with CLEM (Fig. 1c, d). Httex1 72Q positive inclusions were immunostained and imaged by confocal microscopy (Fig. 1c) and then subjected to serial sectioning for analysis by EM (Fig. 1d). The EM micrographs of the Httex1 72Q inclusions revealed a surprisingly complex morphology characterized by a halo-like structure with a dense core and a heavily stained outer shell. The outer layer of the inclusions contained fibrillar structures that appeared to be tightly packed and radiating from the core of the inclusion. EM-dense cytoplasmic structures were detected in both the core and the periphery and could reflect the incorporation or sequestration of disrupted organelles inside the inclusion[12,47,48] (Fig. 1d, white arrowheads). The halo-morphology of Httex1 inclusions was consistent across many imaged inclusions (Supplementary Fig. 3a).

To further characterize the structural properties and distribution of Httex1 within the inclusion, we used cryo fixation via high-pressure freezing, which preserves the cellular ultrastructure in its native state. EM imaging revealed radiating fibrils at the periphery and more tightly organized and stacked fibrils in the core (Supplementary Fig. 4a). Electron-dense membranous structures were observed in the core and more to the periphery. These results suggest that both the core and shell contain aggregated fibrillar forms of mutant Htt but with a distinct structural organization. The structural organization of Httex1 72Q inclusions combined with the primary localization of Htt antibodies in their periphery suggests that Htt fibrils in the outer layer of these inclusions serve as active sites for the recruitment of soluble Htt, the growth of Htt fibrils, and the interaction of Htt with other proteins and cellular organelles.

To determine whether the formation of cytoplasmic inclusions involves interactions with or recruitment of membranous structures such as ER and mitochondria, we imaged inclusion positive cells by EM under conditions that preserve the internal membranes of cellular organelles, i.e., in the absence of detergents commonly used in the ICC procedure. EM images revealed that the core and periphery of the Httex1 72Q inclusion contained many small membranous structures (Fig. 1e, yellow arrowheads). An ER network and mitochondria were present at the periphery of these inclusions, suggesting that these regions might act as active sites for the recruitment and interaction of soluble Htt with other proteins and cellular compartments during inclusion formation and maturation. Most of the mitochondria surrounding the cytoplasmic Httex1 72Q inclusions exhibited damaged or markedly reduced numbers of cristae (Fig. 1e, green arrowheads). We next generated a 3D model of the inclusions and the organelles in their vicinity (Fig. 1e, f and Supplementary Movie 1). The 3D model confirmed that inclusions are formed in a crowded region with ER and mitochondria all around it. The 3D model also suggested that the electron-dense membranous structures recruited inside the

Httex1 inclusions were mostly composed of endomembranes and vesicles (labeled in white), consistent with a previous report[12]. Interestingly, the ER adopted a specific "rosette-like" or "stacked cisternae" morphology (highlighted by a red arrowhead) in the periphery of some Httex1 72Q inclusions. Finally, it is noteworthy that the mitochondria (labeled in yellow, Fig. 1f) were not detected inside the inclusion but rather at the periphery of the inclusions. Interestingly, despite their proximity to the nucleus, the inclusion did not compromise the nuclear membrane integrity (Supplementary Fig. 3a).

Next, we sought to identify the endomembrane compartments within Httex1 72Q inclusions using a panel of antibodies or dyes labelling intracellular compartments. The mitochondrial (Tom 20 and Mitotracker) and ER (BiP/Grp78) markers were strongly detected near Httex1 72Q cytoplasmic inclusions (Supplementary Fig. 5a–c). The autophagy flux marker, p62, was enriched in the periphery of Httex1 72Q inclusions (Supplementary Fig. 5d). Markers of aggresome formation, such as Vimentin and HDAC6, were also enriched in the periphery and in close proximity to Httex1 72Q inclusions (Supplementary Fig. 5e, f). Moreover, when cytoskeletal proteins such as actin and tubulin were overexpressed (fused with RFP), they were observed mainly at the periphery of the Httex1 72Q inclusions (Supplementary Fig. 5g, h). This observation is consistent with the colocalization of F-actin (stained by phalloidin) with Httex1-72Q inclusions (Fig. 1b, white arrowheads). None of the organelles' markers were detected inside the core of these inclusions (Supplementary Fig. 5), although the EM images clearly showed the presence of membranous structures in the center (Fig. 1e). This confirms the poor accessibility of antibodies and dyes to stain the core of the Httex1 inclusions likely due to their compactness and highlights the importance of using EM to decipher the ultrastructural properties and composition of the pathological inclusions.

Altogether our data suggest that Httex1 aggregates could form at the surface of membranes as suggested by Suoponki and colleagues[49], which could promote early aggregation events as well as perturbation of membranous structures and their recruitment into cytoplasmic Httex1 inclusions.

**Cytoplasmic and nuclear Httex1 inclusions in HEK cells exhibit distinct ultrastructural properties.** Both cytoplasmic and nuclear inclusions have been observed in HD patients' brains and transgenic mouse models of HD[50–52]. Therefore, we compared the structural and organizational properties of Httex1 72Q inclusions in the cytoplasm (~85%) and nucleus (~15%) of the transfected HEK cells containing inclusions[41]. Using immunofluorescence-based confocal microscopy, we did not observe significant differences in the size or overall morphology between the Httex1 72Q inclusions in the nucleus and the perinuclear region (Fig. 2a). However, EM clearly showed that the nuclear inclusions formed by the Httex1 72Q were enriched in fibrillar structures (Fig. 2b) but did not exhibit the classical core and shell organization nor did they contain the membranous structures trapped within the cytoplasmic inclusion (Fig. 1d, e). This suggests that the intracellular environment is a key determinant of the structural and molecular complexity of the inclusions and that nuclear and cytoplasmic inclusion formation occurs via different mechanisms.

**The length of the polyQ domain is another key determinant of inclusion formation.** We and others have previously shown that the polyQ repeat length strongly influences the conformation and aggregation properties of Httex1, with higher polyQ favoring the formation of a more compact polyQ domain and accelerating Htt

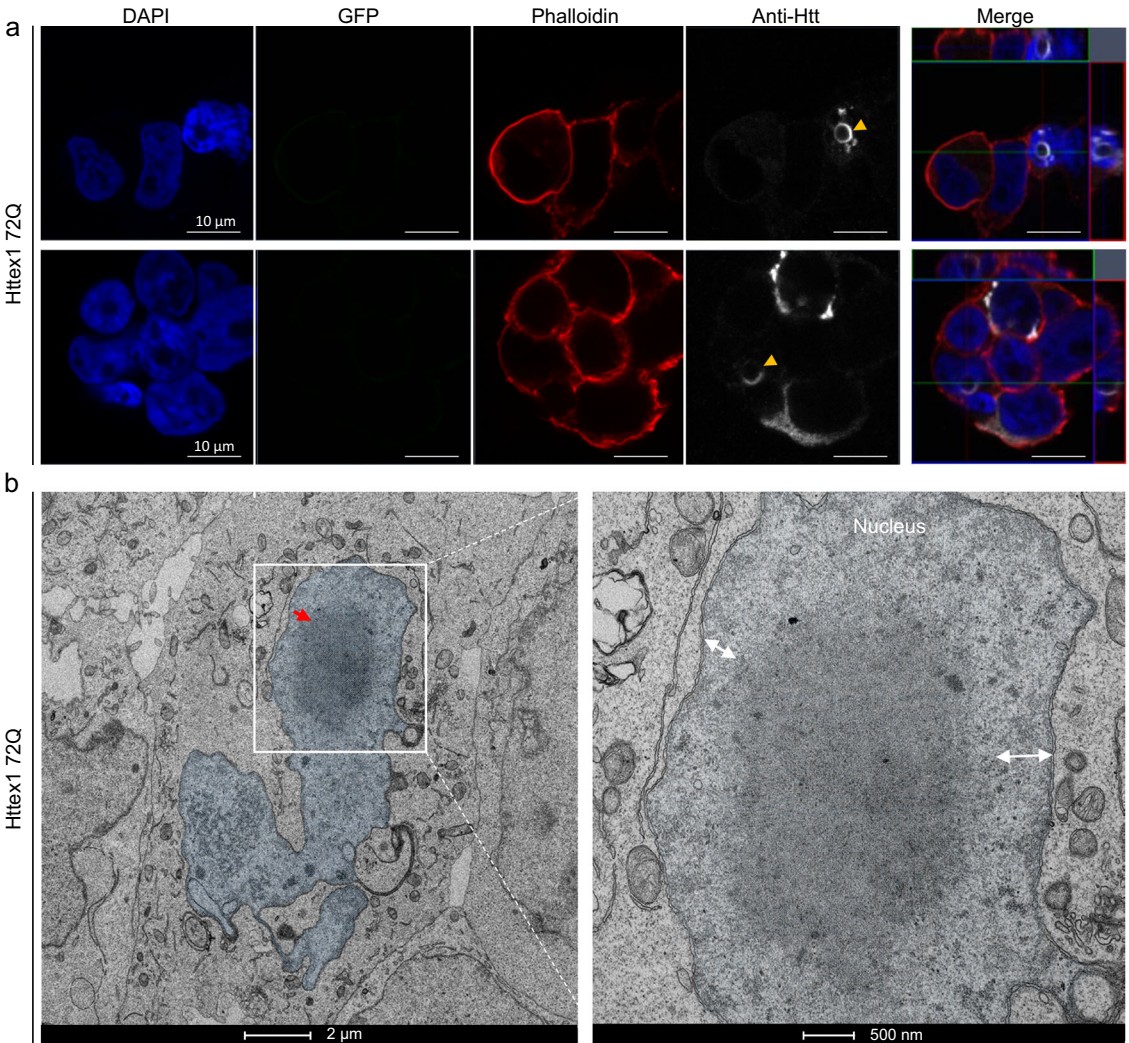

**Fig. 2 The nuclear inclusions formed by Httex1 72Q do not exhibit the classical core and shell organization observed for the cytosolic inclusions. a** Representative confocal images of Httex1 72Q nuclear inclusions, 48 h after transfection. Httex1 expression (grey) was detected using a specific primary antibody against the N-terminal part of Htt (amino acids 1–17; 2B7 or Ab109115). The nucleus was counterstained with DAPI (blue), and phalloidin (red) was used to stain the actin F. Httex1 nuclear inclusions are indicated by the yellow arrowheads. Scale bars = 10 µm. **b** Electron micrograph of a representative nuclear Httex1 72Q inclusion. The white square indicates the area shown at higher magnification in the right-hand panel. The nucleus is highlighted in blue, and the double arrows indicate the distance between the nuclear inclusion and the nuclear membrane. No interaction between the nuclear inclusion and the nuclear membrane was observed. Scale bars = 2 µm (left-hand panel) and 500 nm (right-hand panel).

aggregation in vitro and in vivo[34,45,53]. However, the polyQ dependence on Htt inclusion formation has predominantly been assessed mainly by ICC and in the context of Htt fused to fluorescent proteins and polyQ repeats much longer than the pathogenic threshold (64Q-97Q[12]; 64Q-150Q[30]; 43Q-97Q[14]). Therefore, we next investigated whether the length of the polyQ repeat also influences the organization and ultrastructural properties of cytoplasmic Htt inclusions in HEK cells. Toward this goal, we investigated the level of aggregation and the structural organization of the Httex1 inclusions carrying different polyQ lengths (16Q, 39Q vs. 72Q). Consistent with previous data from our group, no significant cell death was observed in HEK cells overexpressing Httex1 16Q or 39Q constructs even after 96 h[41]. However, cells expressing Httex1 72Q underwent apoptosis after 96 h. As expected, no inclusions were formed upon overexpression of Httex1 16Q even after 72 h post-transfection (Supplementary Fig. 6). Httex1 39Q inclusions were detected predominantly in the cytoplasm of the HEK cells at all the time points examined (24–72 h), though at lower numbers than in the

Httex1 72Q conditions: Httex1 39Q (16%) vs. Httex1 72Q (38%) of transfected cells[41].

The dark shell structure that delimits the core from the periphery of the Httex1 72Q inclusions (Fig. 1d, e) was absent in the Httex1 39Q inclusions (Fig. 1g). In addition, the Httex1 39Q inclusions appeared less dense compared to those of the Httex1 72Q. These observations were consistent for all eight inclusions imaged per condition (Supplementary Fig. 3b).

Similar to what we observed for Httex1 72Q, the 3D reconstruction of the Httex1 39Q inclusions clearly showed alteration of the ER organization, as well as the localization of mitochondria near the inclusions (Fig. 1h and Supplementary Movie 2). The electron-dense membranous structures found inside the inclusions were identified as endomembranes and vesicles. Httex1 39Q expressing cells also contained specific ER-cisternae at the periphery of the inclusions (Fig. 1h, red arrowheads). Altogether, our data establish that polyQ expansion plays a critical role in determining the final architecture and ultrastructural properties of the Httex1 inclusions.

**Removal of the Nt17 domain reduces mutant Httex1 aggregation but does not influence the organization and ultrastructural properties of the inclusions**. The Nt17 domain functions as a Nuclear Export Signal (NES) and has been shown to play an important role in regulating the intracellular localization of Htt as well as its aggregation kinetics and extent of inclusion formation[36,54]. Therefore, we sought to assess the role of Nt17 in regulating the ultrastructural properties of Htt inclusions. Towards this goal, we generated Httex1 39Q and 72Q mutants lacking the entire Nt17 domain (ΔNt17) and compared the structural properties of the inclusions formed by these mutants to those formed by Httex1 39Q and Httex1 72Q. Quantitative confocal microscopy revealed a strong reduction in the number of inclusions (~50% reduction) of cells transfected by Httex1 ΔNt17 72Q compared to Httex1 72Q[41]. Surprisingly, inclusions formed by the Httex1 ΔNt17 72Q (Supplementary Fig. 7a, b) exhibited an architecture and organization (central core and peripheral shell) similar to those formed by Httex1 72Q (Fig. 1d, e and Supplementary Fig. 8a). Furthermore, similar to Httex1 39Q, the Httex1 ΔNt17 39Q cytoplasmic inclusions did not exhibit a core and shell architecture. These observations suggest that the Nt17 domain – while playing a crucial role in regulating the kinetics and early events of Htt aggregation – does not influence the morphology or structural organization of Htt cytoplasmic and nuclear inclusions. This is surprising given that in vitro and cellular studies have consistently shown that the Nt17 domain plays an important role in regulating the kinetics and structural properties of Htt aggregation[11,41] and Htt interactions with lipids and membranes[8,9,55–57]. In a recent study using solid-state nuclear magnetic resonance spectroscopy (ssNMR), Boatz et al. showed that the Nt17 domain and part of the PRD (PPII helices) are buried in the core of the fibrils, while the other part of the PRD (random coil) remains dynamic, accessible and regulates multifilament assemblies in vitro[58]. Together, these observations could explain why the deletion of the Nt17 domain does not interfere with the organization of the inclusion and suggest that the interactions between the fibrils and other proteins and/or organelles are most likely mediated by the flexible PRD domain.

**Neutral lipids are incorporated into Httex1 cellular inclusions in a polyQ length-dependent manner**. Although several studies have shown dysfunction of cholesterol metabolism in various cellular and animal models of HD[59,60], the role of lipids in Htt inclusion formation and the lipid composition of cellular huntingtin inclusions remains unknown[61,62]. To gain further insight into the role of lipids in the formation and structural organization of Httex1 inclusions, we next assessed their presence using fluorescent probes targeting different lipid classes. We did not observe the recruitment of ceramide, cholesteryl ester, or phospholipids into mutant Httex1 inclusions (Supplementary Fig. 9a-c). Interestingly, neutral lipids were not found in the center of Httex1 39Q inclusions (Supplementary Fig. 9d white arrowheads) but were enriched in Httex1 72Q inclusions (Supplementary Fig. 9e). This could contribute to the polyQ length-dependent differences in the ultrastructural properties of the Httex1 inclusions (Fig. 1e, g). Neutral lipids were also detected in nuclear inclusions (Supplementary Fig. 9e, yellow arrowheads). Although the Nt17 domain has been shown to act as a lipid- and membrane-binding domain, neutral lipids were also detected in the Httex1 ΔNt17 72Q inclusions, though not inside Httex1 ΔNt17 39Q inclusions (Supplementary Fig. 10a). These results demonstrate that polyQ-dependent interactions between Htt and neutral lipids play an important role in Httex1 aggregation and the formation of both nuclear and cytoplasmic inclusions. Interestingly, these interactions are not dependent on the Nt17 domain. Whether mutant Httex1 interactions with lipids occur prior to aggregation and help initiate its oligomerization or represent late events associated with the maturation of inclusions remains unknown.

**Quantitative proteomics reveals that the formation of Httex1 72Q inclusions involves the active recruitment and sequestration of proteins and organelles**. To gain further insight into the biochemical composition of Htt inclusions, we performed quantitative proteomic analysis and compared the differentially abundant proteins between the Urea soluble fractions of HEK cells overexpressing Httex1 72Q or Httex1 16Q (see experimental details, Supplementary Fig. 11a, c). Western Blot analysis of the Urea fraction confirmed the presence of Httex1 72Q within inclusions, while non-pathogenic Httex1 16Q was only detected in the soluble fraction (Supplementary Fig. 11a).

As expected, no proteins were significantly enriched in the insoluble fraction of HEK cells expressing Httex1 16Q compared to those expressing GFP (Supplementary Fig. 11d). In contrast, 377 proteins were significantly enriched in the insoluble fraction of HEK cells overexpressing Httex1 72Q compared to cells overexpressing Httex1 16Q (Fig. 3a). Among these proteins, we identified the endogenous HTT protein (Supplementary Fig. 11e). This suggests that the aggregation process triggered by the overexpression of Httex1 also leads to the recruitment of endogenous HTT protein.

Classification by cellular component (CC, Fig. 3Bb using the Gene Ontology (GO) showed that 55% of the proteins enriched in the insoluble fractions of HEK cells containing Httex1 72Q inclusions were part of the cytoplasmic compartment, with 24% of these proteins belonging to the endomembrane system, including the endolysosomal apparatus (clathrin-coated endocytotic vesicles, early endosome, endosomes, recycling endosomes, exosomes, and autophagosomes), the vesicles involved in Golgi-ER transport, and membranes from the Golgi and trans-Golgi network. These data are in line with the EM data showing that the Httex1 72Q inclusions were composed of small membranous structures and vesicles. Approximately 14% of the proteins enriched in the insoluble fraction were classified as pertaining to the cytoskeleton compartment, with the actin cytoskeleton being the most predominant, consistent with our confocal results (Fig. 1b). The absence of mitochondrial proteins in the soluble Urea fraction confirms that mitochondria are not sequestered inside the inclusions but, rather, accumulate at the periphery, as shown by our EM imaging. The rest of the cytoplasmic proteins sequestered in the insoluble fraction of the Httex1 72Q-transfected HEK cells were mainly from the perinuclear region or from macromolecular protein complexes such as the ubiquitin-proteasome system (UPS), the mRNA processing bodies, and the stress granules. Interestingly, the insoluble fraction of the Httex1 72Q-transfected HEK cells was also significantly enriched by proteins of the nuclear compartment (~45%). Among the nuclear proteins, ~71.6% belonged to the nucleoplasm. Proteins from the nucleolus, the nuclear bodies, the nuclear envelope, and the nuclear pore were also significantly enriched.

Classification of the enriched proteins by biological function showed that the most highly enriched terms for the GO biological process (Fig. 3c) and the signaling pathways using the Ingenuity Pathway Analysis (Supplementary Fig. 12) were related to the proteins involved in the proteasomal ubiquitin-dependent protein degradation. Proteins from the UPS system[63], including ubiquitin moiety, E3 ubiquitin ligases (e.g., ITCH, RNF34, TRIM32), 26S proteasome subunits, and deubiquitinases formed the major cluster of the differentially enriched proteins in the Httex1 72Q inclusions. These results are in line with previous studies showing

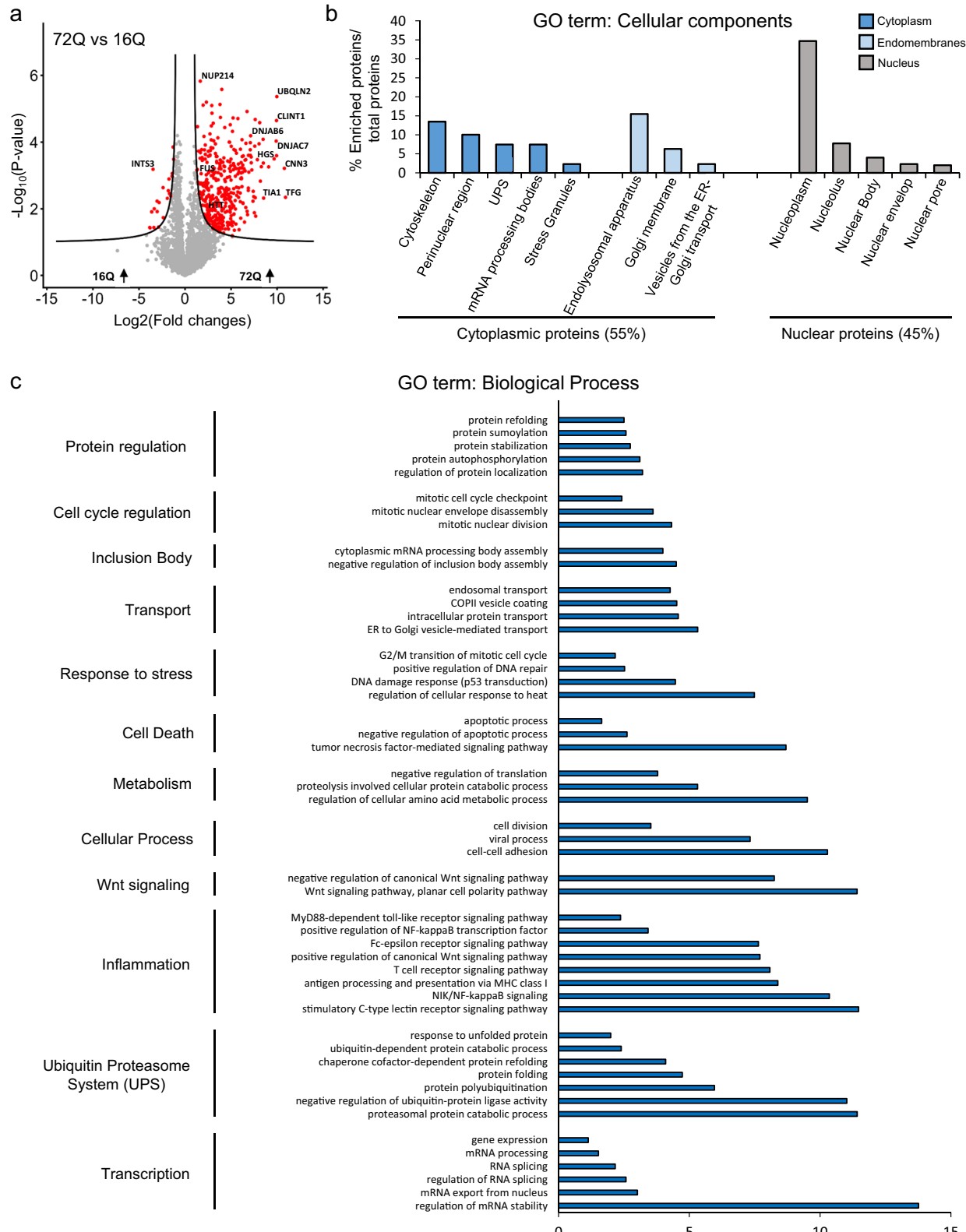

**Fig. 3 Proteomic analysis of the enriched protein contents found in the insoluble fraction of HEK expressing Httex1 72Q reveals protein aggregation and stress response pathways.** Urea soluble proteins from HEK expressing Httex1 72Q or Httex1 16Q of 3 independent experiments were extracted 48 h after transfection and analyzed using LC-MS/MS. **a** Identified proteins were plotted using a volcano plot. Black lines represent the threshold of significance at a false discovery rate (FDR) < 0.05 and an S0 of 0.5 which were used for the subsequent analysis. **b, c** Proteins significantly enriched in Httex1 72Q (red dots in the right part of the volcano plot) were classified by cellular component (**b**) or biological processes (**c**) using Gene Ontology (GO) term and DAVID enrichment (−log$_{10}$(p-value) > 1) analyses. Source data are provided as a Source Data file.

that Htt inclusions formed in HD patients or several cellular and in vivo models sequester several key components of the UPS system, including 26S proteasomes[64] (ADRM1[65]), deubiquitinases[66] (e.g., NEDD4 and USP5), and E3 ubiquitin ligases (e.g., ITCH, TRAF6, UBE3A, UHRF2, and Parkin) and induce impairment of the UPS[67–69]. In addition, our analysis revealed that among the proteins significantly up- or down-regulated in Httex1 72Q compared to Httex1 16Q, 42 proteins are known Htt interactors linked to the UPS, identified from the HDinHD database (Supplementary Data 1).

Proteins involved in autophagosome formation (optineurin[70]), maturation (ubiquilins[71]), and the process of autophagosome-lysosome fusion[72] (TOLLIP-interacting proteins and proteins involved in endolysosomal trafficking) were also enriched inside the Httex1 72Q inclusions. Both TOLLIP[73–75] and optineurin[76–79] are critical for the efficient clearance of polyQ protein aggregates[80] and, in particular, for the degradation of Htt aggregates. The depletion of TOLLIP in HeLa cells increases GFP-Htt-103Q-induced toxicity[73], while optineurin knockdown promotes Htt aggregation.

In addition, several chaperones from the Hsp70 and the DnaJ/HSp40 families (e.g., DNAJB6, DNAJB2, and other proteins from the Hsp40 and Hsp70 families), as well as ubiquilin-2, found previously enriched in Htt inclusions[30,81,82], were also enriched inside the Httex1 72Q inclusions. This cluster of proteins was associated with the BAG2 signaling pathway (Supplementary Fig. 12), one of the top canonical signaling pathways that regulate the interplay between the chaperones from the Hsp70/Hsc70 family and ubiquitin. Interestingly, it has recently been shown that the Hsp70 complex, together with the Hsp40/110 chaperone family, formed a disaggregase complex that can directly bind to Htt aggregates[83]. After disaggregation, Ubiquilin-2 interacts with Htt and shuttles the disaggregated species to the proteasome to promote its complete degradation[84]. The enrichment of the disaggregase chaperones network in the Httex1 72Q inclusions suggests that they were actively recruited but failed in their attempt to clear the Httex1 72Q aggregates. Our proteomic analysis also revealed the enrichment of several biological processes and signaling pathways related to RNA binding proteins, transcription factors, RNA splicing, mRNA processing, and stability, as well as chromatin and nucleotide-binding proteins (Fig. 3c and Supplementary Fig. 12). Dysregulation of transcriptional gene pathways has been reported in several animals and cell HD models[85–87] as well as in HD post-mortem tissues and HD peripheral blood cells[88–90].

Together, our proteomic and CLEM data provide strong evidence that the formation of the Httex1 72Q inclusion formation involves the active recruitment and sequestration of cellular proteins, lipids, and organelles. This also suggests that sequestration of transcription regulators and depletion of key proteins from the autophagolysosomal, UPS, and chaperone pathways, due to their sequestration inside the Httex1 pathological inclusions, are major contributors to cellular dysfunction and neurodegeneration in HD, as reported in HD human brain tissue[91].

**Httex1 72Q cytoplasmic inclusion formation induces mitochondrial fragmentation, increases mitochondrial respiration and leads to ER-exit site remodeling**. The remarkable accumulation of damaged mitochondria at the periphery of Httex1 72Q inclusions prompted us to investigate how Htt inclusion formation impacts mitochondrial functions. Quantification of mitochondrial length from EM-micrographs revealed a shorter mitochondrial profile associated with Httex1 72Q inclusions, as compared to HEK cells transfected with empty vector (Fig. 4a, b).

Similar levels of the outer mitochondrial membrane protein VDAC1 suggest that this mitochondrial fragmentation was not associated with a decrease in mitochondrial density (Supplementary Fig. 13a). We hypothesized that the fragmentation and recruitment of mitochondria to Httex1 inclusions might be associated with respirational dysfunction. Therefore, we performed high-resolution respirometry on cells transfected with Httex1 16Q and 72Q for 48 h. Aggregates were only detected in mutant Httex1 72Q transfected cells (Supplementary Fig. 13b). We assessed different respirational states of mitochondria by high-resolution respirometry (Supplementary Fig. 13c-d). Httex1 72Q transfection resulted in significantly higher mitochondrial respiration than Httex1 16Q (Supplementary Fig. 4c). Together, our results demonstrate a clear impact of Httex1 72Q inclusion formation on mitochondrial morphology and function.

Our EM analysis and 3D reconstruction further revealed the presence of ER cisternae at the periphery of the Httex1 inclusions. Stacked ER cisternae are usually formed by an increasing concentration of specific resident proteins or stress conditions[92]. The changes we observed in ER organization, together with the enrichment of proteins related to ER-Golgi trafficking inside the inclusions, prompted us to investigate whether ER functions were impaired upon the formation of the Httex1 inclusions.

A major function of the ER is the biogenesis of COPII carriers that ferry proteins and lipids to distal compartments. COPII carriers form at ER exit sites (ERES), which are ribosome-free domains of the rough ER. To determine whether the formation of Httex1 inclusions interferes with the homeostasis of the ERES, we used confocal imaging and quantified the number of ERES, labelled specifically by the COPII component Sec13 protein in HeLa cells expressing Httex1 72Q, Httex1 39Q, or Httex1 16Q (Fig. 4d). Our data show that the number of ERES was significantly reduced (~20%) only in cells containing Httex1 72Q inclusions (Fig. 4e). We next quantified the size of the ERES using the same imaging pipeline (Fig. 4f). Overexpression of Httex1 16Q caused a 20% reduction in the size of the ERES compared to the empty vector (EV). However, the reduction became much more significant in the cells carrying Httex1 39Q or Httex1 72Q, with a ~40% decrease compared to the empty vector and ~20% compared to Httex1 16Q. These results demonstrate that the formation of the Httex1 inclusions interferes with the formation and fusion of ERES. The remodeling of ERES in cells has been described primarily as an adaptive response to the protein synthesis level of ER with the number of ERES proportional to the cargo load. Interestingly, our proteomics results (Fig. 3a) showed that the TGF (Transforming Growth Factor) protein—which plays a central role in the biogenesis and organization of ERES—is sequestered in mutant Httex1 inclusions[93]. In line with our results, it has been previously shown that the depletion of TGF induced a dramatic reduction of the ERES[94]. Alternatively, the reduction of the ERES sites could represent early signs of cell vulnerability and toxicity induced by the presence of the Httex1 inclusions in cells.

**Expression of Httex1 72Q in primary cortical neurons leads to the formation of dense and filamentous nuclear inclusions**. We next investigated the sequence determinants of Htt aggregation and inclusion formation in primary cortical neurons over time. As expected, none of the Httex1 16Q constructs (Httex1 16Q and ΔNt17 Httex1 16Q) overexpressed in neurons induced the formation of aggregates up to 14 days post-transduction (D14). Conversely, overexpression of Httex1 72Q induced the formation of round nuclear inclusions in almost 100% of transduced neurons, already at D3 (Fig. 5a, b, and S14). Less than 1% of the neurons showed cytoplasmic inclusions, either as puncta or with

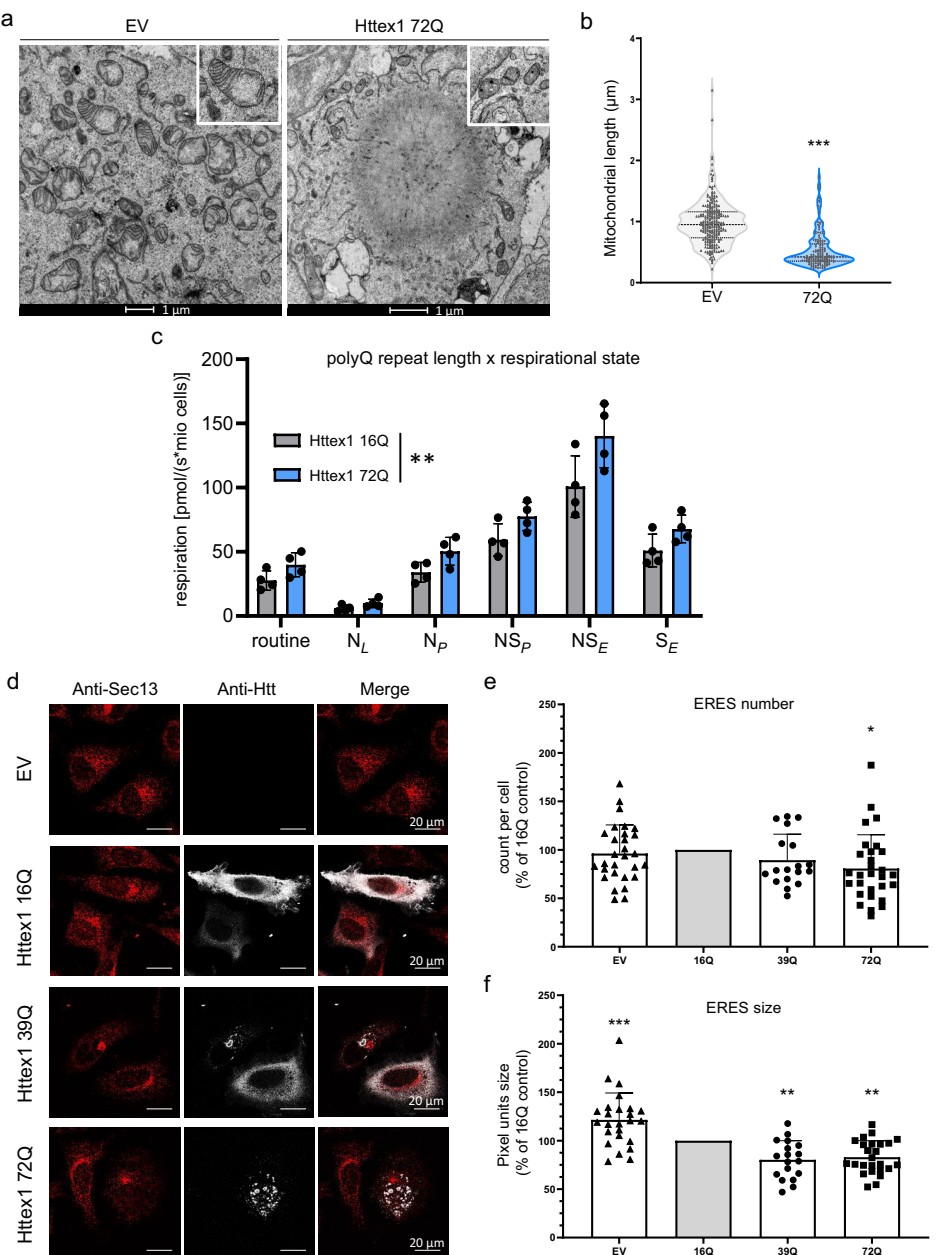

**Fig. 4 Formation of Httex1 72Q inclusions induces mitochondrial alterations and the reduction of ER-exit sites. a** Electron micrographs of mitochondria in HEK cells overexpressing empty vector (EV) or Httex1 72Q. The insets depict higher magnification of the mitochondria found at the periphery of the Httex1 72Q inclusions or in EV controls. Scale bars = 1 μm. **b** Measurement of the mitochondrial length with FIJI reveals a significant reduction in the size of the mitochondrial profile located in the proximity of the inclusions. An unpaired $t$-test was performed (two-sided) resulting in a $p$-value of 0.0006. An unpaired t-test was performed (two-sided) resulting in a $p$-value of 0.0006. **c** HEK cells from 4 independent experiments were gently detached for high-resolution respirometry (HRR) 48 h after transfection with indicated constructs. HRR was performed in respiration media. After the measurements of routine respiration, cells were permeabilized by digitonin, and different respirational states were subsequently induced using a substrate-uncoupler-inhibitor titration (SUIT) protocol. Routine respiration, NADH-driven, or complex 1-linked respiration after the addition of ADP (OXPHOS state) (NP), NADH- and succinate driven, or complex 1 and 2-linked respiration in the OXPHOS state (NSP), and in the uncoupled electron transport system (ETS) capacity (NSE), as well as succinate driven, or complex 2-linked respiration in the ETS state (SE) were assessed. The graphs represent the mean ± SD of 4 independent experiments. Two-way ANOVA showing a significant interaction between the respirational states and the polyQ repeat length ($p$-value = 0.004). **d** Representative confocal images of HeLa cells transfected with Httex1 16Q, 39Q, or 72Q or EV. Cells were fixed 48 h after transfection and immunostained. Httex1 was detected with the MAB5492 Htt antibody (grey), and ER exit sites (ERES) were detected with Sec13 (red). Scale bars = 20 μm. **e**, **f** ERES number (**e**) and size (**f**) quantifications from confocal imaging were performed using FIJI. The graphs represent the mean ± SD of 3 independent experiments represented as a relative percentage to Httex1 16Q control. One-way ANOVA followed by a Tukey honest significant difference [HSD] post hoc test was performed ($p$-values < 0.0001 for panels e and f). *$P$ < 0.05, **$P$ < 0.005, ***$P$ < 0.001 for multiple comparisons.

the ring-like morphology. Although no changes in the subcellular distribution of the inclusions were observed up to D14, we observed significant changes in the size and shape of the nuclear inclusions over time (Fig. 5c). At D3, the majority of the nuclear inclusions were detected as small (<1 μm) nuclear puncta (94%), and only a few appeared as large (~3–4 μm) inclusions (Fig. 5d). The ratio of small and large nuclear inclusions shifted slightly over time from 50:50 at D7 to 43:57 at D14 (Fig. 5d).

We next characterized the ultrastructural properties of the nuclear Httex1 72Q inclusions by CLEM. At D7, these inclusions appeared as dense and roughly round aggregates without distinctive core and shell structural organization (Fig. 6a and S15a). In these thin sections, the intranuclear Httex1 72Q inclusions appeared darker than the surrounding nucleoplasm and structurally different from the nucleolus (Fig. 6a and controls Supplementary Fig. 15c, d). The high density of these aggregates made it challenging to determine if they were made of filamentous structures. However, using electron tomography (ET), we were able to visualize and confirm the presence of filamentous structures inside these nuclear inclusions (Fig. 6b and Supplementary Movie 3). The segmented filaments did not appear to be closely stacked in parallel but rather organized as a network of tortuous filaments. Neither EM nor ET imaging revealed the presence of membranous-, organelle- or vesicle-like structures inside or at the periphery of these inclusions. Although removal of the Nt17 domain (Fig. 6c) did not alter the ultrastructural properties or composition of the inclusions, it accelerated the formation of large aggregates significantly. As early as D3, ~60% of the neurons overexpressing ΔNt17 Httex1 72Q already contained large nuclear inclusions, compared to only ~6% for Httex1 72Q (Fig. 5d). At D14, almost all the aggregates formed in the ΔNt17 Httex1 72Q overexpressing neurons converted into the large nuclear inclusions (~90%) (Fig. 5d), compared to 57% for Httex1 72Q.

Finally, among the small aggregates dispersed throughout the nucleus, a few were observed near the nuclear membrane, which often appeared damaged and ruptured (Figures S15a). Our data are in line with previous EM studies from human HD patients[95] and HD mice models[19,96] showing nuclear ultrastructural changes, including altered nuclear membrane shape, nuclear invagination, and increased nuclear pore density in neurons bearing Httex1 inclusions. Consistent with these observations, the nuclei containing Httex1 72Q inclusions showed enhanced nuclear condensation over time (Fig. 5e), indicating increased neuronal toxicity, consistent with previous reports in HD patients[97] and HD mice model[96]. Interestingly, despite the presence of the large inclusions as early as D3 in the neurons overexpressing ΔNt17 Httex1 72Q, we did not observe an earlier onset of cell death or a higher level of toxicity over time in these neurons compared to those overexpressing Httex1 72Q (Fig. 5e). This suggests a lack of correlation between cell death level and the size of the inclusions, with the large inclusions being less toxic.

These results demonstrate the formation of condensed mutant Httex1 fibrillar aggregates within intranuclear inclusions in neurons and suggest that the process of their formation and maturation is directly linked to neuronal dysfunctions and cell death.

**The addition of GFP to the C-terminal part of Httex1 induces a differential structural organization and toxic properties of Httex1 inclusions**. Given that the great majority of cellular models of HD rely on the use of fluorescently tagged Htt constructs, we next investigated the aggregation properties of mutant Httex1 fused to GFP (Httex1 72Q-GFP and Httex1 39Q-GFP). First, we assessed the morphology of Httex1 72Q-GFP

cytoplasmic inclusions in HEK cells by ICC using a panel of Httex1 antibodies (Fig. 7a, b and S16a). Confocal imaging revealed a diffuse GFP signal throughout the inclusions. Conversely, the Htt antibodies labeled mainly the outermost region of Httex1 72Q-GFP inclusions, but also faintly stained their centers (Fig. 7b). In contrast, the tag-free inclusions' cores were not labeled by all the Htt antibodies tested. Thus, we hypothesized that the presence of the GFP tag results in the formation of less compact Httex1 inclusions. While Actin-F was found to colocalize with tag-free Httex1 inclusions [(Fig. 1b, Httex1 72Q), 39Q (Supplementary Fig. 9d, Httex1 39Q), and ΔNt17 Httex1 72Q (Supplementary Fig. 10a, ΔNt17 Httex1 72Q)], no specific enrichment of Actin-F was detected in the core or periphery of the Httex1 72Q-GFP inclusions. Furthermore, actin filaments were found exclusively at the periphery of the tag-free Httex1 72Q and 39Q (+/− ΔNt17) cytoplasmic inclusions (Fig. 1b and S10A). These results suggest a potential role of actin in the formation of Httex1 inclusion formation or maturation. Consistent with these observations, the actin cytoskeleton was one of the most dysregulated pathways according to a proteomic analysis conducted in human HD brains[98], underscoring its importance in the development of HD pathology.

Next, we performed a more in-depth analysis of the Httex1 72Q-GFP cellular inclusions formed in HEK by CLEM (Fig. 7c, d). The Httex1 72Q-GFP inclusions were organized as a highly dense network of fibrils, which were more homogenously stained (Fig. 7d and S16c) and did not exhibit the core and shell architecture that is characteristic of the tag-free Httex1 72Q inclusion (Fig. 1d and S16d). Closer examination of the inclusions showed that they were composed of densely packed fibrils that exhibited a striking resemblance to the fibrillar aggregates formed by mutant Httex1 proteins in a cell-free system (Fig. 7d and S16b). Structural analysis of Httex1 72Q-GFP by high-pressure freezing fixation (Supplementary Fig. 4b) also revealed densely packed fibrils radiating from the inclusions. The center of the inclusions was not well resolved, but thicker fibrils could clearly be observed radiating at the periphery with increased spacing between them compared to tag-free Httex1 72 inclusion (Supplementary Fig. 4a). A portion of Httex1 72Q-GFP inclusions exhibited perinuclear localization. In some cases, the accumulation of fibrils near the nuclear membrane leads to apparent distortion of the nucleus but without membrane disruption (Fig. 7d and S17a). Overall, we observed no significant differences in diameter and distance from the nucleus for all the Httex1 (+/− GFP) inclusions imaged in HEK cells (Supplementary Fig. 18).

Next, we investigated the ultrastructure properties of Httex1 72Q-GFP inclusions present in the nucleus of the HEK cells (Fig. 7e, f). No significant differences in terms of the organization were observed between nuclear and cytoplasmic Httex1 72Q-GFP inclusions by both GFP detection and antibody staining (Fig. 7e). The Httex1 72Q-GFP nuclear inclusions were also enriched in fibrillar structures. The 3D reconstruction generated from the series of electron micrographs revealed much fewer membranous structures in both Httex1 39Q-GFP and Httex1 72Q-GFP inclusions (Supplementary Fig. 19) compared to tag-free Httex1 inclusions (Fig. 1f, h). Moreover, consistent with EM observations, no neutral lipids were found in Httex1 39Q-GFP and Httex1 72Q-GFP inclusions (Supplementary Fig. 10b-c). Interestingly, neither the length of the polyQ repeat nor the presence or removal of the Nt17 domain seem to significantly alter the size, morphology, or structural properties of the inclusions formed by mutant Httex1 proteins fused to GFP (Supplementary Fig. 20). This is different from what we observed for cells expressing tag-free Httex1, in which the increase of the polyQ length led to inclusions with distinct morphologies and organizational features. These findings suggest that the addition of GFP significantly

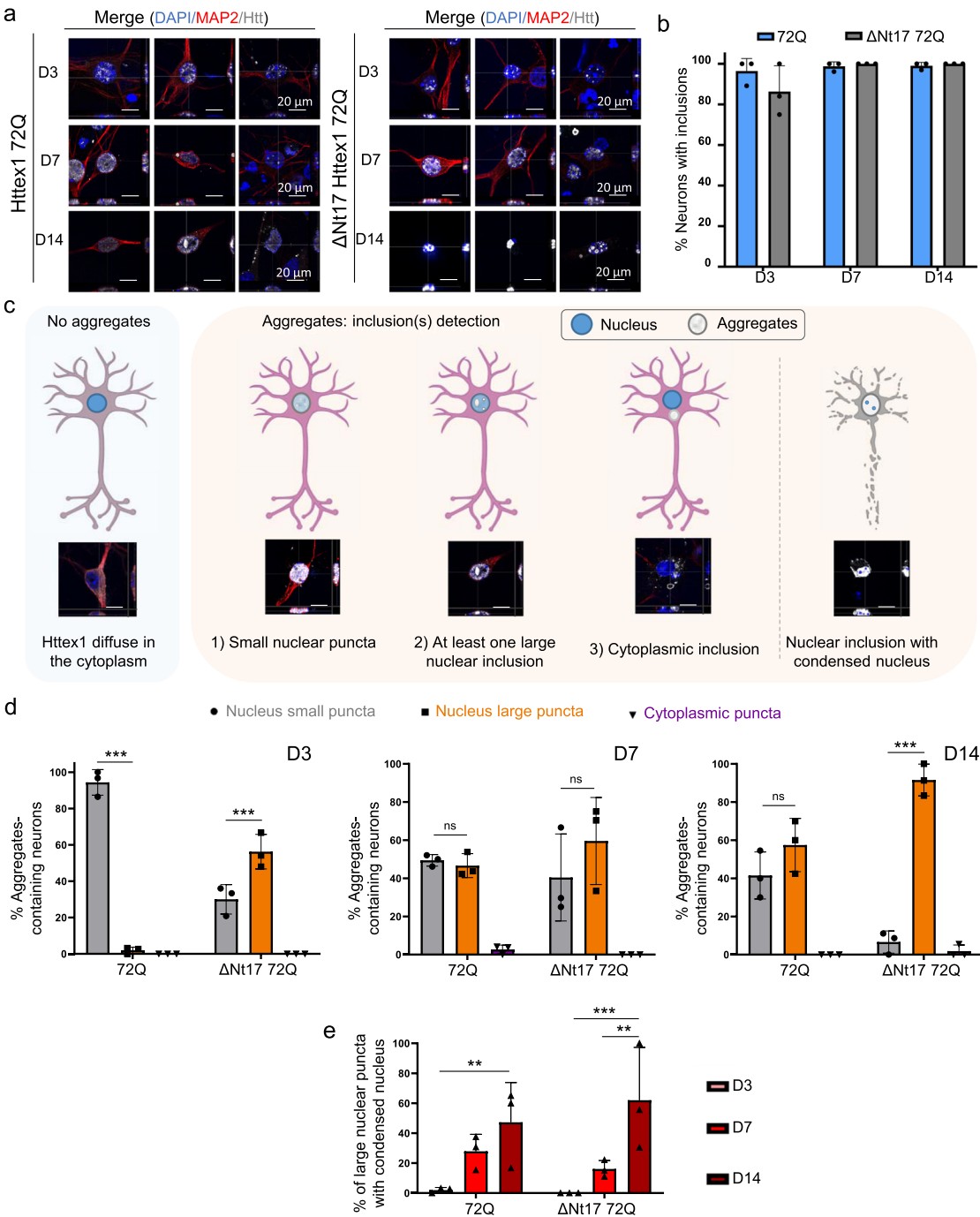

**Fig. 5 Confocal microscopy analysis and classification of the Httex1 inclusions formed in neurons revealed their morphological heterogeneity. a** Httex1 expression was detected by ICC staining combined with confocal imaging in primary cortical neurons, 3 (D3), 7 (D7), and 14 (D14) days after lentiviral transduction. Httex1 mutants were detected with the MAB5492 antibody. The nucleus was counterstained with DAPI (blue), and MAP2 was used to visualize the neurons (red). Scale bars = 20 μm. **b** Image-based quantification of the number of neurons containing Httex1 inclusions over time. The graphs represent the mean ± SD of 3 independent experiments. **c** Primary neurons were classified via the detection of Httex1 as diffuse or by the morphology of the detected Httex1 aggregates: (1) Small nuclear puncta; (2) at least one large nuclear inclusion, and (3) cytoplasmic inclusion. In addition, a subclass was created for neurons containing a large nuclear inclusion associated with nuclear condensation. Scale bar = 20 μm. **d** Image-based quantification and classification of the different morphologies of Httex1 inclusions based on the panel **c**. A minimum of $n = 145$ cells per condition were examined over 3 independent experiments. Data are presented as mean values + /− SD. Statistical analysis: One-way ANOVA followed by a Tukey [HSD] post hoc test was performed. $p$-values < 0.0001 for top and bottom panels and *$P < 0.05$, **$P < 0.005$, ***$P < 0.001$ for multiple comparisons. **e** Image-based quantification of neurons containing a large nuclear inclusion with a nuclear condensation. A minimum of $n = 145$ cells per condition were examined over 3 independent experiments. Data are presented as mean values + /− SD. Statistical analysis: Two-way ANOVA revealed no significant interaction ($p$-value = 0.0598) but a significant row factor samples ($p$-value = 0.0076) and column factor time ($p$-value = 0.0002) with *$P < 0.05$, **$P < 0.005$, ***$P < 0.001$ for multiple comparisons.

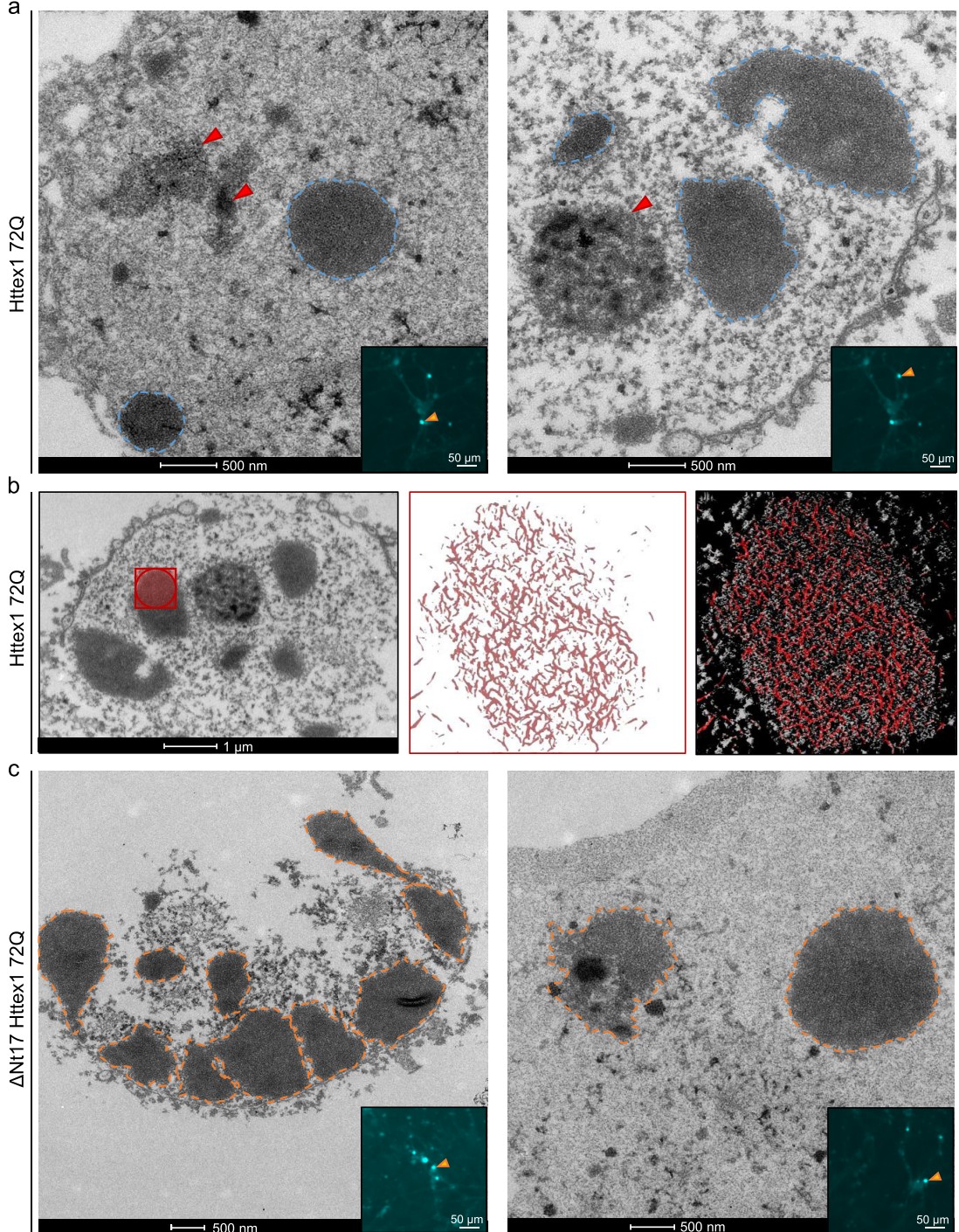

**Fig. 6 Ultrastructural analysis of the nuclear Httex1 72Q inclusions formed in primary cortical neurons shows granular and filamentous structures by CLEM and Tomography. a** Representative electron micrographs of Httex1 72Q inclusions formed 7 days after lentiviral transduction in mouse cortical primary neurons. Blue dashed lines represent the inclusions, and the red arrowheads the nucleolus. Scale bars = 500 nm for the EM images and 50 µm for the fluorescent image (inset). **b** Electron micrograph and corresponding tomogram (selected area in red) of a neuronal Httex1 72Q inclusion in panel a. Segmentation of the tomogram reveals the presence of a multitude of filaments. **c** Representative electron micrographs of ΔNt17 Httex1 72Q inclusions formed seven days after lentiviral transduction in mouse cortical primary neurons. Orange dashed lines represent the inclusions, and the red arrowheads the nucleolus. Scale bars = 500 nm for the EM images and 50 µm for the fluorescent image (inset).

alters the mechanism of Httex1 aggregation and inclusion formation.

To obtain an even clearer picture of these differences, we next performed EM under detergent-free conditions to preserve the internal membranes and structures of the inclusions. We

observed that the length of the polyQ repeat did not influence the size of the Httex1 39Q-GFP and 72Q-GFP inclusions (Supplementary Fig. 18b) or their overall architecture in HEK cells. The Httex1 39Q-GFP inclusions are composed of radiating fibrils and are thus similar to the Httex1 72Q-GFP inclusions,

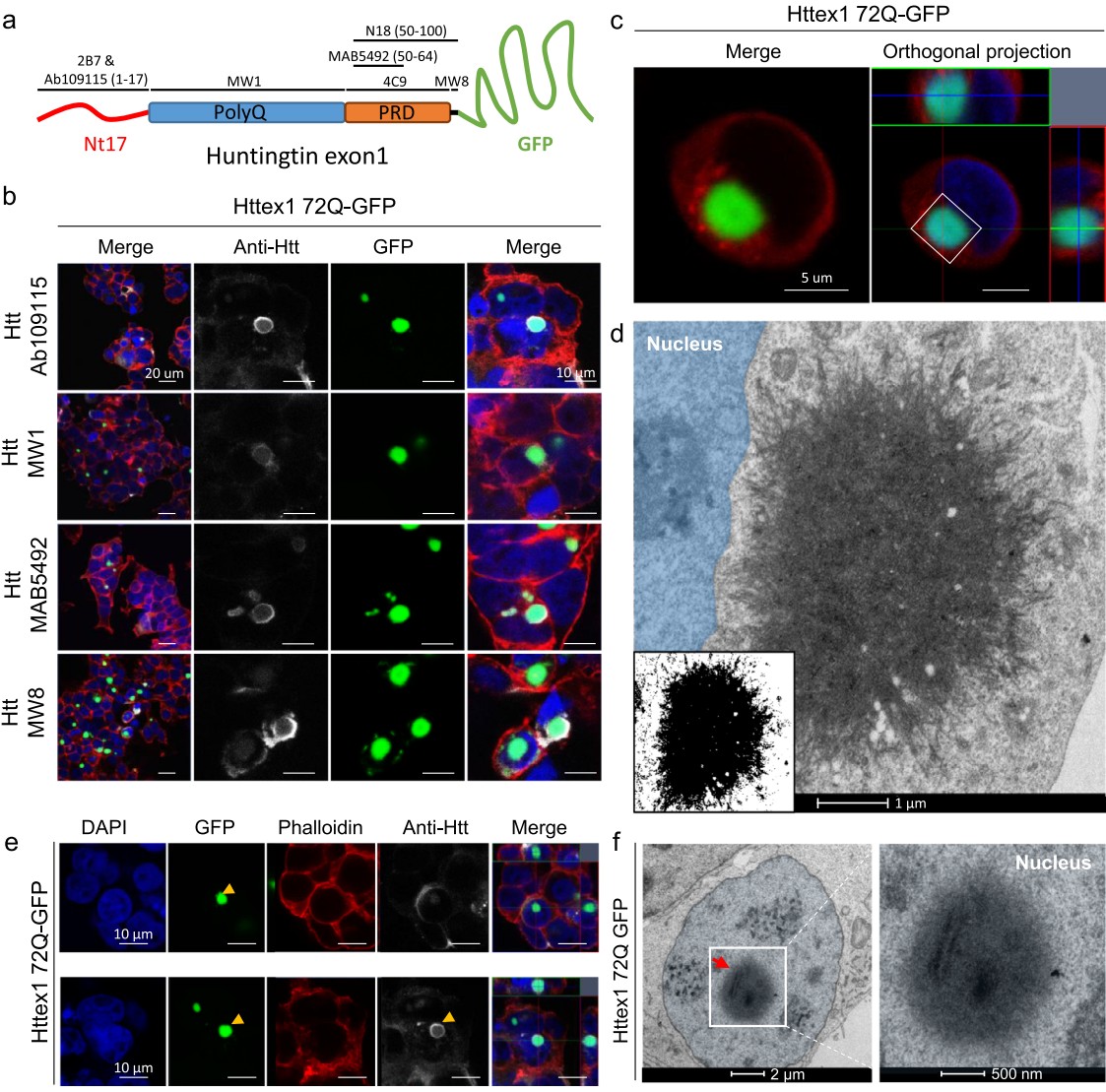

**Fig. 7 The addition of GFP to the C-terminal part of Httex1 induces a differential structural organization as revealed by ICC and CLEM. a** Epitope mapping of the Httex1 antibodies. **b** Httex1 72Q-GFP inclusions formed 48 h after transfection in HEK cells were detected by ICC staining combined with confocal imaging. All Htt antibodies showed strong immunoreactivity to the periphery of the Httex1 inclusions and modest immunoreactivity to the core. The nucleus was counterstained with DAPI (blue) and the F-actin with phalloidin (red). Scale bars = 20 μm (left-hand panels) and 10 μm (middle and right-hand panels). **c, d** 48 h post-transfection, HEK cells were fixed, and ICC against Httex1 was performed and imaged by confocal microscopy (**c**). Scale bars = 5 μm. The selected area of the cells (white square) was then examined by EM (**d**). A binary image (inset) was generated from the electron micrograph using a median filtering and Otsu intensity threshold, allowing for a better distinction of the inclusions' morphology. Scale bars = 500 nm. **e** Representative confocal images of Httex1 72Q-GFP nuclear inclusions formed in HEK cells 48 h after transfection. Httex1 expression (grey) was detected using a specific antibody against the N-terminal part of Htt (amino acids 1-17; 2B7 or Ab109115), and GFP (green) directly visualized in the appropriate channel. The nucleus was counterstained with DAPI (blue), and phalloidin (red) was used to stain filamentous actin. Httex1 nuclear inclusions are indicated by yellow arrowheads. Scale bars = 10 μm. **f** Electron micrograph of a representative Httex1 72Q-GFP inclusion. The white square indicates the area shown at higher magnification in the right-hand panel. The nucleus is highlighted in blue. Scale bars = 2 μm (left-hand panel) and 500 nm (right-hand panel).

although slightly less dense. The 3D reconstruction of the inclusions showed the presence of ER and mitochondria in their periphery, but few membranous structures were internalized (Supplementary Fig. 19b yellow arrowhead) as compared to tag-free Httex1 39Q and Httex1 72Q inclusions. A similar analysis of Httex1 ΔNt17 39Q-GFP and Httex1 ΔNt17 72Q-GFP cellular inclusions (Fig. S20a–d) also revealed no effect of Nt17 deletion on the ultrastructure of the inclusions or their interactions with the surrounding organelles. These results confirm that mutant Httex1 aggregation and inclusion formation mechanisms are significantly altered by the addition of the GFP.

We also assessed how the presence of the GFP tag might influence the kinetics of aggregation, the morphology, subcellular localization, and toxicity of the nuclear inclusions in primary cortical neurons (Fig. 8 and S21). First, we observed that the presence of the GFP tag slows down the aggregation rate of Httex1 72Q in contrast to the tag-free Httex1 72Q, the aggregation was significantly delayed, as evidenced by the absence of nuclear or cytoplasmic aggregates or inclusions at D3 (Fig. 8a, b). However, at D7, almost all the Httex1 72Q-GFP proteins appeared in the form of small nuclear puncta (~35%) or large nuclear inclusions (~60%) (Fig. 8c). The proportion of small puncta vs. large inclusions did not change over time, up to D14 (Fig. 8c).

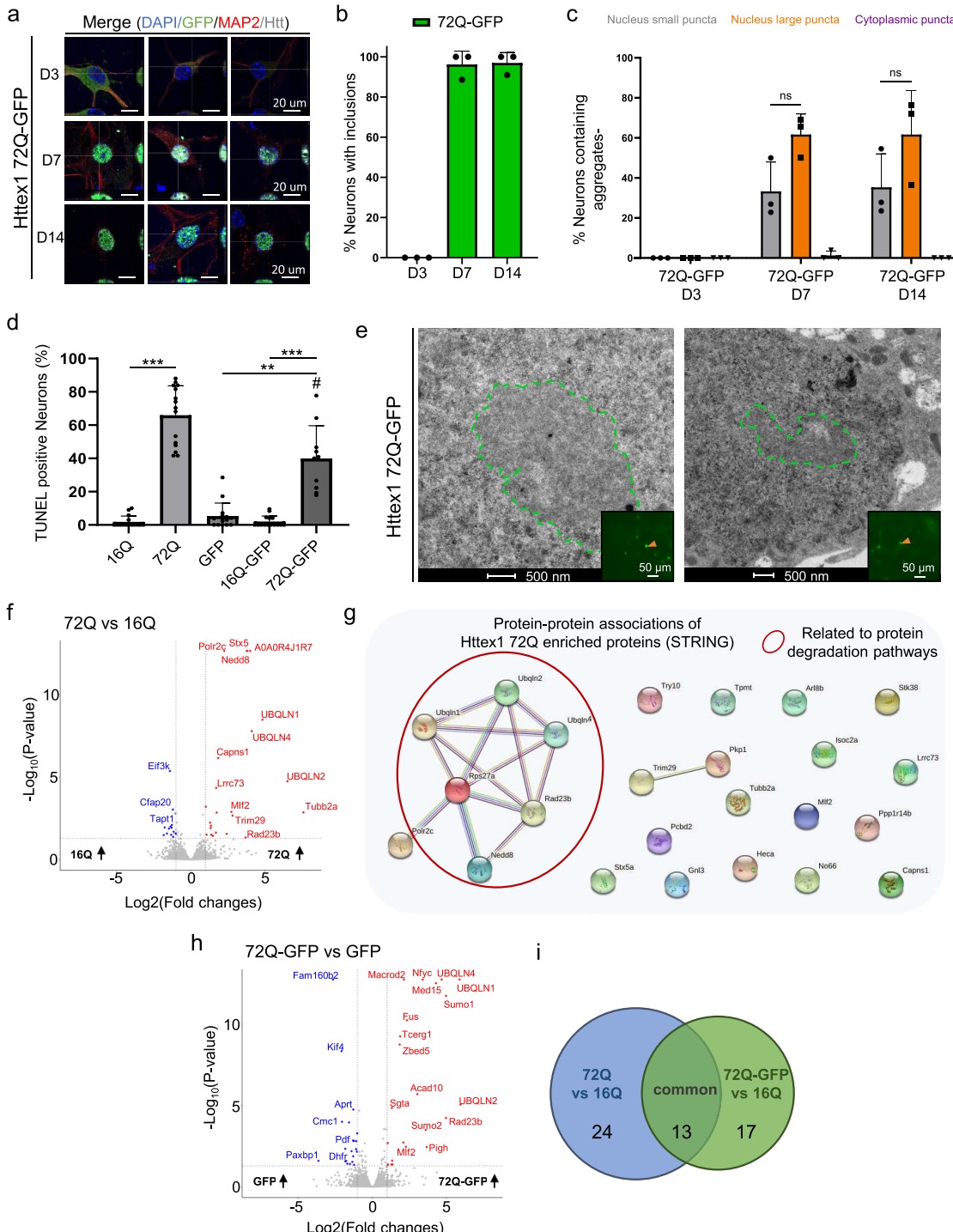

Interestingly, the subcellular localization, morphology, and size distribution of the inclusions observed by confocal imaging were not impacted by the presence of the GFP tag. However, the Httex1 72Q-GFP aggregates exhibited a significantly reduced toxicity compared to tag-free Httex1 72Q in neurons (Supplementary Fig. 21d). Even at D14, chromatin condensation did not exceed 10% in these neurons (Supplementary Fig. 21d), whereas ~50% of the neurons overexpressing the tag-free Httex1 72Q construct were already dead (Fig. 5e). In addition, the TUNEL cell death assay revealed a dramatic increase (~60%) in DNA fragmentation in the cortical neurons bearing the intranuclear Httex1 72Q inclusions compared to only 40% in neurons expressing Httex1 72Q-GFP (Fig. 8d).

Next, we investigated how the presence of the GFP tag might impact the ultrastructural properties of these neuronal inclusions (Fig. 8e and S22). The nuclear inclusions formed in the presence of the GFP tag exhibited a round shape but displayed a less dark staining density compared to the inclusions formed by the untagged Httex1 72Q protein (Fig. 6a). Due to their lower density, the presence of filamentous structures organized as an entangled arrangement throughout the GFP tagged inclusion could be detected. Finally, these inclusions were localized throughout the nucleus but always distant from the nuclear membrane (Supplementary Fig. 22).

Altogether, our results demonstrate that the addition of the GFP significantly slows the initiation of mutant Httex1

**Fig. 8 The GFP tag alters the kinetic, ultrastructural properties, protein content and toxicity of Httex1 72Q inclusions in neurons. a** Httex1 72Q-GFP was directly visualized with GFP (green) at 3 (D3), 7 (D7), and 14 (D14) days after lentiviral transduction. The nucleus was counterstained with DAPI, and neurons were stained with the MAP2 antibody. Scale bar = 20 μm. **b** Image-based quantification of the number of transduced neurons bearing Httex1 72Q-GFP inclusions over time. **c** Morphological and subcellular localization classifications (see Fig. 5c) of the 72Q-GFP inclusions formed in transduced neurons. $n = 219$ cells were examined over 3 independent experiments. **d** TUNEL cell death assay in cortical neurons transduced with Httex1. **b, c, d** The graphs represent the mean ± SD of 3 independent experiments. **b, d** One-way ANOVA followed by an HSD post hoc test was performed. *$P < 0.05$, **$P < 0.005$, ***$P < 0.001$ for multiple comparisons. #$P < 0.05$ (72Q vs. 72Q-GFP). **c** Two-way repeated-measures ANOVA revealed no significant differences. **e** Representative EM images of Httex1 72Q-GFP inclusions formed in cortical neurons at D7 post-transduction. Green dashed lines represent Httex1 inclusions. Scale bars = 500 nm and 50 μm (inset). **f-i** Urea soluble proteins extracted from cortical neurons expressing Httex1 72Q or Httex1 16Q (**f**) or Httex1 72Q-GFP or GFP (**h**) at D7 post-transduction were analyzed using LC-MS/MS. $n = 3$ independent experiments. Identified proteins were plotted using volcano plots (f, Httex1 72Q vs. Httex1 16Q; h, Httex1 72Q-GFP vs. GFP). Dotted lines represent the false discovery rate <0.05, and an absolute log2 fold change threshold of significance of 1 was assigned. Upregulated proteins are represented in red, downregulated in blue and non-significant proteins in grey. **g** The online platform STRING underscores an association between the upregulated Httex1 72Q proteins and protein related to the degradation pathways (red circle). **i** Venn diagram comparison of proteins enriched in Httex1 72Q vs. Httex1 16Q (blue) to Httex1 72Q-GFP vs. Httex1 16Q (green). 24 proteins were unique to Httex1 72Q vs. Httex1 16Q (44.4%), 17 unique to Httex1 72Q-GFP vs. Httex1 16Q (31.5%) and 13 (24.1%) common to both co-aggregated proteins. Source data are provided as a Source Data file.

fibrillization and inclusion formation in the nucleus but accelerates the maturation of the aggregates once formed, resulting in predominantly large inclusions with reduced toxicity in neurons. The lack of physical interaction between the Httex1 72Q-GFP inclusions and the nuclear envelope and/or the absence of small puncta aggregates in neurons expressing Httex1-72Q-GFP could explain their reduced toxicity (Fig. 8d, e).

**The Httex1 72Q and Httex1 72Q-GFP nuclear and cytoplasmic inclusions exhibit distinct proteome composition.** To gain further insight into the biochemical composition of the Htt nuclear inclusions, the molecular interactions and mechanisms driving their formation and maturation, we next investigated the proteome content of nuclear inclusions formed in primary neurons (Supplementary Fig. 23). Proteomic analysis showed that 23 proteins were significantly enriched in the Httex1 72Q inclusions compared to Httex1 16Q (Fig. 8f). These proteins were analyzed by protein-protein association using the online platform STRING (Fig. 8g) and classified using GO term analysis by Cellular component, Molecular function, and Biological process (Supplementary Fig. 24). The proteins identified include the transcription factors PCBD2 and Mlf2, the RNA/DNA binding protein (GNL3), DNA-dependent RNA polymerase (Polr2c), histone lysine demethylase (No66), DNA genome nucleotide excision repair (Rad23B), and proteins linked to chromosome segregation (ARL8B and Tubb2a). In line with our results, ARL8B, Tubb2A, Polr2 sub-units, Mlf2, GNL3, and Rad23B proteins were also found enriched in the insoluble fractions of Q175 HD mice brains[99] and Tubb2a in R6/2 mice[100] nuclear inclusions. Interestingly, the gene expression level of these nuclear proteins was also significantly increased in HD post-mortem brain (PCBD2)[101] and symptomatic HD patients (Mlf2, GNL3, Polr2, and ARL8B)[102]. Altogether, this suggests that the loss of key nuclear proteins due to their sequestration by the pathological Htt inclusion is compensated in neurons by increasing gene expression levels. However, this seems insufficient to compensate for the loss of their biological functions, as reflected by the high level of nuclear alterations observed in our neuronal HD model, including chromatin condensation, nuclear fragmentation, and nuclear envelope integrity loss (Fig. 6).

In addition to the nuclear proteins, STRING analysis (Fig. 8g), together with GO term classification of cellular components (Supplementary Fig. 24a), the molecular functions (Supplementary Fig. 24B), and the biological process annotations (Supplementary Fig. 24c) revealed the enrichment of a cluster of proteins [Ubiquilin (ubqln) 1, 2 and 4; Rps27a, Rad23B and Nedd8, red circle, Fig. 8g] related to the protein degradation machinery, including the UPS,

autophagy and ERAD pathways. In line with our findings, all these proteins were also found enriched in the insoluble fractions of the Q175 HD mice brains, while the Ubiquilin family members (Ubqln 1, 2, or 4)[103–105] were also shown to be sequestered in R6/2 mice nuclear inclusions[100] and in the Httex1 nuclear inclusions formed in the PC12 neuronal rat precursor[106]. Interestingly, ubiquilin proteins have been previously shown to translocate from the cytoplasm to the nucleus or have an upregulated expression level in the nucleoplasm concomitantly with the early stage of the formation of the neurofibrillary tangles in AD post-mortem brain tissues[107]. Furthermore, Nedd8, originally shown to colocalize with ubiquitin and the proteasome components in the cytoplasmic inclusions[108,109] found in several neurodegenerative disorders, has recently been shown to promote nuclear protein aggregation as a defense mechanism against proteotoxicity[110]. Moreover, our analysis revealed 5 proteins known as Htt interactors (based on the HDinHD database) among the proteins significantly different between Httex1 72Q compared to Httex1 16Q in primary neurons (Supplementary Data 1).

Finally, we also assessed how the presence of the GFP tag might influence the protein content of the neuronal intranuclear inclusions (Fig. 8h). Similar to the Httex1 72Q aggregates, ~45% of the proteins enriched in the insoluble fraction of the Httex1 72Q-GFP aggregates were related to nuclear biological processes and functions, including transcription factors (Mlf2[106], nfyc-1, TCERG1, Med15), DNA-chromatin binding proteins (ZBED5, N6AMT1, and Actl6b), DNA genome nucleotide excision repair protein (Rad23B), and the DNA/RNA-binding protein FUS. In addition, Httex1 72Q-GFP aggregates were also enriched in proteins from the promyelocytic leukemia nuclear bodies (Sumo 1 and 2) known as nuclear membraneless compartments involved in genome maintenance such as DNA repair, DNA damage response, telomere homeostasis, and which is also associated with apoptosis signaling pathways (Supplementary Fig. 25). Furthermore, proteins from the ubiquitin-proteasome system (Bag5, Fus, Rad23b, Sgta, Sumo1, Sumo2, Ubqln1, Ubqln2, Ubqln4), the ERAD (Sgta, Ubqln1, Ubqln2), autophagy (Ubqln1, Ubqln2, Ubqln4), and sumoylation (Sumo1, Sumo2) pathways were also highly enriched in the insoluble fraction of the Httex1 72Q-GFP aggregates (Supplementary Fig. 25a, b). Interestingly, the proteome of the Httex1 72Q-GFP aggregates formed in our HD neuronal model shared 40% of the proteins also found enriched in the insoluble fraction of Httex1 74Q-GFP nuclear inclusions formed in the PC12 neuronal rat precursor cell line[106]. This highlights a strong similarity between the pathways and proteins involved in the aggregation of Httex1 and inclusion formation in rodent neuronal cells.

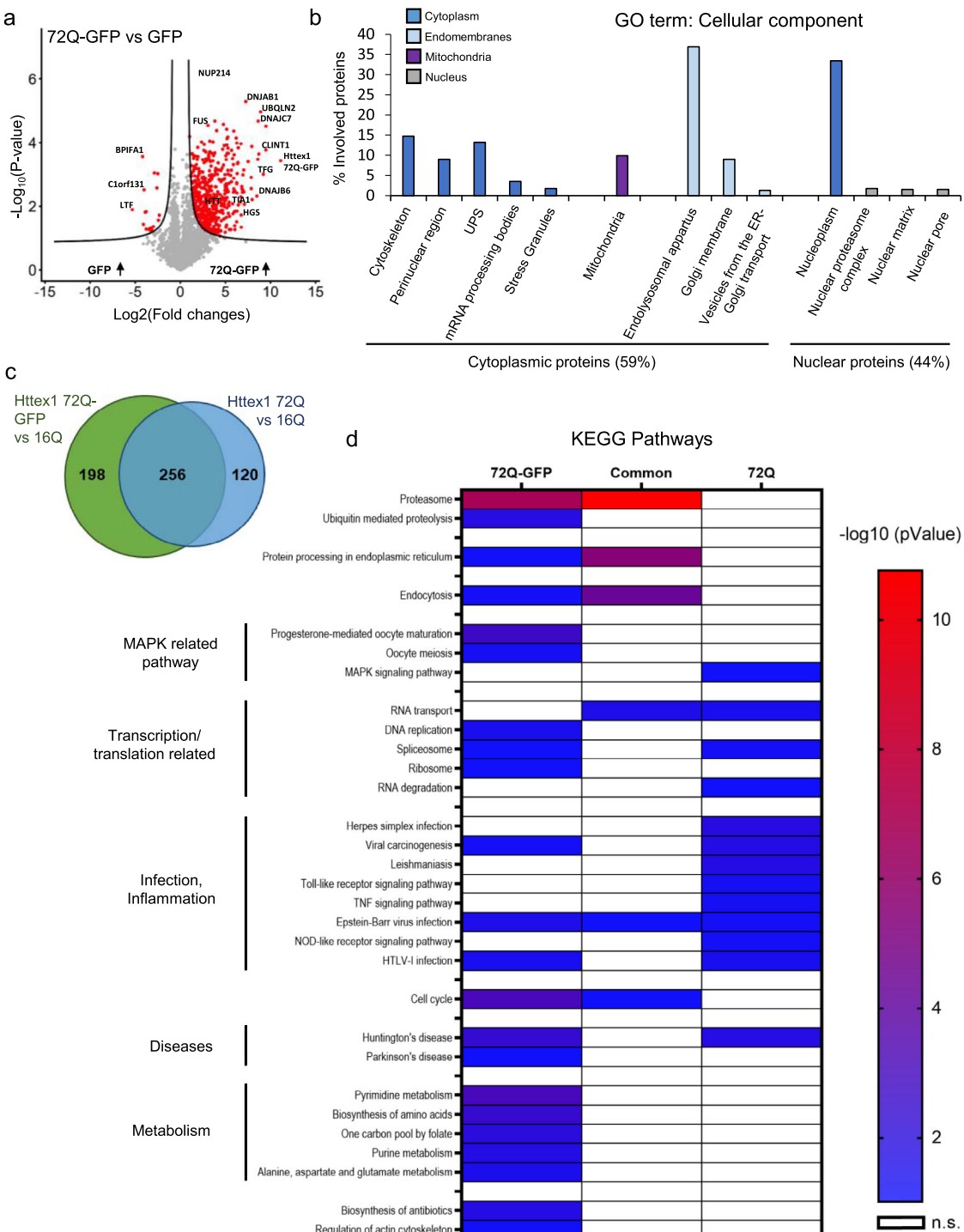

We next compared the proteins identified between tag-free Httex1 72Q inclusions and Httex1 72Q-GFP (Fig. 8i). 24 proteins were unique to Httex1 72Q (44.4%), 17 unique to Httex1 72Q-GFP (31.5%) and only 13 (24.1%) common to both type of inclusions. Most of the unique proteins found in the tag-free or GFP-tagged Httex1 72Q inclusions were related to nuclear functions, the ubiquitin-proteasome system, ERAD, and autophagy pathways. Moreover, among the 13 common proteins, most were linked to the protein degradation machinery (i.e., ubiquilins 1, 2, and 4, and the Rad23b and Nedd8 proteins).

Although the tag-free Httex1 72Q and Httex1 72Q-GFP inclusions share several proteins related to the degradation machinery, more than 75% of the co-aggregating proteins are different, which could explain the drastic differences in neuronal toxicity associated with each type of these inclusions.

As the majority of the inclusions formed in neurons are nuclear, we used HEK cells to further assess how the GFP tag influences the proteome of the cytoplasmic Httex1 inclusions (Supplementary Fig. 11a). Volcano plot analysis showed 492 proteins significantly enriched in the insoluble fraction of the HEK cells overexpressing Httex1 72Q-GFP as compared to those expressing GFP (Fig. 9a). We observed a significant enrichment of the endogenous HTT protein, suggesting that, as for the tag-free Httex1 72Q, mutant Httex1-GFP inclusions can recruit the endogenous HTT protein (Supplementary Fig. 11e).

**Fig. 9 Proteomic analysis of Httex1 72Q-GFP in the Urea soluble fraction revealed 55% differences compared to tag-free Httex1 72Q for the proteins co-aggregated with Httex1 in HEK cells. a** Identified proteins were illustrated using a volcano plot for the comparison of protein levels identified in the Urea soluble fraction 48 h after Httex1 72Q-GFP or GFP transfection in HEK cells, in 3 independent experiments. Mean difference (Log2 (Fold-Change) on the X-axis) between the Urea soluble fraction of HEK cells overexpressing Httex1 72Q-GFP or GFP were plotted against significance (Log10 (p-Value) on the Y-axis (T-Test)). A false discovery rate (FDR) of 0.05 and a threshold of significance S0 = 0.5 were assigned for the subsequent analyses. **b** Cell compartment classification of the proteins significantly enriched in the Urea soluble fraction of the HEK cells overexpressing Httex1 72Q-GFP versus those expressing GFP. Gene Ontology (GO) enrichment analyses were determined by DAVID analysis (Significant for p values < 0.05 adjusted for multiple testing based on Benjamini and Hochberg). **c** To compare the results obtained with Httex1 72Q-GFP to tag-free Httex1 72Q, we represented the proteins enriched in Httex1 72Q vs. Httex1 16Q compared to the proteins enriched in Httex1 72Q-GFP vs. Httex1 16Q using a Venn diagram. In total, 256 proteins (44.6%) were found similar between the two conditions, 198 proteins (34.5%) were unique for Httex1 72-GFP enrichment, and 120 proteins (20.9%) were for Httex1 72Q enrichment. **d** Kyoto Encyclopedia of Genes and Genomes (KEGG) Pathway Analysis of co-aggregated proteins with Httex1 72Q and Httex1 72Q-GFP. The heat map represents significant pathways (KEGG) enriched in the Urea soluble fractions extracted from the Venn diagram (**c**): co-aggregated proteins unique to Httex1 72Q, unique to Httex1 72Q-GFP, and common to both conditions, all compared to Httex1 16Q control. Source data are provided as a Source Data file.

The cytoplasmic components enriched in the insoluble fraction of Httex1 72Q-GFP were similar to those found with tag-free Httex1-72Q and were part of the endomembrane system (~50%, light blue Fig. 9b), the cytoskeleton, the perinuclear region, the UPS, mRNA processing bodies, and stress granules (Fig. 9b, dark blue). However, in contrast to the tag-free Httex1 72Q insoluble fractions, in which no mitochondrial protein was found to be enriched, 10% of the proteins enriched in the Httex1 72Q-GFP insoluble fraction were related to the mitochondria compartment (purple Fig. 9b). The nuclear proteins found in the Httex1 72Q-GFP insoluble fraction belong to similar nuclear compartments as those identified previously in the Httex1 72Q insoluble fraction (Fig. 9b, grey).

Biological process (Supplementary Fig. 26) and canonical pathway (Supplementary Fig. 27) analyses revealed that the UPS and chaperone machinery were the most enriched terms, as previously observed in the tag-free insoluble fraction. This indicates that the process of aggregation of Httex1 itself, regardless of the presence of the GFP tag, leads to the sequestration of the key cellular machinery responsible for protein folding and protein degradation. This could compromise the cell's ability to prevent the accumulation or clearance of Httex1 aggregates. Although in different proportions, most of the biological processes and pathways significantly enriched in the tag-free Httex1 72Q insoluble fraction, such as inflammation, transcription, HD signaling, and cell death, were also detected in the Httex1 72Q-GFP insoluble fraction.

Having identified the proteins significantly enriched in both tag-free and GFP-tag 72Q inclusions, we next determined which proteins were unique to each type of inclusion. Toward this goal, we used a Venn diagram to compare the lists of proteins significantly enriched in our volcano plot analyses [(Httex1 72Q vs. Httex1 16Q insoluble fractions) vs. (Httex1 72Q-GFP vs. Httex1 16Q insoluble fractions)] (Fig. 3a and S11c). Figure 9c shows that ~45% (256 proteins) were found in both the tag-free and the GFP-tag Httex1 72Q inclusions. In total, 120 proteins (20.9%) were unique to the Httex1 72Q insoluble fraction, and 198 proteins (34.5%) were unique to the Httex1 72Q-GFP. Overall, we found 55% different proteins among the proteins that co-aggregate with or are sequestered in Httex1 72Q vs. 72Q-GFP inclusions, both compared to Httex1 16Q. We used the Kyoto Encyclopedia of Genes and Genomes (KEGG) pathway analysis to classify the list of proteins in Httex1 inclusions (Fig. 9d). The proteasome, protein processing in the endoplasmic reticulum, and endocytosis were among the most enriched pathways from the common co-aggregated proteins. Those 3 terms were also enriched significantly for Httex1 72Q-GFP, indicating the involvement of other unique proteins enriched in those pathways compared to Httex1 72Q. The KEGG classification revealed two distinct clusters for Httex1 72Q and Httex1 72Q-GFP, as well as

additional differences (Fig. 9d). Proteins related to infection and inflammatory pathways (e.g., Herpes simplex infection, TNF signaling, and Toll-like receptor signaling) were found to be unique to Httex1 72Q, whereas proteins related to metabolism were specifically enriched for Httex1 72Q-GFP. Our proteomic analysis highlights that the addition of a fluorescent tag such as GFP significantly alters not only the mechanism of Htt inclusion formation but also the Htt interactome and thus the biochemical composition of inclusions.

**Httex1 72Q inclusions induce higher respirational dysfunction and a stronger decrease of ERES number compared to Httex1 72Q-GFP in HEK cells.** Next, we investigated how the aggregation of mutant Httex1-GFP and the formation of cytoplasmic inclusions affect the mitochondrial and ER-related functions. First, we measured the mitochondrial respiration in different respirational states in HEK cells (Supplementary Fig. 13c, d). Compared to our findings with the tag-free mutant Httex1 72Q (Fig. 4c), the increased mitochondrial respiration was significant but less pronounced for Httex1 72Q-GFP than for Httex1 16Q-GFP (Supplementary Fig. 13e). Given the role of mitochondria as a major source of reactive oxygen species (ROS), with implications of ROS in both neurodegenerative disease and cellular protective signaling cascades[111,112], we suspected that mitochondrial ROS production might be affected by the mitochondrial fragmentation observed in regions close to tag-free Httex1 72Q and absent for Httex1 72Q-GFP (Supplementary Fig. 28a, b). To test this hypothesis, we used an amplex red assay to measure mitochondrial ROS (superoxide and hydrogen peroxide) production concurrent with mitochondrial respiration (Supplementary Fig. 13d). We observed no significant differences in mitochondrial ROS production between the tag-free or GFP tagged Httex1 72Q or 16Q (Supplementary Fig. 28c).

Mitochondrial fragmentation[113] and inflammation[114] are hallmarks of Htt-induced neurodegeneration. Recent studies reported a major role of mitochondrial fission and fusion homeostasis in HD and also revealed specific mitochondrial fragmentation induced by mutant Htt using electron microscopy in STHdh neurons[115,116]. Interestingly, only tag-free Httex1 72Q overexpression results in mitochondrial fragmentation and is characterized by the strong accumulation of inflammation-linked proteins in the aggregates. We speculate that the observed hyperactivation of mitochondrial respiration (in the absence of significantly increased ROS production) is an adaptation of the inclusion-bearing cells in order to generate sufficient energy by oxidative phosphorylation for increased energetic demands for unfolding or clearance of aggregating proteins. A similar hyperactivation of mitochondrial respiration has recently been reported for alpha-synuclein aggregates[117]. These authors hypothesized that the hyper-respiration might represent a

pathogenic upstream event to alpha-synuclein pathology. We think that this may also be the case for untagged Httex1 72Q induced pathology. However, based on our extensive proteomic analyses of the inclusions formed by Httex1 72Q or Httex1 72Q-GFP and comparing the functionality of organelles in our cellular model, we can provide a more elaborate hypothesis. In the tag-free Httex1 72Q condition, mitochondrial hyperactivity coincided with both mitochondrial fragmentation and the detection of more inflammation-related proteins in the aggregates as compared to the GFP condition. We speculate that mitochondrial fragmentation initially may be protective, as it enhances mitochondrial mobility necessary for cell repair processes[118]. Upregulation of mitochondrial respiration would be a reasonable consequence of elevated ATP requirements for mitochondrial and protein transport related to aggregation formation and proteostatic processes including ATP-requiring chaperone-, proteasome and autophagy processes.

Finally, we also investigated whether Httex1-GFP inclusions interfere with the homeostasis of ERES by comparing the size and the number of ERES in HeLa cells overexpressing Httex1 16Q-GFP to those expressing mutants GFP-tagged Httex1 39Q or 72Q (Supplementary Fig. 29a). We found that cells containing Httex1 72Q-GFP inclusions caused a decrease in ERES number (Supplementary Fig. 29b), although non-significant, compared to Httex1 72Q, which showed a 20% reduction (Fig. 4e). The reduction of ERES size was significant for cells expressing Httex1 72Q-GFP but not Httex1 39Q-GFP compared to Httex1 16Q-GFP (Supplementary Fig. 29c). Thus, the effect of Httex1-GFP inclusions on ERES is present but is less pronounced compared to that observed in cells containing tag-free Httex1 inclusions. These observations are consistent with recent cryo-ET studies suggesting decreased ER dynamics near Httex1-GFP inclusions[12].

Overall, our results showed that the cellular organelle responses and adaptation to inclusion formation were different for the Httex1 72Q-GFP inclusions than for the Httex1 72Q inclusions, consistent with the GFP-dependent changes observed in the proteome and the ultrastructural properties of Httex1 cytoplasmic inclusions.

## Discussion
Pathological inclusions in neurodegenerative diseases have been shown to have a complex organization and composition. For example, it has been recently demonstrated that Lewy bodies (LB) isolated from Parkinson's disease brains or LB-like inclusions in primary neurons are composed of not only filamentous and aggregated forms of alpha-synuclein but also a complex milieu of lipids, cytoskeletal proteins, and other proteins and membranous organelles, including mitochondria and autophagosomes[119–122]. Studies in neuronal cultures also showed that the recruitment of lipids and membranous organelles during LB formation and maturation contribute to organelles' dysfunctions and lead to synaptic dysfunction and neurodegeneration. In line with these findings, our EM data, together with 3D reconstructions, revealed the presence of membrane fragments and vesicles entrapped in the core of the Httex1 inclusions. Consistent with these observations, our proteomic analysis revealed that 24% of the proteins enriched in the inclusions fraction belong to the endolysosomal compartments, the Golgi apparatus, and the trans-Golgi network (Fig. 3). In addition, mitochondria and ER were found in the periphery of inclusions, as previously reported in cellular models[15] and human tissue[95,123]. We hypothesized that sequestration of key functional proteins, together with lipids, endomembranes, and organelles inside the Httex1 inclusions, could challenge cellular homeostasis. In line with this hypothesis, our electron micrographs revealed that the mitochondria associated with Httex1 72Q inclusions were

fragmented and often exhibited disorganized or depleted cristae (Fig. 4a). These changes in mitochondrial morphology were associated with dysregulation of the mitochondrial respiration (Fig. 4c), consistent with previous studies demonstrating that mutant Htt aggregates interact directly with outer mitochondrial membranes (in STHdh cells)[124] and induce mitochondrial fragmentation (in primary neurons)[125]. Defects in mitochondrial respiration have also been observed in HD patients' brains, especially defects of complexes II and IV of the respiratory chain may impair oxidative phosphorylation[126–128].

The ER at the periphery of the Httex1 72Q inclusions were also affected at the structural level, as shown by their morphological reorganization in rosette or "stacked cisternae", and at the functional level, as evidenced by the dysregulation of the ERES homeostasis (Fig. 4d-f). The formation of ER rosettes indicates the accumulation of proteins in the smooth ER[92] and is thought to result from low-affinity binding and export defects, which can be caused by unfolded Htt proteins but is not necessarily linked to ER stress. Kegel and colleagues[129] also reported the presence of ER cisternae next to aggregates formed by the expression of a large Htt fragment fused to a FLAG tag in clonal striatal cells. The formation of Httex1 inclusions seems to drive the reorganization of the ER network in their periphery but did not lead to ER protein sequestration, as shown by the quantitative proteomic analysis, which did not reveal significant ER proteins trapped inside the inclusions. The presence of ribosomes and ER membrane deformation was previously detected close to the periphery of Httex1 inclusions by cryo-ET[12] and was linked to a strong reduction in ER dynamics. In the same study[12], the periphery of the inclusions was immunoreactive to several components of the ER-associated degradation (ERAD) machinery (e.g., Erlin-2, Sel1L). These observations are consistent with our ICC data showing the presence of ER chaperone BIP at the outer periphery but not inside the inclusions (Supplementary Fig. 5). We also observed ERES modulation, which can be explained by the sequestration of specific proteins required for their fusion[130]. Consistent with these observations, our proteomic analysis showed the enrichment of proteins specific to the ER-Golgi trafficking. ERES modulation can also be explained by a reduction of the biosynthetic capacity of this compartment, as ERES have been shown to adapt to the amount of secretory burden to which they are exposed[131]. The recruitment and perturbation of the ER network might contribute to cytotoxicity during inclusion formation, but on its own does not appear to be sufficient to cause overt toxicity. Consistent with this hypothesis, the ERAD and $Ca^{2+}$ signaling pathways have been extensively reported to be dysregulated in various cellular and animal models using different Htt fragments[132–137]. Overall, we showed that cytoplasmic Httex1 inclusion formation and maturation involves a complex interplay between Httex1 aggregates and organelles.

Although previous studies have suggested the formation of ring-like nuclear inclusions, this architecture was defined by the peripheral staining of nuclear proteins and not by the peripheral staining of Htt as observed in the case of cytoplasmic inclusions (Fig. 1). Our in-depth characterization of the Httex1 72Q inclusions at the ultrastructure level revealed distinct differences in the organization and architecture between the cytoplasmic and nuclear Htt inclusions. The cytoplasmic Httex1 72Q inclusions exhibit a dense core and shell morphology and are composed of highly organized fibrillar aggregates in both the core and the periphery. In contrast, nuclear Htt inclusions, in both HEK cells or primary cortical neurons, exhibited less complex ultrastructural properties, appeared as homogenous structures without the core and periphery morphology and were devoid of endomembranes or vesicles (Figs. 2 and 6a). These observations are consistent with most of the Htt nuclear inclusions found in HD

mice models, which do not display the core and shell organization[19,32].

Our new data on nuclear neuronal Htt inclusion revealed that tag-free mutant Httex1 expression in primary neurons leads to the formation of granulo-filamentous nuclear inclusions, similar to previous ultrastructural studies performed in HD patients brains and in vivo models of HD, including R6/2 mice[19,32] and transgenic rats[138]. However, in most of these studies, the ultrastructural properties of the filaments were less evident compared to the well-resolved filamentous structure we and others[12] showed using electron tomography. For example, Tagawa and colleagues also observed nuclear inclusions of tag-free Httex1, but their data did not reveal the level of ultrastructural details and insights that our study provides[139].

Our results also suggest that the intracellular environment and interactions with lipids and endomembranes or vesicles are key determinants of the structural and molecular complexity of the inclusions and that different mechanisms drive the formation and maturation of the nuclear and cytoplasmic inclusions. Such differential characteristics between cytoplasmic and nuclear inclusions might account for the differential cellular dysfunction and toxicity associated with these two types of Htt inclusions[29,140–143]. Furthermore, the differential toxicity seen for Htt nuclear inclusions in primary neurons compared to HEK 293 cells indicate a cell-type dependent Htt toxicity that might arise from distinct Htt mechanisms of aggregation or interaction with cellular proteins or differences in the resilience of the different cells based for example on distinct oxidative stress tolerance and immune functions/inflammation.

We speculate that the formation of the core of the inclusions is guided predominantly by intermolecular interactions involving the polyQ domain and could be initiated by rapid events that are potentially driven by phase separation, as recently described by several groups[14,39,144,145]. The rapid formation of the aggregate's core does not allow for the regulated recruitment of other proteins and organelles (Figs. 1, 2, and 10a), as evidenced by the fact that most antibodies against proteins found in Htt inclusion stain the periphery rather than the core of the inclusions[48]. The Nt17 domain most likely plays key roles in the initial oligomerization events and, possibly, the packing or lateral association of the fibrils. Once this core of dense fibrils is formed, the fibrils at the periphery continue to grow through the recruitment of endogenous soluble proteins (Figs. 3 and 10a). Because this growth phase is slower, it allows the fibrils to interact with and/or recruit other proteins into the inclusions. Consistent with this hypothesis, Matsumoto and colleagues showed that the transcription factors TATA-binding protein (TBP) and CREB-binding protein (CBP)—containing a polyQ expansion—were present only at the periphery of the inclusions and not in the core[47]. Interestingly, although the length of the polyQ repeat did not seem to significantly influence the density of the fibrils at the core of the inclusions, the peripheral organization of the Htt fibrils and the formation of the outer shell showed strong polyQ repeat length dependence (Figs. 4 and 10a). This model is supported by studies from Hazeki et al. and Kim et al., demonstrating that the detergent-insoluble core of cellular Httex1 inclusions represents the skeletal structure of the inclusions, while the active surface dynamically interacts with Htt species and proteins being processed[30,38]. Altogether, our results highlight that the formation of Htt inclusions occurs in two major phases with, first, the formation of the core driven primarily by the polyQ repeat domain, and then the growth of the inclusions with the addition of Htt fibrils and the recruitment of other proteins and organelles. The second phase appears to be driven by interactions involving both the polyQ and PRD domains and involves the active recruitment and sequestration of lipids, proteins, and

membranous organelles (Fig. 10a). However, we cannot rule out the presence of entrapped oligomers in the core or at the surface close to the growing fibrils in the periphery[146]. Previously, Qin et al., using a FLAG-tagged Htt1-969 fragment (100Q), demonstrated that the Htt species in the core of the inclusions are highly protease-resistant and that oligomeric forms of Htt accumulate with cytoplasmic proteins and vesicles in the periphery of the inclusions[48]. In our study, we demonstrated that cytoplasmic and nuclear Htt inclusions exhibit distinct ultrastructural properties that are differentially influenced by the size of the polyQ repeats.

To the best of our knowledge, this is the first study to directly compare inclusions formed by tag-free mutant native Httex1 proteins to GFP-tagged mutant Httex1 at the structural, interactome, and biochemical levels.

The ultrastructure of Httex1 72Q-GFP inclusions revealed radiating fibrils with homogenous staining and the absence of the core and shell morphology. In line with our findings, previous studies showed that cytoplasmic Httex1 72Q-GFP aggregates are fibrillar[14,147] and resemble other Httex1-GFP aggregates in COS cells[148], in primary striatal neurons[149] as well as aggregates formed by mutant Httex1 fused to a FLAG tag in HEK cells[15]. In a recent cryo-electron tomography study, Bäuerlein et al. examined the ultrastructural properties of mutant Httex1 in the absence and presence of GFP and suggested that the presence of GFP does not significantly alter the organization of the cytoplasmic inclusions in HeLa and neurons[12]. Similar to our findings, they showed that the presence of the GFP tag 1) induces a 50% reduction in fibril density and a 25% increase in fibril stiffness; 2) results in increased spacing between the fibrils in the Httex1-GFP inclusions; and; 3) does not influence the overall size of the inclusions. However, when it comes to the ultrastructural properties of the tag-free mutant Httex1 inclusions, we observed very different results, as they did not observe the formation of inclusions with core and shell morphologies. Examination of the sequence of the constructs they used reveals a Myc-tag at the N-terminal side of their Httex1 proteins, whereas the constructs we used contain only the native sequence of mutant Httex1. While we showed that removal of the Nt17 domain does not influence the ability of tag-free Htt to form inclusions with the core and shell organization, the fusion of additional amino acids in the N-terminus could still significantly alter the kinetics, oligomerization, and aggregation pathway of Httex1 protein. In addition to differences at the ultrastructural levels, our studies showed the recruitment of membranous organelles and vesicles as disrupted structures inside the inclusions in both the core and shell but also in the form of intact ER-mitochondrial network at the periphery of the tag-free inclusions. Less internalized structures were observed in the Httex1-GFP inclusions, consistent with the findings of Bäuerlein et al.[12] Another major difference between the two studies is that Bäuerlein et al. did not observe any polyQ-length effect on the size or ultrastructural properties of the inclusions formed by mutant Httex1-GFP over the polyQ repeat range of 64Q to 97Q. Although the presence of GFP can explain the differences between the two results, it is also possible that these differences are because the authors compared polyQ repeats that are far beyond the pathogenic threshold. We think that in this polyQ range, the difference in aggregation properties between 64 and 97Q would be small compared to what one would observe when comparing mutant Httex1 with polyQ repeats close to the pathogenic threshold (39-43Q). Therefore, in our study, we compared the ultrastructure properties of the inclusions of Httex1 containing 39 or 72Q repeats. In our experiments, the cytoplasmic inclusions formed by mutant Httex1 39Q were found to be less organized and did not exhibit a stable core and shell arrangement as in the case of Httex1 72Q. Finally, in primary neurons, the use of Httex1 72Q-GFP changed dramatically the

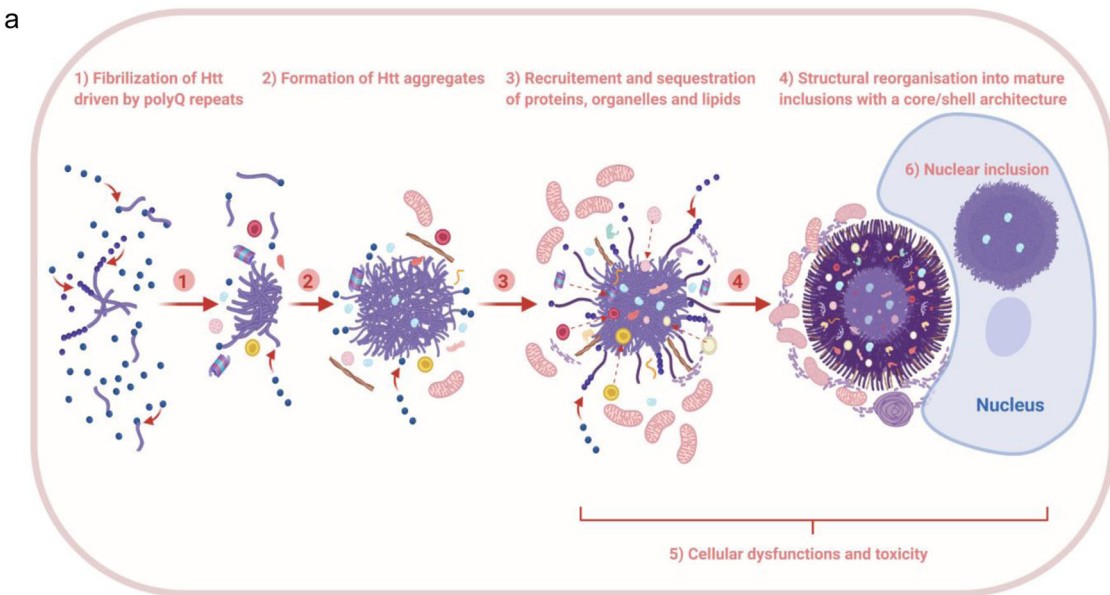

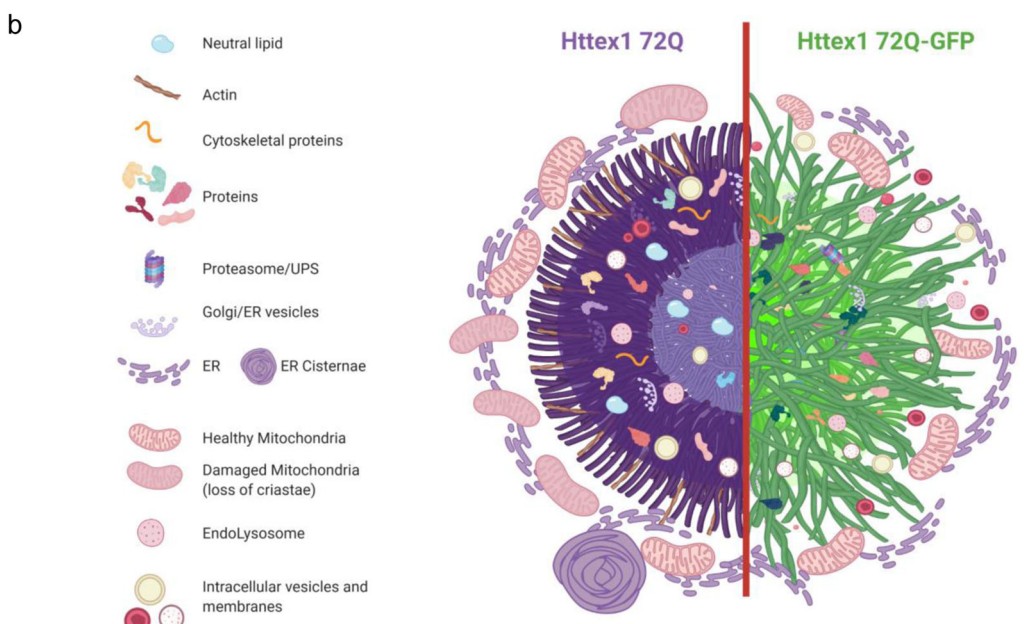

**Fig. 10 The formation of Huntingtin inclusions is driven primarily by the polyQ repeat domain and involves the active recruitment and sequestration of lipids, proteins, and membranous organelles. a** We separated the formation and organization of cytoplasmic Httex1 inclusions into different elements: (1) Even though the Nt17 domain can play a major role in early oligomerization steps, we showed that the fibrilization is driven by the polyQ repeats; (2) The initiation of inclusion formation was previously demonstrated to occur in a phase transition mechanism[14,31] and could involve first the sequestration of proteins and organelles nearby either directly or indirectly; (3) During the growth of the inclusion, many cellular proteins, endomembranes, and lipids are recruited and sequestered inside the cytoplasmic inclusions; (4) Mature cytoplasmic Httex1 inclusions formed in cells display a core and shell structural organization if the polyQ length is relatively long (72Q) but not if it is close to the pathogenic threshold (39Q), even if inclusions are still formed in the second case. Cytoplasmic inclusion formation leads to the accumulation of mitochondria and ER network at the periphery; (5) Httex1 inclusion formation leads to the adaptation of the ER-mitochondrial network and respirational dysfunction, as well as significant toxicity after long-term incubation; (6) Nuclear inclusions do not display a distinct core/shell organization similar to the cytoplasmic inclusions, but they are still detected as a ring, as they are impervious for antibodies. Nuclear tag-free Httex1 inclusions are also enriched in neutral lipids. **b** The depiction shows the distinct structural organization and cellular impact of Httex1 72Q (left) compared to Httex1 72Q-GFP (right). The arrangement and packing of Httex1 fibrils are different depending on the presence of GFP as well as the interactions, recruitment, and perturbation with surrounding organelles.

kinetics of inclusion formation and the proportion of small and large nuclear aggregates compared to tag-free Httex1 72Q. Moreover, neuronal Httex1 72Q-GFP nuclear inclusions were smaller and less dense compared to the tag-free Httex1 72Q inclusions, exhibited a distinct proteome composition, and interacted less with the nuclear membrane. These differences could explain the reduced toxicity of Httex1 72Q-GFP inclusions compared to tag-free Httex1 72Q inclusions.

Altogether, our data suggest that the aggregation processes of tag-free and GFP-tagged Httex1 are distinct (Fig. 10b and S30), thus underscoring the potential limitations of using GFP to investigate the molecular, biochemical, and cellular determinants of Htt inclusion formation and mechanisms of toxicity.

In summary, our integrative imaging and proteomic studies demonstrate that the process of Htt cytoplasmic inclusion formation occurs in at least two phases and involves the active recruitment of lipids, proteins, and organelles. The organization and ultrastructural properties of these inclusions are greatly influenced by the polyQ repeat length and the intracellular environment, as evidenced by the fact that cytoplasmic and nuclear Httex1 inclusions exhibit distinct ultrastructural properties. Equally important, our work emphasizes the importance of elucidating the role of Htt interactions with lipids and membranous organelles in regulating the process of Htt aggregation, inclusion formation, and toxicity. Extending the studies and approaches presented here to other HD model systems and other Htt fragments, will pave the way to a better understanding of the mechanisms of Htt inclusion formation and how it contributes to the development of HD. We recently demonstrated that longer N-terminal Htt fragments such as Htt171 aggregate in vitro via mechanisms that are distinct from those observed for mutant Httex1 where the structured domains outside exon1 could play important roles in regulating early events associated with Htt oligomerization, initiation of Htt aggregation, and inclusion formation[150].

Taken together, our work provides important and novel insights that not only advance our understanding of the mechanisms of Htt aggregation but also point to new directions for therapeutic interventions. We show that Htt aggregation and inclusion formation in the cytosol and nucleus occur via different mechanisms and lead to the formation of inclusions with distinct biochemical and ultrastructural properties. These observations suggest that the two types of inclusions may exert their toxicity via different mechanisms and may require different strategies to interfere with their formation, maturation, and toxicity. Furthermore, our work suggests that identifying modifiers of Htt inclusion growth and aberrant secondary interactions with other proteins and organelles represent an alternative and strategy for interfering with Htt-induced toxicity and slowing disease progression, especially after disease onset. Therefore, we believe that targeting inclusion growth and maturation represents a viable therapeutic strategy.

## Methods

**DNA constructs and purification.** pCMV mammalian expression vector encoding for Httex1 16Q, Httex1 16Q-GFP, Httex1 39Q, Httex1 39Q-GFP, Httex1 72Q, and Httex1 72Q-GFP were kindly provided by Andrea Caricasole (IRBM). ΔN17-Httex1 39Q, ΔN17-Httex1 39Q-GFP, ΔN17-Httex1 72Q, and ΔN17-Httex1 72Q-GFP were purchased from GeneArt (Germany). SIN-PGK expression vector encoding for Httex1 16Q, Httex1 16Q-GFP, Httex1 72Q, and Httex1 72Q-GFP, ΔN17-Httex1 16Q, ΔN17-Httex1 16Q-GFP, ΔN17-Httex1 72Q, and ΔN17-Httex1 72Q-GFP were purchased from GeneArt (Germany). Plasmids were transformed into Chemo-competent *E. coli* stable 3 cells (Stbl3) from Life Technologies (Switzerland), and Maxiprep plasmid purification (Life Technologies, Switzerland) was performed following the manufacturer's instructions.

**Lentivirus production.** On day 0, HEK 293 cells ([HEK-293] ATCC® CRL-1573'") were plated on seven 15 cm dishes. 24 h after plating, cells were transfected with the

plasmid of interest (160 µg SIN-PGK vector plasmid), the plasmid encoding the VSV G envelope protein (55.3 µg pMDG2G) and the plasmid encoding the HIV gag, Pol, TAT, and Rev proteins (102.2 µg pCMVR8.74) using a calcium phosphate transfection kit (Takara, Japan). The media was changed with fresh media 24 h after transfection and then collected and replaced after 8 h and 24 h. The collected media were spun down at 500 g at 4 °C for 5 min to pellet cellular debris. The supernatant was cleared with a 0.22 µm filter and then centrifuged at 50,000 g at 16 °C for 2 h. The supernatant was discarded and the pellet air-dried by inversion. Finally, the pellet was well resuspended in PBS and stored at −80 °C. This method was performed following the protocol from Barde et al.[151].

**Mammalian cell culture and plasmid transfection.** HEK 293 cells ([HEK-293/17] ATCC® CRL-11268'") were maintained in Dulbecco's modified Eagle's medium DMEM (Life Technologies, Switzerland) containing 10% FBS (Life Technologies, Switzerland), 10 µg/ml streptomycin, and penicillin (Life Technologies, Switzerland) in a humidified incubator, and 5% $CO_2$ at 37 °C. Cells were plated at a density of 100,000 per dish in glass-bottom µ-Dishes (IBIDI) or 50,000 cells/well in 24 well plates with a Thermanox Plastic Coverslip (round) 13 mm in diameter (Life Technologies, Switzerland) in order to obtain cells at a 70–90% confluence the day after for the transfection procedure using a standard calcium phosphate procedure[152]. Briefly, 2 µg of DNA was diluted in 30 µl $H_2O$ and 30 µl of 0.5 M CaCl2 before the dropwise addition of 60 µl of 2xHBS, pH 7.2 (50 mM HEPES, pH 7.05; 10 mM KCL; 12 mM dextrose; 280 mM NaCl; 1.5 mM Na2PO4, pH 7.2 dissolved in H2O) under mild vortexing condition.

**Mouse primary cortical cell culture and lentiviral transduction.** First, cortex were isolated from P0 pups of WT mice (C57BL/6JRccHsd, Harlan) in Hank's Balanced Salt Solution (HBSS)[122,153]. Brain tissues were digested by papain (20 U/mL, Sigma-Aldrich, Switzerland) for 30 min at 37 °C and the digestion was stopped using a trypsin inhibitor (Sigma-Aldrich, Switzerland). Next, the tissues were dissociated by mechanical trituration, and the isolated cells were resuspended in adhesion media (MEM, 10% Horse Serum, 30% glucose, L-glutamine, and penicillin/streptomycin) (Life Technologies, Switzerland). Finally, neurons were plated in 6-well plates for biochemical analysis at a density of 500,000 cells/ml, or in 24-well plates containing glass coverslips at a density of 125,000 cells/ml, all previously coated with poly-l-lysine 0.1% w/v in water (Sigma-Aldrich). After 3 h, the adhesion media was removed and replaced by neurobasal medium (Life Technologies, Switzerland) supplemented with L-glutamine, penicillin/streptomycin (100 U/mL, Life Technologies, Switzerland), and B27 supplement (Life Technologies, Switzerland).

Cortical primary neurons were transduced with Httex1 lentiviruses particles at a multiplicity of infection (MOI) 2 after 5 or 7 days in vitro (DIV)[154]. All procedures were approved by the Swiss Federal Veterinary Office (authorization number VD 3392). Mice housing conditions: 22 °C, 12–14 h light per day, and relative humidity from 50 to 60%.

**Immunocytochemistry (ICC).** At the indicated time-point, HEK 293 cells or cortical primary neurons were washed twice with PBS pH 7.4 (1X) (Life Technologies, Switzerland) and fixed in 3.7% formaldehyde (Sigma-Aldrich, Switzerland) in PBS (PFA) for 15 min at room temperature (RT). After a blocking step with 3% BSA (Sigma-Aldrich, Switzerland) diluted in 0.1% Triton X-100 (Applichem, Germany) in PBS (PBST) for 30 min at RT, cells were incubated with the primary antibodies (Supplementary Fig. 1) anti-Htt raised against the Nt17 domain (2B7, CHDI [Cure Huntington's Disease Initiative]; Ab109115, Abcam) or the PolyQ (MW1, CHDI) or Proline-Rich Domain (PRD) (MAB5492, Millipore; 4C9, CHDI; N18, Santa-Cruz and EGT 414), or against Htt (S830) at a dilution of 1/500 in PBST for 2 h at RT. Cells were then rinsed five times in PBST and incubated for 1 h at RT with the secondary donkey anti-mouse Alexa647, donkey anti-rabbit Alexa647, or donkey anti-goat 568 antibodies (Life Technologies, Switzerland) used at a dilution of 1/800 in PBST and DAPI (Sigma-Aldrich, Switzerland) at 2 µg/ml, all diluted in PBST. In addition, HEK cells were counterstained with Phalloidin Atto594 (Sigma-Aldrich, Switzerland), which has a high affinity to filamentous F-actin, while cortical neurons were detected with a MAP2 antibody.

Cells were then washed five times in PBST, and a last one in double-distilled H2O, before being mounted in polyvinyl alcohol (PVA) mounting medium with DABCO (Sigma-Aldrich, Switzerland). Cells were examined with a confocal laser-scanning microscope (LSM 700, Zeiss, Germany) with a 40 × 1.3 oil objective (Plan-Apochromat) and analyzed using Zen software (Zeiss, Germany) or using a confocal laser-scanning microscope (Inverted Leica SP8, Germany) with a 63×/1.4 oil objective (HC PL APO) and analyzed using LASX Leica software.

**Image-based quantification Httex1 expression in primary neurons.** A minimum of five areas per condition was imaged for each independent experiment, as described above. Each experiment was performed three times. The cell counter feature was used in the LASX Leica software to quantify the morphological expression of Httex1 in neurons at different time points. The classification was done according to Fig. 5c with the detection of Httex1 classified as diffuse; small nuclear puncta; large nuclear inclusion, or cytoplasmic inclusion. In each

condition, approximately 150 neurons were quantified for each condition, and a minimum of three independent experiments was performed.

**Immunofluorescence staining of ER exit sites**. A total of 100,000 HeLa cells were seeded into a 6-well plate on glass coverslips. After 24 h, cells were transfected with the different variants of GFP-tagged or tag-free Httex1 using Fugene 6 according to the manufacturer's instructions. An empty vector was used as a negative control. 48 h after transfection, cells were fixed in 4% PFA for 20 min and stained using an anti-Htt antibody (Millipore, mouse monoclonal (MAB5492)). Briefly, after cells were washed with PBS containing 20 mM glycine, slides were incubated in a blocking buffer composed of 3% BSA (Bovine Serum Albumin) in 0.1% Triton X-100 and PBS for 30 min at RT. Subsequently, cells were incubated with the primary antibodies against Htt and Sec13 (R&D Systems) to label ER exit sites, followed by washing and incubation with Alexa-Fluor tagged secondary antibodies. Slides were washed with PBS and embedded in a polyvinyl alcohol mounting medium with DABCO (Sigma-Aldrich). Cells were imaged with 63× objective using a Zeiss LSM 700 confocal microscope.

In the case of Httex1-GFP, cells were washed in PBS containing 20 mM glycine followed by permeabilization in PBS containing 0.2% Triton X-100. Subsequently, cells were incubated with primary antibody to stain ER exit sites diluted in 3% BSA in PBS. After being washed with PBS, cells were incubated with the appropriate Alexa-Fluor tagged secondary antibodies in 3% BSA-PBS. Slides were washed with PBS and embedded in a polyvinyl alcohol mounting medium with DABCO (Sigma-Aldrich). Cells were imaged with 63× objective using a Zeiss LSM 700 confocal microscope.

Quantification of ER exit sites' number and size was performed using the analyze particles tool in Image J after thresholding for pixel size and intensity.

**Correlative light and electron microscopy (CLEM)**. HEK 293 cells were grown at 600,000 cells/ml on 35 mm dishes with alpha-numeric searching grids etched on the bottom glass (MatTek Corporation, Ashland, MA, USA). 48 h after transfection with either Empty vector, Httex1 72Q, or Httex1 72Q-GFP, cells were fixed for 2 h with 1% glutaraldehyde (Electron Microscopy Sciences, USA) and 2.0% PFA in 0.1 M phosphate buffer (PB) at pH 7.4. Similarly, primary cortical neurons grown on gridded glass dishes (MatTek Corporation, Ashland, MA, USA) or manually annotated 13 mm plastic coverslips [Thermanox 174950] (Thermo Fisher Scientific, Waltham, USA) were washed 7 days post-transduction and fixed similarly to HEK cells described just above.

After washing with PBS, ICC was performed as described above. Intra-cellular inclusions were stained with an Htt antibody (Millipore MAB5492, aa 1-82), and the cells of interest were imaged with a fluorescence confocal microscope (LSM700, Carl Zeiss Microscopy) with a 40x objective. The precise position of the selected cells was recorded using the alpha-numeric grid etched on the dish bottom. The cells were then fixed further with 2.5% glutaraldehyde and 2.0% PFA in 0.1 M PB at pH 7.4 for another 2 h. After five washes of 5 min with 0.1 M cacodylate buffer at pH 7.4, cells were post-fixed with 1% osmium tetroxide in the same buffer for 1 h and then washed with double-distilled water before being contrasted with 1% uranyl acetate water for 1 h. The cells were then dehydrated in increasing concentrations of alcohol (2 × 50%, 1 × 70%, 1 × 90%, 1 × 95%, and 2 × 100%) for 3 min each wash. Dehydrated cells were infiltrated with Durcupan resin (Electron Microscopy Sciences, Hatfield, PA, USA) diluted with absolute ethanol at 1: 2 for 30 min, at 1:1 for 30 min, at 2:1 for 30 min, and twice with pure Durcupan for 30 min each. After 2 h of incubation in fresh Durcupan resin, the dishes were transferred into a 65 °C oven so that the resin could polymerize overnight. Once the resin had hardened, the glass CS on the bottom of the dish was removed by repeated immersion in hot water (60 °C), followed by liquid nitrogen. The cell of interest was then located using the previously recorded alpha-numeric coordinates, and a razor blade was used to cut this region away from the rest of the resin. This piece was then glued to a resin block with acrylic glue and trimmed with a glass knife using an ultramicrotome (Leica Ultracut UCT, Leica Microsystems). Next, ultrathin sections (50–60 nm) were cut serially from the face with a diamond knife (Diatome, Biel, Switzerland) and collected on 2 mm single-slot copper grids coated with formvar plastic support film. Sections were contrasted with uranyl acetate and lead citrate and imaged with a transmission electron microscope (Tecnai Spirit EM, FEI, The Netherlands) operating at 80 kV acceleration voltage and equipped with a digital camera (FEI Eagle, FEI).

**Sample processing for electron microscopy imaging without cell permeabilization**. 48 h after transfection, HEK 293 cells were fixed in PFA 2% and glutaraldehyde 2% in phosphate buffer 0.1 M (pH 7.4) for 1 h and 30 min. To preserve the internal membranes of the cells, no ICC was performed. Cells were then washed 3 times for 5 min in cacodylate buffer (0.1 M, pH 7.4). Next, they were post-fixed with 1% osmium tetroxide plus 1.5% potassium ferrocyanide in cacodylate buffer (0.1 M, pH 7.4) at RT for 40 min, followed by post-fixation with 1% osmium tetroxide in cacodylate buffer (0.1 M, pH 7.4) at RT for 40 min. Samples were washed twice for 5 min in distilled water, then further stained in 1% uranyl acetate in water for 40 min and washed once in double-distilled water for 5 min. The samples were dehydrated in increasing concentrations of ethanol for 3 min each wash (2 × 50%, 1 × 70%, 1 × 90%, 1 × 95%, 2 × 100%) and then embedded in epoxy

resin (Epon had the formula: Embed 812: 20 g, DDSA: 6.1 g, NMA: 13.8 g, DMP 30: 0.6 g) (Electron Microscopy Sciences, USA) through the continuous rotation of vials. The embedding process starts with a 1:1 ethanol:epon mix for 30 min, followed by 100% EPON for 1 h. EPON was then replaced with fresh EPON for 2 h. Finally, samples were embedded between coated glass slides and placed in an oven at 65 °C overnight. 50 nm thick serial sections were cut with an ultramicrotome (UC7, Leica Microsystems, Germany) and collected on formvar support films on single-slot copper grids (Electron Microscopy Sciences, USA) for transmission electron microscopy imaging (TEM). TEM images were taken at 80 kV filament tension with a Tecnai Spirit EM microscope, using an Eagle 4 k × 4 k camera. At least 8 cells were imaged per condition at 2900× and 4800× magnification.

Images were aligned using Photoshop CC 2019 software (Adobe, USA) and different organelles (Nucleus, Mitochondria, Endoplasmic Reticulum, Httex1 inclusions) were first segmented manually using the arealist function in the trackEM2 plugin in the FIJI software. We then moved to a custom-developed machine learning-based pipeline, tailored specifically to 3D microscopy data (www.ariadne-service.ch) after validation using the manual segmentation for reference. At the end of the segmentation process, the different segmented areas were exported as serial image masks, then visualized as objects in the 3D viewer plugin in FIJI and exported as wavefront in the Blender® 3D modeling software (Blender Foundation). Using Blender®, the 3D axes were first corrected according to the model orientation and the Z scale was adjusted. Httex1 inclusion, mitochondria, and the nucleus were smoothed and the Httex1 inclusion was adjusted to visualize intra-aggregates membranous structures within the inclusion. Additional measurements were performed from electron micrographs in cells containing Httex1 inclusions using FIJI. The binary image (inset) Fig. 1d was generated from the electron micrograph using a median filtering and Otsu intensity threshold, allowing for a better distinction between the core and the shell ultrastructure of the tag-free Httex1 72Q inclusion. The mitochondrial profile length corresponds to the maximal length of each mitochondria in one EM plane. The distance from the inclusion was not taken into account, as the measurements were performed in one plane. Instead, the average length of all detected mitochondria was taken into account and showed significant differences in the Httex1 72Q condition compared to the EV control.

**High pressure freezing and embedding for TEM**. 48 h after transfection, HEK 293 cells were cultured on sapphire discs (6 mm diameter) and frozen rapidly under high pressure (HPM100, Leica Microsystems). The discs were placed into cryotubes containing acetone with 1% osmium tetroxide, 0.5% uranyl acetate, and 5% water at −90 °C. They were then left at this temperature for 24 h before being warmed at 0 °C over the next 72 h where they were then washed with pure acetone, and then infiltrated with increasing concentrations of Epon resin. Once in 100% resin the samples were left at room temperature for 24 h, and then the resin hardened in an oven at 65 °C for 48 h. Serial sections were collected on single-slot copper grids with a formvar support film, and then stained with lead citrate and uranyl acetate. These were imaged inside a transmission electron microscope at 80 kV (Tecnai Spirit, FEI Company), using a CCD camera (Eagle, FEI Company).

**Electron tomography (ET)**. Six tilt series were collected with the Tomo 4.0 software (Thermo Fisher Scientific) on a Tecnai F20 TEM operated at 200 kV (Thermo Fisher Scientific), on a Falcon III DD camera (Thermo Fisher Scientific) in linear mode at 29'000× magnification. The tilt series were recorded from −58° to 58° using a continuous tilt scheme, with an increment of 2°. Tilt series alignments and tomogram reconstruction were performed with Inspect3D v4.1.2 (Thermo Fisher Scientific) using 22 iterations of SIRT.

The resulting tomogram was subjected for filling the missing wedge with a new deep learning-based tool in EMAN2 build after 03/20/2020[155]. Missing wedge corrected tomograms were subjected for template-free, semi-automated convolutional neural network (CNN) based semi-automated tomogram annotation in EMAN2. Six tomograms were imported into the EMAN2 project manager and shrunk by a factor of two. The tomograms were then inspected slice by slice, and annotated by manual selection of a few 64 × 64 pixel tiles containing elongated features that resembled fibrils, in order to train the CNN. In addition, quite a few regions in the tomogram which do not contain elongated features, were manually annotated as negative examples. Selected positive examples were manually segmented with pixel accuracy. Both positive and negative examples were provided for the training of the neural network. The trained CNN was then applied to the original tomogram for complete annotation. Images and movies were generated with 3dmod from IMOD package[156,157].

**Preparation of protein samples for biochemical analyses**. Samples were generated, in duplicate, of the HRR experiment for analysis of mitochondrial markers by WB and FT. Transfected cells were lysed in 75 μl of RIPA buffer (150 Mm NaCl, 1 μ NP40, 0.5% Déoxycholate, 0.1% Sodium dodecyl sulfate (SDS), 50 Mm Tris pH 8). Cell lysates were incubated at 4 °C for 20 min and then cleared by centrifugation at 4 °C for 20 min at 16,000 g. Supernatants were collected as soluble protein fractions and stored at −20 °C after LB5x addition and 5 min of boiling. BCA was performed on the RIPA soluble fraction. Pellets were washed with 500 ul of PBS, then centrifuged again for 5 min at 16,000 g. Supernatants were discarded and the

pellet resuspended in 30 μl of PBS supplemented with SDS 2% and sonicated with a fine probe [3 times, 3 s at the amplitude of 60% (Sonic Vibra Cell, Blanc Labo, Switzerland)]. Cellulose acetate membrane was first equilibrated with 2% SDS (in PBS) for 5 min and the main filter fold arranged on top of 2 Watman papers inside the Bio-Dot Apparatus (#1706545, USA). Samples were loaded and filtered by vacuum. The membrane was washed 3 times by 0.5% SDS in PBS and applying the vacuum. The acetate membrane was then removed and washed once in PBS-Tween 1%. WB and Filter-Trap membranes were blocked overnight with Odyssey blocking buffer (LiCor) and then incubated at RT for 2 h with different primary antibodies (anti-huntingtin MAB5492 and anti-VDAC1) diluted in the same blocking buffer (1/5000). Membranes were washed in PBS-Tween 1% and then incubated with secondary antibody diluted in blocking buffer (1/5000) for 1 h at RT before a final wash in PBS-Tween 1%. The protein detection was performed by fluorescence using Odyssey CLx from LiCor. The signal intensity was quantified using Image Studio 3.1 from LiCor.

**Preparation of samples for mass spectrometry**. Samples were generated in triplicate for quantitative mass spectrometry analysis. At the indicated time-point, HEK cells or primary neurons were lysed in PBS supplemented by 0.5% NP40, 0.5% Triton x100, 1% protease cocktail inhibitor (Sigma PB340), and 1% Phe-nylmethanesulfonyl Fluoride (PMSF, Applichem). Cell lysates were incubated at 4 °C for 20 min and then cleared by centrifugation at 4 °C for 20 min at 20,000 g. Supernatants were collected as non-ionic soluble protein fractions. Pellets were washed and resuspended in PBS supplemented by 2% N-Lauroylsarcosine sodium salt (Sarkosyl, Sigma) with protease inhibitors. The pellets were briefly sonicated with a fine probe [3 times, 3 sec at the amplitude of 60% (Sonic Vibra Cell, Blanc Labo, Switzerland)], incubated 5 min on ice, then centrifuged at 100,000 g for 30 min at 4 °C. Supernatants were collected as Sarkosyl soluble fractions. Pellets were washed with the previous buffer and resuspended in PBS supplemented by 2% Sarkosyl and 8 M Urea and briefly sonicated as done previously. Laemmli buffer 4x was added to samples before being boiled at 95 °C for five minutes. Samples were then separated on a 16% SDS-PAGE gel before being analyzed by Coomassie staining and WB.

For WB analyses, nitrocellulose membranes were blocked overnight with Odyssey blocking buffer (LiCor, Switzerland) and then incubated at RT for 2 h with Htt primary antibodies (MAB5492, Millipore, Switzerland) diluted in PBS (1/5000). Membranes were washed in PBS-Tween 1% and then incubated with fluorescently labeled secondary antibody diluted in PBS (1/5000) for 1 h at RT before a final wash in PBS-Tween 1%. The protein detection was performed by fluorescence using Odyssey CLx from LiCor. The signal intensity was quantified using Image Studio 3.1 from LiCor.

For proteomic identification, the samples separated by SDS-PAGE were then stained with Coomassie blue (25% Isopropanol [Fisher scientific, United-States], 10% acetic acid [Fisher scientific, United-States], and 0.05% Coomassie brilliant R250 [Applichem, Germany]). Each gel lane was entirely sliced and proteins were In-gel digested[158]. Briefly, samples were reduced in 10 mM dithioerythritol (DTE) (Merck, 1.24511.0005) alkylated in 55 mM iodoacetamide (IAA) (Sigma I1149), and gel pieces were dried. Digestion was performed overnight at 37 °C using mass spectrometry grade trypsin (Thermofisher Scientific 90057) at a concentration of 12.5 ng/μl in 50 mM Ammonium Bicarbonate (Sigma A6141) and 10 mM CaCl2 (Merck 1.02382.0500). The resulting peptides were extracted in 70% ethanol (Merck 1.117272500), 5% formic acid (Merck 1.00264.1000) twice for 20 min with permanent shaking. Samples were further dried by vacuum centrifugation and stored at −20 °C. Peptides were desalted on stagetips[159] and dried under a vacuum concentrator. For Liquid chromatography with tandem mass spectrometry (LC-MS/MS) analysis, samples were resuspended in 2% acetonitrile (Biosolve 0001207802BS), 0.1% Formic Acid and were separated by reversed-phase chromatography on a Dionex Ultimate 3000 RSLC nano UPLC system connected in-line with an Orbitrap Lumos (Thermo Fisher Scientific, Waltham, USA). A capillary precolumn (Acclaim Pepmap C18, 3 μm-100Å, 2 cm × 75 μm ID) was used for sample trapping and cleaning. A capillary column (75 μm ID; in-house packed using ReproSil-Pur C18-AQ 1.9 μm silica beads; Dr. Maisch; length 50 cm) was then used for analytical separations at 250 nl/min over 90 min biphasic gradient. Acquisitions on the Orbitrap Exploris 480 were performed through Top Speed Data-Dependent acquisition mode using a 2 s cycle time. First mass spectrometry (MS) scans were acquired at a resolution of 60'000 (at 200 m/z) and the most intense parent ions were selected and fragmented by High energy Collision Dissociation (HCD) with a Normalized Collision Energy (NCE) of 30% using an isolation window of 1.4 m/z. Fragmented ion scans were acquired with a resolution of 15'000 (at 200 m/z) and selected ions were then excluded for the following 20 s. Acquisitions on the Qexactive HF were performed through the Top N acquisition (15). First MS scans were acquired at a resolution of 60'000 (at 200 m/z) and the most intense parent ions were selected and fragmented by High energy Collision Dissociation (HCD) with a Normalized Collision Energy (NCE) of 27% using an isolation window of 1.4 m/z. Fragmented ion scans were acquired with a resolution of 15'000 (at 200 m/z) and selected ions were then excluded for the following 20 s.

Protein identification and quantification were performed with the search engine MaxQuant 1.6.2.10[160]. The Human (*Homo Sapiens*, release 2019_06) or mouse (*Mus musculus*, release 2020_10) Uniprot database and Httex1 Sequence was used.

Carbamidomethylation was set as a fixed modification, whereas oxidation (M), phosphorylation (S, T, Y), acetylation (Protein N-term), and glutamine to pyroglutamate were considered as variable modifications. A maximum of two missed cleavages was allowed. "Match between runs" was enabled. A minimum of 2 peptides was allowed for protein identification, and the false discovery rate (FDR) cut-off was set at 0.01 for both peptides and proteins. Label-free quantification and normalization were performed by Maxquant using the MaxLFQ algorithm, with the standard settings[161]. In Perseus[162], reverse proteins, contaminants, and proteins identified only by sites were filtered out. Data from the Urea fraction were analyzed separately following the same workflow. Biological replicates were grouped together, and protein groups containing a minimum of two LFQ values in at least one group were conserved. Missing values were imputed with random numbers using a gaussian distribution (width = 0.7, down-shift = 1.9 for Urea fraction). Differentially expressed proteins were highlighted by a two-sample t-test, followed by a permutation-based correction (False Discovery Rate). Significant hits were determined by a volcano plot-based strategy, combining t-test P-values with ratio information[163]. Significance curves in the volcano plot corresponded to an S0 value of 0.5 and an FDR cut-off of 0.05. Further graphical displays were generated using homemade programs written in R (version 3.6.1)[164]. In primary neurons, a bioinformatics pipeline was implemented using the Differential Enrichment analysis of Proteomic data (DEP)[105]. DEP has been shown to have greater sensitivity for detecting true differences between conditions compared to pairwise between-condition t-tests, as the overall variability of all samples is used to inform the variance-stabilising normalization and differential statistical test approach. This pipeline was implemented in R (4.0.3 (2020-10-10)) and the code is available here: https://github.com/jannahastings/httex-proteomics-202107 (https://doi.org/10.5281/zenodo.5469407). Venn diagrams were generated from (http://bioinformatics.psb.ugent.be/webtools/Venn/). IPA analysis was performed with the software (https://digitalinsights.qiagen.com/products-overview/discovery-insights-portfolio/analysis-and-visualization/qiagen-ipa/). DAVID 6.8 was used for GO term analysis (https://david.ncifcrf.gov/). Microsoft Excel was used for further analysis and in the comparative analysis, known HTT interactor proteins were selected using the HDinHD dataset (https://www.hdinhd.org/) and restricted to the Human and mouse datasets among cell- or animal-based studies exclusively.

**Respirometry and amplex red fluorometry**. HEK 293 cells were transfected 24 h after plating with Htt 16Q, Htt 72Q, Htt 16Q-GFP, or Htt 72Q-GFP. 48 h after transfection, cells were gently detached using 0.05% trypsin, resuspended in DMEM, counted, and immediately used for high-resolution respirometry.

One million cells were transferred to MiR05 (0.5 mM EGTA, 3 mM MgCl2, 60 mM potassium lactobionate, 20 mM taurine, 10 mM KH2PO4, 20 mM HEPES, 110 mM sucrose, and 0.1% (w/v) BSA, pH = 7.1) in a calibrated, 2 ml high-resolution respirometry oxygraph chamber (Oroboros Instruments, Austria) kept stably at 37 °C. Mitochondrial ROS production ($O_2^-$ and $H_2O_2$) was measured using amplex red fluorometry and O2K Fluo-LED2 modules (Oroboros Instruments, Austria)[165]. Briefly, for amplex red fluorometry, cells were added to oxygraph chambers after the addition of 10 μM amplex red, 1 U/ml horseradish peroxidase, and 5 U/ml superoxide dismutase to MiR05 and calibration with known $H_2O_2$ concentrations. Fluorescence was measured during the subsequently applied high-resolution respirometry protocol.

Routine respiration (and mitochondrial ROS production) was measured from intact cells, after which plasma membranes were permeabilized by the application of an optimized (integrity of mitochondrial outer membranes verified by the cytochrome c test) concentration of digitonin (5 μg/mL).

Oxygen flux at different respirational states on permeabilized cells was then determined using the substrate-uncoupler-inhibitor-titration (SUIT) protocol[166,167]. Briefly, NADH-pathway (N) respiration in the LEAK and oxidative phosphorylation (OXPHOS) state was analyzed in the presence of malate (2 mM), pyruvate (10 mM), and glutamate (20 mM) before and after the addition of ADP (5 mM), respectively ($N_L$, $N_P$). The addition of succinate (10 mM) allowed for the assessment of NADH- and succinate-linked respiration in OXPHOS ($NS_P$) and in the uncoupled state ($NS_E$) after the incremental ($\Delta0.5$ μM) addition of carbonyl cyanide m-chlorophenyl hydrazine (CCCP). The inhibition of Complex I by rotenone (0.5 μM) yielded succinate-linked respiration in the uncoupled state ($S_E$). Tissue-mass specific oxygen fluxes were corrected for residual oxygen consumption, Rox, measured after additional inhibition of the mitochondrial electron transport system, ETS, Complex III with antimycin A. For further normalization, fluxes of all respiratory states were divided by ET-capacity to obtain flux control ratios, FCR. Terminology was applied according to http://www.mitoeagle.org/index.php/MitoEAGLE_preprint_2018-02-08.

Mitochondrial ROS values were corrected for background fluorescence and respirational states before the addition of the uncoupler used for analysis.

**Toxicity assays in primary neurons**. The culture supernatant of primary cortical neurons was collected at the indicated time-point, and the level of lactic acid dehydrogenase (LDH) was measured using the CytoTox 96® Non-Radioactive Cytotoxicity Assay (Promega, Switzerland)[122]. Briefly, 50 μl of the CytoTox 96® reagent was mixed with 50 μl of the collected media and incubated for 30 min at RT. After incubation, the reaction was terminated with 50 μl of the stop solution, and the absorbance was measured at 490 nm using a Tecan infinite M200 Pro plate

reader (Tecan, Maennedorf, Switzerland), which proportionally indicates the number of cells with a permeabilized membrane leading to LDH release in the culture media. In addition, DNA fragmentation-associated cell death was measured by TUNEL assay[168]. In brief, cortical neurons were washed and fixed in 4% PFA for 15 min at RT at the indicated time-points. Neurons were permeabilized in 0.1% Triton X-100 in 0.1% citrate buffer, pH 6.0, before being incubated with the terminal deoxynucleotide transferase and TMR red dUTP (In Situ Cell Death Detection kit; Roche, Switzerland) for 1 h at 37 °C. ICC was next performed as described above with the neuronal marker NeuN to select the neurons. A minimum of one hundred neurons was counted for each condition and for each independent experiment, done in triplicate.

**Statistics and reproducibility**. The micrographs Figs. 1b, e, g, 2a, 4d, 5a, 6a, b, 6c, 7b, e, 8a, e, S2, S3, S5, S6, S7a, S8, S9, S10, S14a, S14b, S14c, S14d, S15, S17, S19, S20a, S20b, S21a, S21b, S21c, S22, S28a and S29a are representative images of three independents experiments. The micrographs 1c, d from CLEM (Supplementary Fig. 16a, c) represent one inclusion and are comparable to the two other independent repeats of the micrographs 1e, 2b, 4a, and S3a. The micrographs 7c, d from CLEM (Supplementary Fig. 16a, d) represent one inclusion and are comparable to the two other independent repeats of the micrographs 7f and S17a. 3D reconstructions 1f, 1h, S7b, S20c, and S20d were generated once from representative electron micrographs S3a, S3b, S8a, S17a, and S17b respectively. The micrographs figures S4 are representative of several inclusions from one experiment. Full and unprocessed WB and Coomassie scans are available in the source data file and Figures S31 and S32.

**Statistical analysis**. All experiments were independently repeated at least 3 times. The statistical analyses were performed using Student's t-test, one-way ANOVA test followed by a Tukey-Kramer or HSD post-hoc tests, two-way ANOVA, and repeated measures ANOVA using KaleidaGraph (RRID:SCR_014980) or Graph-Pad Prism 9.1.1. The data were regarded as statistically significant at $p < 0.05$. Gene Ontology (GO) enrichment analyses were determined by DAVID analysis (Significant for $p$ values < 0.05 adjusted for multiple testing based on Benjamini and Hochberg)[169].

**Reporting summary**. Further information on research design is available in the Nature Research Reporting Summary linked to this article.

## Data availability

All proteomic databases generated in this study are available via ProteomeXchange (PRIDE) with the identifier PXD021742 for HEK cells and PXD028323 for primary neurons. The Uniprot databases for Human (Homo Sapiens, release 2019_06) or mouse (Mus musculus, release 2020_10) are available on (https://www.uniprot.org/). Proteomic datasets identifier and links are available in the source data file. All the full scan western blots and Coomassie are displayed in the source data file as well as Figs. S31 and S32. All the other data are provided in the article and Supplementary information are available from the corresponding author on reasonable request. Source data are provided with this paper.

## Code availability

The R code for proteomic analysis and visualization generated in this study has been deposited in Zenodo with the identifier 5469407 and is available at [https://doi.org/10.5281/zenodo.5469407]. Code identifier and link are also available in the source data file.

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

## Acknowledgements

This work was supported by funding from the Ecole Polytechnique Fédérale de Lausanne (EPFL) and the CHDI Foundation (H.A.L, N.R, A.L.M, N.M, J.B and A.P). We thank Dr. Arne Seitz and the staff at the Bio-imaging Core Facility (BioP, EPFL) for their technical support and valuable discussions. We are grateful to the Proteomics Core Facility (EPFL): Dr. Diego Chiappe, Dr. Romain Hamelin, and Dr. Florence Armand for their guidance, helpful discussions, and technical support with the proteomic studies. We also thank Stéphanie Rosset and Anaëlle Dubois at the Bio-EM Core facility (EPFL) as well as Dr. Davide Demurtas from the CIME core facility (EPFL) for their technical support. We are grateful to the research associates in the Lashuel Lab: Driss Boudeffa, Jonathan Ricci, and Yllza Jasiqi for their continuous technical assistance. We thank Lorene Aeschbach for the preparation of primary neuronal cultures. We are grateful to Dr. Pedro Magalhães (LMNN, EPFL) for his help and guidance on the proteomic analysis as well as comments on the manuscript. We are grateful to Rafaël Galvan for double-checking the references. We thank Dr. Andrea Caricasole (IRBM) for kindly providing the pCMV mammalian expression vector encoding for the basic Httex1 constructs. We would also like to thank Dr. Magda Zachara (UPDEPLA, EPFL) for the Bodipy staining and useful discussions. The graphical abstract, Fig. 5c, S11a, S13c, and Fig. 10 were created with BioRender.com (Agreement number UZ22NP1I1I).

## Author contributions

H.A.L conceived and supervised the study. H.A.L, N.R, and A.L.M.M designed all the experiments and wrote the paper. N.R performed and analyzed the confocal imaging, the electron microscopy and the 3D imaging reconstruction, quantitative proteomic experiments, and the biochemistry experiments. A.L.M.M performed and analyzed the CLEM experiment and analyzed the proteomic data. N.M designed and analyzed the proteomic data. J.B designed, performed, and analyzed the high-resolution respirometry experiments. A.P helped to perform and analyze the confocal imaging experiments. M.C prepared the samples for CLEM and acquired the EM and ET images. G.K supervised the EM-related experiments and 3D imaging reconstruction and contributed to the interpretation of the data. S.N generated, annotated, and analyzed the tomograms. J.H. analyzed the proteomic data. V.R and H.F designed, performed, and analyzed the experiments related to ER. All authors reviewed and contributed to the writing.

## Competing interests

H.A.L has received funding from the industry to support research on neurodegenerative diseases, including from Merck Serono, UCB, and Abbvie. These companies had no specific role in the conceptualization, preparation, and decision to publish this work. H.A.L is also the co-founder and Chief Scientific Officer of ND BioSciences SA, a company that develops diagnostics and treatments for neurodegenerative diseases based on platforms that reproduce the complexity and diversity of proteins implicated in neurodegenerative diseases and their pathologies. All other authors declare no competing interests.
