## [Peer Review File · Nature Communications]

Nuclear and cytoplasmic huntingtin inclusions exhibit distinct biochemical composition, interactome and ultrastructural propertiesREVIEWER COMMENTS

Reviewer #1 (Remarks to the Author):

In the current manuscript, authors investigated the ultrastructural properties and protein composition of Htt inclusions in cells overexpressing mutant exon 1 of the Htt protein. Authors demonstrated that Htt inclusion formation and maturation are complex processes that, although initially driven by polyQ-dependent Htt aggregation, also involve polyQ and PRD domain-dependent sequestration of lipids and cytoplasmic and cytoskeletal proteins related to Huntington's disease (HD) dysregulated pathways, recruitment and accumulation of remodeled or dysfunctional membranous organelles and impairment of the protein quality control and degradation machineries. Authors provided a more in-depth analysis of the impact of fluorescent fusion proteins, on the biochemical and ultrastructural properties of Htt inclusions and their impact on cellular components and organelles. These observations are important and worth reporting. Most methods are highly innovative and acceptable. Authors put huge effort into this important work and highly appreciable. Concerns –

1. Most of the experiments were conducted using HEK 293 cells, why not in other cells such as STHDhQ111/Q111.
2. Although findings are highly innovative and important, how the present work will advance the HD field in terms of therapeutics. Hope this is not inclusions paper (there are so many published already)
3. Literature survey mostly well done, but important recent EM studies on mitochondria – Yin et al Hum Mol Genet 2016 and Manczak and Reddy, Hum Mol Genet 2015.
4. Hyper activation of mitochondrial respiration is observed in cells with 72polyQ, is it due to compensatory response due to mitochondrial dysfunction in HD neurons? – please comment
5. ER-mitochondrial association is increased in mutant Htt cells – if yes, what are the thoughts on this concept.
6. Autophagy/mitophagy seems to be defective in mutant htt cells – any thoughts on autophagy/mitophagy inducers to enhance Autophagy/mitophagy HD.

Reviewer #3 (Remarks to the Author):

Riguet et al.

“Disentangling the sequence, cellular and ultrastructural determinants of Huntingtin nuclear and cytoplasmic inclusion formation.”

Summary:

This study investigates the nature of aggregates formed by a small piece of mutant HTT in cells both at the ultrastructure level and biochemical composition. Strengths include the rigorous use of correlative light electron microscopy (CLEM) and serial reconstruction to visualize Htt aggregates in a native space. A major weakness is the lack of neuronal context in all studies and the use of only Htt exon1, although an important comparison to Htt exon1-GFP which has been widely used in the field offers eye-opening comparisons. Mass spectrometry identifies both nuclear and cytoplasmic components within aggregates. Unfortunately, the mass spectrometry study offers limited useful information since it too was performed in HEK cells. The authors even conclude in the Discussion that the cell environment contributes to process of aggregate formation and protein recruitment (lines 742-747). If this is true then surely the work needs to be performed in neurons. The work does extend findings of previous laboratories and further implicates cellular membranes as sites of mutant Htt pathology.

Conceptual concerns

The most concerning aspect of this study is that it is performed entirely in Hek293 cells using an overexpressed fragment of Htt. The HD field largely has moved on from this type of artificial system since it misses several steps in human HD disease progression. It may, however, produce a somewhat accurate picture of end stage HD. But several questions remain unanswered: Are the same proteins recruited to early aggregates versus older aggregates? Also, by EM are membranes

incorporated early in small aggregates? What happens with longer pieces of HTT that contain the second membrane binding site and other protein interaction domains (such as HAP40)? The authors point out that HTT was found in the aggregates- was there evidence for full length HTT? How does this relate to what really happens in neurons expressing full length which is subject to proteolysis and also possibly exon1 readthrough product?

The study clearly identifies membranes within cytoplasmic aggregates although it is unclear if aggregates start to form on the surface of membranes as suggested years ago by Suopponki et al. ((2006) J. Neurochem. 96, 870–884.) or are simply nested within ER membranes. The ultrastructure results are reminiscent of findings by DiFiglia and colleagues (2000) who also showed membranes associated with aggregates formed by a larger tagged piece of HTT and in a neuronal subtype using electron microscopy; cisternae which were probably ER membranes were also described adjacent to aggregates. Those inclusions had highly electron dense regions and lacked the fibrillar aspect observed using exon1. The ring like structure with dense core and “halo” described for cytoplasmic aggregates has also been described by Qin and colleagues as pointed out by the authors in the Discussion. Kegel et al. also showed that numerous fragments of Htt co-distribute with membranes, supporting the authors findings with Exon1 and suggesting these current findings may be applicable in neuronal cell types and when generated from full length Htt. The authors find the presence of GFP on the C-terminus of Htt Exon1 alters fibrillar packing, changes antibody binding and alters protein and neutral lipid recruitment. One wonders whether the GFP fragment is more similar to a longer piece of Htt which might naturally occur such as a caspase fragment. The results somewhat mirror findings from Bäuerlein et al., although add new data due to the lack of GFP tag.

Major Experimental concerns:

1. For mass spectrometry is interesting but really needs to be performed on aggregates from neurons (preferable human neurons but mouse would do) or from human HD brain. There is no way to know if synaptic or other neuronal specific components such as channels are incorporated into aggregates.
2. Mitochondrial quantification is not clear and not describe in methods. I do not see how mitochondria length was determined. What does mitochondria “close to the inclusion” mean? How many nm? If all mitochondria in the cell with aggregates are counted are they all different? Do mitochondria away from the aggregate morphology?
3. Line 468 and Figure 3, Since authors say no inclusions formed with Httex 1 Q16, why do they think ERES size was reduced 20%?
4. It is not clear what is significantly different here. Where are asterix in fig?

Specific Comments:

1. The title is not appropriate; There is no sequence of events distinguished by these experiments. Unless you are referring to amino acid sequence? Not clear.

2. line 320 “neutral lipids were found to be enriched in the center of Httex1 72Q inclusions (Figure S9E, white arrowheads”
Which neutral lipids? Please identify or give examples.

3. line 357 “Among these proteins, we identified the endogenous Htt protein (Figure S11E).”
Is there evidence among the detected peptides that full length HTT is present? Could normal HTT be recruiting lipids through its second membrane binding region distal to exon1 or recruitment through palmitoylation at C214? Are there other proteins detected that could be interacting with membranes? (Since aa1-17 is not necessary?).

4. line 371 Approximately 14% of the 372 proteins enriched in the insoluble fraction were classified as pertaining to the cytoskeleton 373 compartment, with the actin cytoskeleton being the most predominant in terms of this 374 classification, consistent with our confocal results (Figure 1B).
This results about cytoskeleton is very interesting but not discussed much.

5. For the mass spectrometry, how many proteins were identified HTT interactor proteins?

Discussion:

6.

696 The ER at the periphery of the Httex1 72Q inclusions were also affected at the structural
697 level, as shown by their morphological reorganization in rosette or “stacked cisternae”, and at
698 the functional level as is evident from the dysregulation of the ERES homeostasis (Figure 5).
699 Formation of ER rosettes indicates the accumulation of proteins in the smooth ER59 and is 700 It
of low-affinity binding and export defects, which can be caused by
701 unfolded Htt proteins but is not necessarily linked to ER stress.

This is interesting but....

The presence of ribosomes

705 and ER membrane deformation was previously detected close to the periphery of Httex1
706 inclusions by cryo-ET15 and was linked to a strong reduction in ER dynamics.

Shown before??

7.

two major phases

791 with, first, the formation of the core driven primarily by the polyQ repeat domain, and then the
792 growth of the inclusions with the addition of Htt fibrils and the recruitment of other proteins
793 and organelles. The second phase appears to be driven by interactions involving both the
794 polyQ and PRD domains and involves the active recruitment and sequestration of lipids,
795 proteins, and membranous organelles (Figure 8A).

Does this mean you have to start with exon1 sized product to start the aggregate?

8.

On Line 647-649, The authors state “Our detailed analyses of the ultrastructure of the Httex1 72Q
inclusions by CLEM and detergent-free EM revealed their fibrillar nature but also showed a previously
unreported core and shell structural organization for mutant Httex1 inclusions.” Yet
Qin et al. 2004 describes a protease resistant fibrillar core with a protease sensitive shell composed
of globule structures that recruit endocytic proteins, a paper which the authors have actually
referenced. What finding then is previously unreported?

9.

Line 682 In addition,

683 the periphery of the inclusions was decorated with mitochondria and ER, as previously
684 reported for the Htt inclusions in cellular models¹⁸ and human tissue^{74,75}.

These changes in mitochondria morphology were associated with dysregulation of the
690 mitochondria respiration (Figure 4), consistent with previous studies demonstrating that
691 mutant Htt aggregates interact directly with outer mitochondrial membranes (in STHdh
692 cells)⁷⁶ and induce mitochondrial fragmentation (in primary neurons)⁷⁷.

It is not clear to me what your work adds above these studies?

10. I noticed there is not much in discussion about neutral lipids- where are they in cell and what is
recruiting them- longer pieces of HTT that have incorporated? Other proteins binding to them?

11, How do these MS findings compare to that of Hosp et al paper where mass spec was used MS on
aggregates in R6/2?

12. The authors seem to have skirted the issue of whether aggregates are toxic or not. While Arrasate
et al. 2004) and Saudou et al., (1998) were referenced, they are only discussed in terms of their use
of GFP tags. Why study aggregates if they are not toxic?

1461 87. Leitman J, Ulrich Hartl F, Lederkremer GZ. Soluble forms of polyQ-expanded
1462 huntingtin rather than large aggregates cause endoplasmic reticulum stress. Nat
1463 Commun. 2013;4:1-10. doi:10.1038/ncomms3753

13. In results, Line 453, "Httex1 72Q, Httex139Q, or Httex1 16Q" described being used, but later, Q39 is referred to. Is this intentional or a mistake (Q39 should be Q139?)

14. Line 458, define EV.

15.

Figure 1D.

Does a figure such as this exist in HD patients? Is this type of inclusion an artifact of overexpressing mutant exon1? Although a protein the size of exon1 may occur from expression of the RNA Intron readthrough in HD patients, especially in cells where somatic expansion has already occurred, or by proteolysis of acid peptidases, there are otherwise very low levels of small fragments in cells expressing endogenous full length mutant HTT, often detectable only by methods such as HTRF or MSD. Thus, the level of Exon1 sized protein in cells is likely only enough just for seeding. Therefore, this seems like a very artificial situation in which the entire aggregate is driven by the same size fragment. While of some esoteric interest, it is not very informative in terms of actual disease manifestation. I do agree that this work has yielded some interesting, though predictable findings. The CLEM nicely presents how the ER and mitochondria interact with this aggregate. Are the aggregates observed here including mostly ER membrane because you have focused on perinuclear aggregates?

Please find below our revised manuscript now entitled “***Nuclear and cytoplasmic huntingtin inclusions exhibit distinct biochemical composition, interactome and ultrastructural properties.***”

We would like to thank the referees for their very positive feedback on the quality, depth and significance of our work. We also greatly appreciate all their constructive comments and valuable suggestions and recommendations, which have tremendously improved the clarity and quality of the manuscript. We have made every effort to experimentally address their comments, provide the requested data, and perform all suggested analyses.

One of the main requests was that we should conduct similar studies to assess the biochemical and ultrastructural properties of nuclear Htt inclusion formation in primary neurons, which we have now done. Below you will find the list of the new figures/data that have been added to the revised manuscript in response to their comments, recommendations and requests:

- Fig. 5:** Confocal microscopy analysis and classification of the Httex1 inclusions formed in neurons revealed their morphological heterogeneity over time.
- Fig. 6:** Ultrastructural analysis of the nuclear Httex1 72Q inclusions formed in primary cortical neurons shows granular and filamentous structures by CLEM and Tomography.
- Fig. 8:** The presence of the GFP tag alters the kinetic, ultrastructural properties, protein content and toxicity of Httex1 72Q inclusions in primary neurons.
- Fig. S14:** Most of the neurons overexpressing Httex1 72Q show the presence of nuclear aggregates.
- Fig. S15:** Representative electron micrographs of primary neurons expressing Httex1 showing inclusion formation by Httex1 72Q and Δ Nt17 Httex1 72Q, but not 16Q, or GFP.
- Fig. S21:** Neurons overexpressing Httex1 72Q-GFP show the presence of nuclear aggregates only from D7.
- Fig. S22:** CLEM analysis of the ultrastructure properties of Httex1 72Q-GFP formed in glial cell and neurons.
- Fig. S23:** The enrichment of Httex1 aggregates in the soluble urea fraction of primary neurons was confirmed by Western Blot analysis.
- Fig. S24:** Proteomic analysis of Httex1 72Q vs. Httex1 16Q Urea soluble fraction revealed strong enrichment of the Ubiquitin-Proteasome System.
- Fig. S25:** Proteomic analysis of Httex1 72Q-GFP vs. GFP Urea soluble fraction revealed strong enrichment of the Ubiquitin-Proteasome System.

Please find below a point-by-point response to the referees’ specific comments and suggestions.

Major comments Referee #1

In the current manuscript, authors investigated the ultrastructural properties and protein composition of Htt inclusions in cells overexpressing mutant exon 1 of the Htt protein. Authors demonstrated that Htt inclusion formation and maturation are complex processes that, although initially driven by polyQ-dependent Htt aggregation, also involve polyQ and PRD domain-dependent sequestration of lipids and cytoplasmic and cytoskeletal proteins related to Huntington's disease (HD) dysregulated pathways, recruitment and accumulation of remodeled or dysfunctional membranous organelles and impairment of the protein quality control and degradation machineries. Authors provided a more in-depth analysis of the impact of fluorescent fusion proteins, on the biochemical and ultrastructural properties of Htt inclusions and their impact on cellular components and organelles. **These observations are important and worth reporting. Most methods are highly innovative and acceptable. Authors put huge effort into this important work and highly appreciable**

We thank referee 1 for the very positive feedback, and appreciation of the quality, depth, innovative nature and importance of our work, which he/she agree are worth reporting.

1. Most of the experiments were conducted using HEK 293 cells, **why not in other cells such as STHDhQ111/Q111.**

Response:

Although the STHDhQ111/Q111 cells are mouse striatal-like neurons expressing a humanized exon 1, the level of Htt inclusions formation is very low in these cells (Trettel et al. 2000; Q. Wang et al. 2011). As one of the key objectives of our study is to decipher the sequence (Nt17, polyQ repeat length, and GFP) determinants of cytoplasmic and nuclear Htt inclusion formation, we selected a cellular model that is characterized by abundant Htt inclusions formation in which manipulations of the Htt sequence are possible. Httex1 expression in HEK293 results in the formation of abundant and predominantly cytosolic Htt aggregates/inclusions. This remains one of the most commonly used cellular models of Htt aggregation and inclusion formation (Aron et al. 2018; Gerson et al. 2020; Jiang et al. 2020; Nguyen, Hamby, and Massa 2005; Ratovitski et al. 2009; Schilling et al. 2007; Scior et al. 2018; Singer et al. 2021; Sophie Vieweg et al. 2021; Waelter et al. 2001; Zheng et al. 2013). One additional advantage of HEK cells is that they offer unique opportunities to investigate the mechanisms and cellular determinants of Htt cytoplasmic inclusions, which are rare in primary neurons. Furthermore, they exhibit both cytoplasmic and nuclear inclusions, thus enabling investigating and comparing the effect of the cellular environment on Htt aggregation under identical conditions.

Finally, several papers have been published in Nat. Comms. Using this model, including recently published studies on the role of Htt splicing and the pathogenic contribution of Httex1 in HD (Neueder et al. 2018) or to investigate the role of protein aggregation and clearance mechanisms involved in HD (Klaips et al. 2020) and other neurodegenerative diseases (Park et al. 2020; P. Wang et al. 2017).

However, we fully agree with referee 1 about the importance of conducting our study in neuronal cells. Therefore, following her/his recommendation, we have now investigated Httex1 inclusions formation in primary neurons and investigated how sequences flanking the polyQ domain or the presence of a GFP tag influences: 1) the level and kinetics of aggregation in neurons (confocal and image-based quantification), 2) the subcellular localization of the newly formed aggregates (confocal imaging), 3) their ultrastructural properties (CLEM and

tomography analyses), 4) their protein composition (biochemistry and proteomic analyses) and 5) their toxicity properties (cell death assays). The results of all these studies have been included in the revised manuscript and in the new figures (Figures 5, 6, 8 and Figures S14, S15 and S21-25)

2. Although **findings are highly innovative and important**, how the present work will advance the HD field in terms of **therapeutics**. Hope this is not inclusions paper (there are so many published already)

Response: We thank the referee for highlighting the innovative aspects and importance of our work. We believe that our work provides important and novel insights that not only advance our understanding of the mechanisms of Htt aggregation, but also point to new directions for therapeutic interventions.

1. At the mechanistic level, our work shows that Htt aggregation and inclusion formation in the cytosol and nucleus occur via different mechanisms and lead to the formation of inclusions with distinct biochemical and ultrastructural properties. These observations suggest that the two types of inclusions may exert their toxicity via different mechanisms and may require different strategies to interfere with their formation, maturation and toxicity.
2. We show that Htt cytoplasmic inclusion formation occurs via two phases:
 - a. A first phase involves the rapid formation of a dense fibrillar core and is driven predominantly by intermolecular interactions involving the polyQ domain via phase separation-like mechanisms.
 - b. A second phase is associated with the recruitment of soluble Htt, fibril growth, and the active recruitment and sequestration of lipids, proteins, and membranous organelles. Our work points to this second phase as an important contributor to Htt toxicity and suggests that targeting inclusion growth and maturation represent a promising therapeutic strategy.
3. In addition, we reported a polyQ and PRD domain-dependent sequestration of lipids and cytoplasmic and cytoskeletal proteins related to HD dysregulated pathways. Moreover, the accumulation of mitochondria and ER network at the periphery of inclusions led to functional defects, including mitochondrial respiration adaptation and ER trafficking modulation, known to be dysregulated in HD. In the new work we performed in primary neurons, we clearly establish that neuronal intranuclear inclusions evolve over time from small aggregates to large granulo-filamentous inclusions and that this process is associated with increased cellular toxicity. These findings suggest that identifying modifiers of Htt inclusion growth and aberrant secondary interactions with other proteins and organelles represent an alternative strategy for interfering with Htt-induced toxicity and slowing disease progression, especially after disease onset. Equally important, our work emphasizes the importance of elucidating the role of Htt interactions with lipids and membranous organelles in regulating the process of Htt aggregation, inclusion formation, and toxicity.

These points are now highlighted in the discussion section of the paper (Conclusion sections page 37-38, lines 932-958). Finally, we would like to emphasize that our comparative analysis of untagged and GFP-tagged mutant Httex1 aggregation and inclusions formation will inform future efforts to develop models that reproduce HD pathology more faithfully and the

underscore the need for developing label-free techniques to investigate disease-relevant mechanisms that underpin inclusion formation

3. Literature survey mostly well done, but important recent EM studies on mitochondria – Yin et al Hum Mol Genet 2016 and Manczak and Reddy, Hum Mol Genet 2015.

Response: We thank the referee for bringing these references to our attention. They are now included and commented on in the revised manuscript (Manczak and Hemachandra Reddy 2015; Yin, Manczak, and Reddy 2016), page 29, lines 721-724. These studies show that mutant Htt induces mitochondrial fragmentation in STHdh neurons, which is in line with our results.

4. Hyper activation of mitochondrial respiration is observed in cells with 72polyQ, is it due to compensatory response due to mitochondrial dysfunction in HD neurons? – please comment.

Response: As mentioned in the end of the result section, a similar hyperactivation of mitochondrial respiration has recently been reported for alpha-synuclein aggregates (Ugalde et al. 2020). The authors hypothesized that the hyper-respiration might represent a pathogenic upstream event to alpha-synuclein pathology. We think that this may also be the case for untagged Httex1 72Q induced pathology. Based on our extensive proteomic analyses (Figures 3 and 9), we are able to extend this hypothesis. In the tag-free Httex1 72Q condition, mitochondrial hyperactivity coincided with both mitochondrial fragmentation and the detection of more inflammation-related proteins in the aggregates as compared to the tagged condition. We speculate that mitochondrial fragmentation may be initially protective, as it enhances mitochondrial mobility that is necessary for cell repair processes (Horn et al. 2020). Upregulation of mitochondrial respiration would be a logical consequence of elevated ATP requirements for mitochondrial and protein transport related to aggregation formation and proteostatic processes, including ATP-requiring chaperone proteasome and autophagy processes. The combination of mitochondrial fragmentation and hyperactivity could result in the release of mitochondrial damage-associated molecular patterns (DAMPs), such as mitochondrial DNA, membrane components, metabolites or ROS – which are potent triggers of inflammatory processes (Grazioli and Pugin 2018). We think that these tag-free Httex1 72Q specific processes drive the characteristic pathology formation. The discussion section has been expanded in order to add more clarity as requested by the referee (see page 29, lines 724-743).

5. ER-mitochondrial association is increased in mutant Htt cells – if yes, what are the thoughts on this concept.

Response: We agree with the referee that this mechanism is important to explore in the future but is beyond the scope of this current study.

It has been recently reported that a decrease of ER-mitochondrial association in R6/1 primary neurons (Cherubini, Lopez-Molina, and Gines 2020), could alter mitochondrial dynamics and bioenergetics as well as Ca²⁺ homeostasis. These results are in line with our observations.

6. Autophagy/mitophagy seems to be defective in mutant htt cells – any thoughts on autophagy/mitophagy inducers to enhance Autophagy/mitophagy HD.

Response: The role of autophagy/mitophagy is of great interest to our lab. To explore the interplay between these pathways and Htt aggregation and toxicity, we sought to develop well-characterized models that would enable. With these models in hand, we plan to pursue these studies soon. We recently reported a detailed mechanistic study on the role of Httex1 phosphorylation and enhanced autophagy clearance of mutant Httex1 monomers in regulating Htt inclusion formation and Htt-induced toxicity (Hegde et al. 2020).

Major comments Referee #3

This study investigates the nature of aggregates formed by a small piece of mutant HTT in cells both at the ultrastructure level and biochemical composition. **Strengths include the rigorous use of correlative light electron microscopy (CLEM) and serial reconstruction to visualize Htt aggregates in a native space. A major weakness is the lack of neuronal context in all studies and the use of only Htt exon1**, although an important comparison to Htt exon1-GFP which has been widely used in the field offers **eye-opening comparisons**. Mass spectrometry identifies both nuclear and cytoplasmic components within aggregates. **Unfortunately, the mass spectrometry study offers limited useful information since it too was performed in HEK cells.**

The authors even conclude in the Discussion that the cell environment contributes to process of aggregate formation and protein recruitment (lines 742-747). **If this is true then surely the work needs to be performed in neurons.** The work does extend findings of previous laboratories and further implicates cellular membranes as sites of mutant Htt pathology.

We thank referee #3 for her/his positive feedback and for highlighting our rigorous approach to characterize Httex1 aggregation at the biochemical and structural levels and appreciation of the new insights we provide regarding the impact of GFP on the ultrastructural, biochemical and toxic properties of the Htt inclusion.

We agree with referee #3 that extending our work by investigating Htt inclusion formation in neurons is important. Following his/her recommendations, we have now performed similar studies (confocal imaging, toxicity, CLEM and proteomic analysis) to characterize Htt nuclear inclusions at the ultrastructural and biochemical levels, including performing mass spectrometry studies on neuronal Htt inclusions, as requested by the referee (Figures 5, 6, 8 and Figures S14, S15 and S21-25).

Conceptual concerns:

1. The most concerning aspect of this study is that it is performed entirely in Hek293 cells using an overexpressed fragment of Htt. The HD field largely has moved on from this type of artificial system since it misses several steps in human HD disease progression. **It may, however, produce a somewhat accurate picture of end stage HD.**

Response:

We understand this point of concern. Following referee #3' comments, additional work has been conducted in primary neurons (confocal imaging, toxicity, CLEM and proteomic analysis): Figures 5, 6, 8 and Figures S14, S15 and S21-25.

The original goal of the study was to assess the importance of the polyQ length and the Nt17 domain in the mechanisms and ultrastructural properties of cytoplasmic Httex1 inclusions. To do this, we selected a robust cellular system that exhibits abundant cytoplasmic inclusions and allows for sequence modifications. More details on the rationale to use HEK cells are provided in the reply to the comment #1 of the referee 1 page 2-3.

Furthermore, other mammalian overexpression-based cellular models of Htt aggregation (including non-differentiated neuronal cells) are still being extensively used by scientists in the HD field (Bäuerlein et al. 2017; Goold et al. 2019; Y. E. Kim et al. 2016; Lu et al. 2019; Luo et al. 2018; Ramdzan et al. 2017; Shen et al. 2016; Sun et al. 2020; Ylä-Anttila, Gupta, and Masucci 2021).

The great majority of the cellular models rely on the expression of N-terminal fragments of Htt to study inclusion formation (Cisbani and Cicchetti 2019). In the absence of overexpression, such as in STHdh mouse striatal cell lines (Trettel et al. 2000; Q. Wang et al. 2011) or patient-derived cells, only very few Htt inclusions are observed. Therefore, these models do not allow interrogating the entire process of Htt aggregation, inclusion formation and maturation of the endogenous proteins in cells.

Use of only Htt exon1:

Response: We are convinced that the exon 1-based models remain relevant to study the pathogenesis of HD, with the understanding that no single model recapitulates all the disease-relevant processes. It has been shown that depending on the CAG repeat length, exon 1 of the huntingtin gene (*htt*) can undergo aberrant splicing, resulting in production of a small polyadenylated transcript that encodes the exon 1 Htt protein (Sathasivam et al. 2013). In addition, every knock-in mouse model –and transgenics for the full-length human HTT gene (e.g. YAC128)— produce the Httex1 protein, and disease progression in KI-mice correlates with the level of incomplete splicing and appearance of aggregates (Franich et al. 2019; Sathasivam et al. 2013). Incomplete splicing of *Htt* leading to Httex1 protein expression has been shown to occur in HD patients brains (Neueder et al. 2017), and Httex1 protein was previously described as a key component of the intracellular inclusions found in HD *post-mortem* brains (Neueder et al. 2017; Schilling et al. 2007; Wellington et al. 2002).

Furthermore, the expression of pathogenic Httex1 was sufficient to induce HD-like features, including aggregates formation and toxicity in mice (Davies et al. 1997; Mangiarini et al. 1996; Martindale et al. 1998), *Drosophila* (Barbaro et al. 2015), and *C. elegans* (H. Wang et al. 2006). These models reproduce different aspects of Htt aggregation and have been instrumental in advancing our understanding of the sequence, molecular, and structural determinants of Htt aggregation and inclusion formation (Gu et al. 2009; Rockabrand et al. 2007; Steffan et al. 2004; Tam et al. 2009; Thakur et al. 2009; Thompson et al. 2009; Zheng et al. 2013).

Since the discovery of the *htt* gene in 1993, very few studies have shown evidence of inclusion formation in cells expressing full-length Htt (M. Kim et al. 1999; Lunkes and Mandel 1998; Martindale et al. 1998). Most of these studies were based only on fluorescence microscopy and rarely exhibited inclusion formation. Furthermore, even if inclusions were detected, it was not clear whether they were formed by full-length Htt or N-terminal Htt fragments. Moreover, the formation of Htt aggregates was not validated and characterized at the biochemical level. Therefore, to study Htt aggregate formation in a cellular model, smaller N-terminal fragments of Htt are commonly used (Httex1, Htt171, Htt586). However, mutant Httex1 remains the most commonly used fragment because it is associated with robust inclusions formation compared to other N-terminal fragments, including Htt171 or the longer caspase fragments (500-600) (Cooper et al. 1998; Hackam et al. 1998; Lunkes and Mandel 1998; Martindale et al. 1998).

Even though the Htt586 fragment is naturally produced by caspase 6 cleavage of the Htt protein (Barbaro et al. 2015; Graham et al. 2010; Landles et al. 2010; Wellington et al. 2002), we are not aware of any studies that have reported robust aggregation and inclusion formation in cells expressing this fragment (El-Daher et al. 2015; Warby et al. 2008).

- But several questions remain unanswered: Are the same proteins recruited to early aggregates versus older aggregates?

Response: We agree that this is an important question that should be addressed. It has been previously shown that the soluble and insoluble brain proteomes of R6/2 mice evolve and change with disease progression (Hosp et al. 2017).

We have performed time-dependent studies of Htt aggregation in cells. These studies showed that the first steps of inclusion formation occur very rapidly (less than 1h for Httex1 72Q), most likely driven by phase separation (Figure I) (Peskest et al. 2018). We are in the process of conducting further studies to systematically capture and characterize the dynamics of these early events. These studies are important but challenging and would require more time and new approaches, including the use of label-free methods, which we believe is necessary given the impact of fluorescent proteins on Htt aggregation and inclusion formation. We believe that this work is beyond the scope of this manuscript.

Figure I. The dynamic of Httex1-GFP inclusion formation is driven by the polyQ domain. A. Time-lapse fluorescence microscopy of 72Q-GFP aggregation. **B.** Intensity of EGFP over time of transfected cells with Httex1 72Q-GFP rescaled on the IC50. The mean of the relative slope of Httex1 72Q-GFP intensity at the inflection point of aggregation is indicated on the graph. **C.** Surface plot display of the EGFP intensity over time of cell (A). **D.** Time-lapse fluorescence microscopy of 39Q-GFP aggregation. **E.** Intensity of EGFP over time of transfected cells with Httex1 39Q GFP rescaled on the IC50. **F.** Surface

plot display of the EGFP intensity over time of cell (A). The mean of the relative slope of Httex1 72Q-GFP intensity at the inflection point of aggregation is indicated on the graph.

3. Also, by EM are membranes incorporated early in small aggregates?

Response: The assessment of the role of membranes in early aggregation events at present is impeded by technical limitations. To be able to identify membranous structures with preserved integrity in the aggregates, detergents have to be avoided. Therefore, this precludes the use of ICC and confocal imaging to identify the aggregates formed from Httex1 72Q that do not carry the GFP tag, which makes the selection of the cells carrying the small aggregates formed early in the aggregation process extremely challenging. Hence, we are currently unable to perform EM on the small aggregates. We are still optimizing the conditions using fluorescent lipid probes, which we hope will allow us to monitor the early events of aggregate formation and capture their dynamics and ultrastructural properties by EM.

However, based on our conceptual model and the literature, we believe that the first phase of Httex1 aggregation occurs through the rapid formation of a dense fibrillar core and is driven predominantly by intermolecular interactions involving the polyQ domain via phase separation-like mechanisms, as recently proposed (Peskett et al. 2018).

4. What happens with longer pieces of HTT that contain the second membrane binding site and other protein interaction domains (such as HAP40)?

Response:

We are not aware of any Htt fragments that have been shown to retain the ability to bind HAP40. The HAP40 binding sites are formed by multiple residues from different parts of the protein, including the N-HEAT, the bridge C-HEAT huntingtin regions (Guo et al. 2018). Thus, we do not have any guidance as to which is the right relevant Htt fragment that could be used to answer the question raised by the referee. We are not sure what the referee means by the “second membrane binding site.”

We agree that additional studies are needed to more precisely define the disease-relevant Htt N-terminal fragments and those involved in Htt aggregation in HD brain. In a recent study from our group, we demonstrated that the longer N-terminal fragment Htt171 aggregates *in vitro* via different mechanisms and form Htt aggregates that are distinct from those formed by Httex1 (Kolla et al. 2021). We also showed that the structured domains outside exon1 could play important roles in regulating early events associated with Htt oligomerization and the initiation of Htt aggregation and inclusion formation. We also performed additional studies in primary neurons using two additional N-terminal fragments, namely the Htt171 and Htt586. We observed neuronal intranuclear inclusions upon overexpression of Htt171 82Q but not Htt586 82Q, consistent with previous data (Carnemolla, Michelazzi, and Agostoni 2017; Cooper et al. 1998; El-Daher et al. 2015; Saudou et al. 1998; Warby et al. 2008).

5. The authors point out that HTT was found in the aggregates- was there evidence for full length HTT?

Response: Yes, our proteomic analysis revealed several peptides corresponding to the HTT sequences outside the Httex1 domain. These peptide sequences were detected for both in

the Urea soluble fractions from cells expressing Httex1 72Q or Httex1 72Q-GFP (see Figure S11E).

6. How does this relate to what really happens in neurons expressing full length which is subject to proteolysis and also possibly exon1 readthrough product?

Response: Previous studies reported that full-length Htt undergoes proteolysis leading to the generation and accumulation of different N-terminal fragments, some of which are found in N-Htt inclusions (Barbaro et al. 2015; Graham et al. 2010; Landles et al. 2010; Lunkes et al. 2002; Wellington et al. 2002).

As mentioned above (see comment #1 from referee 3 page 5-7), inclusions were rarely detected in cells expressing full-length Htt (FL-Htt). Unfortunately, the characterization of these inclusions was limited to fluorescence microscopy using N-terminal antibodies or peptide tags (FLAG) (Kegel et al. 2000; M. Kim et al. 1999), which does not allow one to distinguish between aggregates formed by full-length Htt or its N-terminal fragments. 1) Cooper and colleagues reported only 1% of N2a cells transfected with both 23Q and 82Q FL-Htt forming inclusions (Cooper et al. 1998); 2) Hackam and colleagues observed less than 1% of the transfected HEK cells with FL-Htt 128Q formed aggregates (Hackam et al. 1998); 3) Kim and colleagues reported less than 10% of cells with nuclear inclusions in a mouse striatal cell line expressing FLAG-tag FL-Htt (M. Kim et al. 1999); 4) Kegel and colleagues expressed mutant FL-Htt in clonal striatal cells and observed only cytoplasmic Htt vacuoles (Kegel et al. 2000). Unfortunately, none of these studies performed biochemical studies to assess the sequence or distribution of Htt species in these inclusions, thus making it difficult to assess if they are formed as a result of the aggregation of full-length Htt or N-terminal fragments of the protein.

Overall, these studies do not show strong evidence of FL-Htt initiating the aggregation process. This, combined with the fact that full-length mutant Htt does not fibrillize *in vitro*, suggests that Htt fragments rather than full-length Htt are the primary drivers of inclusion formation in HD. The presence of the full-length inclusion may reflect its recruitment at later stages of inclusion formation, possibly through polyQ-mediated interactions with N-terminal aggregates.

7. The study clearly identifies membranes within cytoplasmic aggregates although it is unclear if aggregates start to form on the surface of membranes as suggested years ago by Suopanki et al. ((2006) J. Neurochem. 96, 870–884.) or are simply nested within ER membranes.

Response: We thank referee #3 for highlighting this point. We do not exclude the potential role of membranes in promoting early aggregation events as previously suggested by Suopanki and colleagues (Suopanki et al. 2006).

Increasing evidence points to a complex interplay between amyloid-forming proteins and lipids, and membranous organelles in the formation of pathologic inclusions in PD. Our work presented here and previous studies by DiFiglia and colleagues suggest that this could also be the case for cytoplasmic Htt inclusion formation. However, more studies are needed to dissect these interactions and determine the role of Htt-membrane interactions at different stages of the process, i.e., protein misfolding, fibril formation and inclusion formation and maturation. This is an exciting line of research that we are very much interested in pursuing but was beyond the scope of this manuscript. The manuscript has been updated to highlight these possibilities page 10, lines 216-218.

8. The ultrastructure results are reminiscent of findings by DiFiglia and colleagues (2000) who also showed membranes associated with aggregates formed by a larger tagged piece of HTT and in a neuronal subtype using electron microscopy; cisternae which were probably ER membranes were also described adjacent to aggregates. Those inclusions had highly electron dense regions and lacked the fibrillar aspect observed using exon1.

Response: We agree and thank the referee for highlighting similarities in our findings and those previously reported by DiFiglia and colleagues (2000). We believe that these observations support the use of HEK293 as a useful model to investigate the mechanisms of Htt and polyQ aggregation and inclusion formation and the cellular bias underlying Htt-induced toxicity. The work by DiFiglia and colleagues (2000) is now highlighted and discussed in the discussion section of the paper (see page 32, lines 792-794).

9. The ring like structure with dense core and “halo” described for cytoplasmic aggregates has also been described by Qin and colleagues as pointed out by the authors in the Discussion.

Response: We agree with referee #3 about the relevance of the study from Qin and colleagues in the context of our findings. This is why we have discussed this work (Qin et al. 2004) in our paper (see page 35, lines 875-878 in the original version of the papers). They observed a core and halo organization of Htt inclusions by EM using a longer fragment than Httex1 (FLAG-tagged Htt1-969 fragment (100Q)).

Similar to our study, Qin and colleagues detected ring-like cellular inclusions using Htt targeted antibodies by fluorescence microscopy. However, whereas the report by Qin *et al.* provides only a morphological description of these inclusions, our work provides a more in-depth characterization of these inclusions at the biochemical and ultrastructural levels.

One notable difference is that they detected Htt in the core of their Htt1-969 inclusions in MCF-7 cells using the mutant specific EM48 antibody, unlike Httex1 inclusions in our HEK cell model. However, one should be cautious in interpreting the electron micrographs of Qin *et al.* as they used an immunoperoxidase labeling method, which could by itself give rise to this halo staining. Conversely, we were able to detect the ring-like structure with the dense core and “halo” in detergent- and antibodies-free conditions.

Therefore, it remains unclear if the ring-like morphologies of mutant Httex1 and Htt1-969 are identical.

10. Kegel et al. also showed that numerous fragments of Htt co-distribute with membranes, supporting the authors findings with Exon1 and suggesting these current findings may be applicable in neuronal cell types and when generated from full length Htt.

Response: We agree and thank referee #3 for bringing this work to our attention. As mentioned above, on point 8 page 10, we now discuss and cite this work in the revised version of the manuscript (see page 32, lines 792-794).

11. The authors find the presence of GFP on the C-terminus of Htt Exon1 alters fibrillar packing, changes antibody binding and alters protein and neutral lipid recruitment. One wonders whether the GFP fragment is more similar to a longer piece of Htt which might naturally occur such as a caspase fragment. The results somewhat mirror findings from Bäuerlein et al., although add new data due to the lack of GFP tag.

Response: We thank the referee for this interesting perspective. In our opinion, it is unlikely that the GFP sequence at the C-terminus of Httex1 acts similar to longer Htt fragments in cells because of large differences in the amino acid sequences and secondary structure properties of the Htt sequence (HEAT repeats) compared to the b-barrel structure of GFP. We believe that the sequence and structure of the segments outside exon1 play an important role in regulating the mechanism of Htt aggregation. In fact, we have recently shown that Htt171 aggregates via distinct mechanisms. Whereas poly-Q-mediated processes initially drive Httex1 aggregation, the aggregation of Htt171 is driven by phase separation events mediated by a complex interplay between the helical domain comprising residues 104-171 and the polyQ domain (Figure II, (Kolla et al. 2021)). Moreover, site-specific phosphorylation of Htt outside exon1 (T107 and S116) has also been shown to modulate Htt aggregation. Therefore, we doubt that the GFP sequence will reproduce the fine-tuning of Htt aggregation mediated by *bona fide* PTMs, structured domains, and aggregation motifs outside exon1.

Figure II. Top panel: Huntingtin aggregation is determined by the presence and length of the C-terminal structured domain, as well as the presence of site-specific phosphorylation. **Lower panel:** A schematic model for aggregation of Htt171 and Httex1. **A)** We propose that the initial step for the aggregation of Htt171 (Nt17, blue; polyQ, red; proline-rich domain, yellow; green, C-terminal region) is the association of the C-terminal structured domain to form oligomers. These oligomers undergo conformational reorganization and form polyQ driven β -sheet rich nucleation sites. Then the elongation of the nuclei by recruiting monomers generates fibrils. **B)** The disordered monomeric Httex1 spontaneously assembles into an oligomer formation which is mediated by the association of the α -helical conformation of Nt17. Oligomerization results in a high local concentration of polyQ and a subsequent conformational switch of polyQ in β -sheet structure. Elongation of nuclei by monomer addition generates amyloid fibrils. **C)** The Httex1 acquires a tadpole-like conformation in the monomeric form. The Nt17 interacts with the

polyQ, which enhances intermolecular hydrophobic interactions and facilitates oligomerization. The high local concentration of polyQ in oligomers causes a conformational switch to β -sheet structure, followed by fibril growth. Figure adapted from Kolla et al. 2021.

Experimental concerns:

1. For **mass spectrometry** is interesting but really needs to be performed on aggregates from **neurons** (preferable human neurons but mouse would do) or from human HD brain. There is no way to know if synaptic or other neuronal specific components such as channels are incorporated into aggregates.

Response: We agree. As suggested by referee #3, we have now performed a proteomic analysis of the protein content of the Httex1 inclusions formed in mouse primary neurons. We specifically looked at synaptic proteins or neuronal components in the proteomic analysis as asked by the referee. However, we did not observe specific neuronal proteins, although we found that the GO term "Ion transport" from the Biological Process was significantly enriched due to the presence of Sumo1 and Ubqln1 proteins.

Our analysis shows that the most enriched pathways are related to nuclear processes and protein degradation with the UPS, autophagy and ERAD. Nuclear-related proteins such as transcription factors, RNA/DNA and chromatin binding proteins are in line with the nuclear alterations observed in our neuronal HD model (chromatin condensation, nuclear fragmentation, and nuclear envelope integrity loss). In addition, enrichment of protein degradation pathways indicates either a failure of the cellular degradation systems to clear the aggregates by their sequestration or the upregulation of those pathways to cope with such large inclusions, similar to what we observed in the HEK cells results. More details are now provided and discussed in the revised manuscript (pages 24-26, Lines 589-656 and Figure 8),

We also agree with the referee that performing proteomics on Htt inclusions from HD brains will be of great importance for future studies. However, repeating these experiments in neurons from HD brain is beyond the scope of this project. These experiments remain highly challenging due to the lack of protocols for the reproducible isolation of Htt inclusions from HD brain.

2. **Mitochondrial quantification** is not clear and not describe in methods. I do not see how mitochondria length was determined. What does mitochondria "close to the inclusion" mean? How many nm? If all mitochondria in the cell with aggregates are counted are they all different? Do mitochondria away from the aggregate morphology?

Response: We thank the referee for pointing out this lack of clarity. The measurements were performed on mitochondria identified in close proximity to the Httex1 inclusions (Figures 4A, S28A and B).

The mitochondrial profile length corresponds to the maximal length of each mitochondrion in one EM plane. The distance from the inclusion was not considered, as the measurements were performed from one plane using the image processing software FIJI. Instead, the average length of all detected mitochondria was calculated and yielded significant differences in the Httex1 72Q condition compared to the EV control.

We updated and clarified the material and method section accordingly. Please see the section on "Sample processing for electron microscopy imaging without cell permeabilization"(see page 7 of the supporting information document, lines 180 to 186).

3. Line 468 and Figure 3, Since authors say no inclusions formed with Httex1 Q16, why do they think ERES size was reduced 20%?

Response: ERES number and size from cells containing Httex1 72Q inclusions were compared to ERES in cells expressing Httex1 16Q (diffuse and soluble form of Httex1). We observed a 20% reduction of ERES size for Httex1 16Q compared to the empty vector control (EV) Figure 4D-E.

A similar result was observed for GFP and Httex1 16Q-GFP ERES number (Figure S29A-B). These results indicate that the expression of Httex1 by itself (without the formation of inclusions) decreases the length of ERES, and this effect is significantly increased in the presence of inclusions. The remodeling of ERES in cells has been described primarily as an adaptive response to the protein synthesis level of ER with the number of ERES proportional to the cargo load. Even if the expression of Httex1 already elevates this level in the absence of inclusions, we demonstrated a significantly larger effect in the presence of Httex1 inclusions: Significant reduction of ERES size for Httex1 39Q, 72Q (Figure 4D-E) and 72Q-GFP (Figure S29A-B). The results are discussed page 32 lines 787-810.

4. It is not clear what is significantly different here. Where are asterisk in fig?

Response: Figure 4C: We thank the referee for highlighting this lack of clarity. The asterisks Figure 4C are next to "polyQ repeat length x respirational state". The test used here is a 2-way ANOVA where the row factors are the "respirational states" and the column factors are the "PolyQ repeat length" conditions". Therefore, the asterisks indicate a significant interaction between PolyQ repeat length x respirational states, meaning that the respirational state changes in dependence of the PolyQ repeat length (Httex1 72Q vs. Httex1 16Q). *P < 0.05, **P < 0.005, ***P < 0.001). The graph represents the mean \pm SD of 4 independent experiments. The figure and figure legend of Figure 4C was updated for more clarity.

Minor points, Referee #3

Below is a point-by-point response to the referee's comments and questions:

Specific Comments:

1. The title is not appropriate; There is no sequence of events distinguished by these experiments. Unless you are referring to amino acid sequence? Not clear.

Response: We agree with the referee. We have modified the title to ***“Nuclear and cytoplasmic huntingtin inclusions exhibit distinct biochemical composition, interactome and ultrastructural properties”***

2. line 320 “neutral lipids were found to be enriched in the 321 center of Httex1 72Q inclusions (Figure S9E, white arrowheads” **Which neutral lipids?** Please identify or give examples.

Response: We thank the referee for pointing out this lack of clarity in the text. The neutral lipid stain used in Figure S9D-E is the BODIPY™ 493/503 (4,4-Difluoro-1,3,5,7,8-Pentamethyl-4-Bora-3a,4a-Diaza-s-Indacene), which stains neutral lipids in lipid droplets that are mainly composed of triacylglycerols (TAGs). Phospholipids usually surround lipids droplets. However, in our results, the neutral lipids are found in the core of Httex1 72Q inclusions and are negative for phospholipids and cholesterol esters (Figure S9)

3. line 357 “Among these proteins, we identified the endogenous Htt protein (Figure S11E).” **Is there evidence among the detected peptides that full length HTT is present?**

Could normal HTT be recruiting lipids through its second membrane binding region distal to exon1 or recruitment through palmitoylation at C214? Are there other proteins detected that could be interacting with membranes? (Since aa1-17 is not necessary?).

Response: Our proteomic analysis indicated that full-length HTT is present in the aggregate fraction as shown by the presence of peptides corresponding to the HTT sequence outside Httex1 in both Httex1 72Q and Httex1 72Q-GFP (Figure S11E). See response to point # 1.

The referee raises an interesting hypothesis that is worth investigating in future studies. However, the C214 HTT amino acid was not detected in our proteomic analysis. In addition, the low abundance of full-length HTT inside Httex1 inclusions is unlikely to explain the recruitment of membranes.

To further investigate this, we assessed if the proteins significantly enriched in the Httex1 72Q or Httex1 72Q-GFP urea fraction, compared to Httex1 16Q, contained any proteins with membrane-binding domains. We used the Interpro (Blum et al. 2021) and Pfam (Mistry et al. 2021) protein classification databases to detect any membrane-associated proteins with the protein-enriched for Httex1 72Q and Httex1 72Q-GFP. After removal of transmembrane proteins from the analysis, only two proteins could be detected: the Vesicle transport through interaction with t-SNAREs homolog 1B (VTI1B) and Golgi resident protein GCP60 (ACBD3).

4. line 371 Approximately 14% of the 372 proteins enriched in the insoluble fraction were classified as pertaining to the cytoskeleton 373 compartment, with the actin cytoskeleton being the most predominant in terms of this 374 classification, consistent with our confocal results (Figure 1B). **This results about cytoskeleton is very interested but not discussed much.**

Response: We agree with the referee and have updated the discussion section to elaborate on the potential implications of these findings (see pages 20-21, lines 502-510).

The presence of Actin-F associated with the cytoplasmic inclusions in HEK cells, confirmed both by confocal imaging (Figure 1B, S9D and S10A) and proteomic analyses (Figure 3B), suggests a potential role of the actin in the formation or maturation of Httex1 inclusions. In line with our findings, several studies have shown that the level of Httex1 aggregation and the number of inclusions formed in mammalian cell lines is influenced by the ability of Httex1 to interact with the microtubule cytoskeleton (Liu et al. 2011; Muchowski et al. 2002; Taran, Shuvalova, and Lagarkova 2020) and actin filaments (Angeli, Shao, and Diamond 2010; Liu et al. 2011). In addition, the actin cytoskeleton was detected as one of the top dysregulated pathways from a proteomic analysis conducted in human HD brains (Ratovitski et al. 2016), underscoring its importance in the development of HD pathology.

5. For the mass spectrometry, how many proteins were identified HTT interactor proteins?

Response: To answer this question, we compared the proteins significantly enriched in Httex1 72Q and Httex1 72Q-GFP of the aggregate fraction in HEK cells to known HTT interactor proteins using the HDinHD dataset (<https://www.hdinhd.org/>). Known HTT interactors from HDinHD Human and mouse datasets were selected among cell- or animal-based studies exclusively. In HEK cells, our analysis revealed the enrichment of 42 known Htt interactors in Httex1 72Q and 21 in Httex1 72Q-GFP. In primary neurons, 5 known Htt interactors have been identified in Httex1 72Q and 7 in Httex1 72Q-GFP. The classification of these proteins using GO term analysis showed an enrichment of the molecular functions related to Heat shock protein binding, chaperone binding and polyubiquitin-dependent protein binding, in line with our analysis (See Figures 3 and 9). These results are displayed Table S1, in the manuscript pages 15 lines 360-363 and page 25 lines 624-626; and explained in the material and method (SI document) under “Preparation of samples for mass spectrometry” pages 11-12 lines 296-298

Discussion:

6. 696 The ER at the periphery of the Httex1 72Q inclusions were also affected at the structural level, as shown by their morphological reorganization in rosette or “stacked cisternae”, and at the functional level as is evident from the dysregulation of the ERES homeostasis (Figure 5). Formation of ER rosettes indicates the accumulation of proteins in the smooth ER59 and is 700 It of low-affinity binding and export defects, which can be caused by unfolded Htt proteins but is not necessarily linked to ER stress. The presence of ribosomes and ER membrane deformation was previously detected close to the periphery of Httex1 inclusions by **cryo-ET15** and was linked to a strong reduction in ER dynamics. **Shown before??**

Response: In the study by Bäuerlein *et al.*, the authors indicated ER membrane deformation and reduction in ER dynamics upon inclusion formation but did not report on specific ER

arrangement or the formation of rosette-like structures. In addition, in our study, we looked more precisely at the ER exit sites (ERES) to assess if the ER was functionally affected in the secretory pathway. We observed a decrease in ERES number and size compared to non-mutant Httex1 expressing cells. ERES remodeling indicates an adaptative response due to inclusion formation and a change in ER homeostasis. Moreover, we reported a significant polyQ dependent reduction of ERES number for Httex1 72Q compared to Httex1 39Q. Bauerlein and colleagues limited their study to ER dynamic measurements only in Htt97Q-GFP overexpressing HeLa cells.

7. two major phases with, first, the formation of the core driven primarily by the polyQ repeat domain, and then the growth of the inclusions with the addition of Htt fibrils and the recruitment of other proteins and organelles. The second phase appears to be driven by interactions involving both the polyQ and PRD domains and involves the active recruitment and sequestration of lipids, proteins, and membranous organelles (Figure 8A). **Does this mean you have to start with exon1 sized product to start the aggregate?**

Response: We believe that aggregation could be initiated by Httex1 or other N-terminal longer fragments containing the polyQ domain. Our group just published a paper where we reported that longer Htt fragments (Htt104, Htt140 and Htt171) could also initiate aggregation, albeit via distinct mechanisms as discussed above points #4 page 8 and #11 page 11 of referee 3 (Kolla et al. 2021). We believe that the early aggregation events occur rapidly and could be mediated by the polyQ or other more C-terminal domains outside Httex1 (residues 104-171) depending on the fragment of Htt. In our laboratory, we have failed to induce fibrillization of the full-length Htt protein, despite trying a large number of conditions. To the best of our knowledge, there is no strong evidence demonstrating that full-length HTT aggregates on its own and drives the aggregation process. Therefore, it is our working hypothesis that Htt cleavage and generation of N-terminal fragments containing the polyQ domain is a key prerequisite step for Htt aggregation. The elucidation of the necessary events leading to Htt aggregation remains a key challenge for the field.

8. On Line 647-649, The authors state “Our detailed analyses of the ultrastructure of the Httex1 72Q inclusions by CLEM and detergent-free EM revealed their fibrillar nature but also showed a previously unreported core and shell structural organization for mutant Httex1 inclusions.” Yet Qin et al. 2004 describes a protease resistant fibrillar core with a protease sensitive shell composed of globule structures that recruit endocytic proteins, a paper which the authors have actually referenced. **What finding then is previously unreported?**

Response:

Please also see the response #9 (page 10).

The core and shell structural organization has never been reported for mutant Httex1. Therefore, in our statement, we specifically mention that these structures have not been observed for mutant Httex1 inclusions.

As mentioned above, the work by Qin *et al.*, which we referred to and discussed, was carried out with a much longer fragment of Htt with a different polyQ repeat length, FLAG-tagged Htt1-969 fragment (100Q). Furthermore, our work provides a more in-depth characterization of

these inclusions at the biochemical and ultrastructural levels, whereas the report by Qin *et al.* provides only a morphological description of these inclusions.

In addition, Qin and colleagues did not assess the role of the polyQ length or Nt17 in regulating the ultrastructure of Htt inclusions by electron microscopy.

9. Line 682 In addition, the periphery of the inclusions was decorated with mitochondria and ER, as previously reported for the Htt inclusions in cellular models¹⁸ and human tissue^{74,75}. These changes in mitochondria morphology were associated with dysregulation of the mitochondria respiration (Figure 4), consistent with previous studies demonstrating that mutant Htt aggregates interact directly with outer mitochondrial membranes (in STHdh cells)⁷⁶ and induce mitochondrial fragmentation (in primary neurons)⁷⁷. **It is not clear to me what your work adds above these studies?**

Response: The previous studies, using STHdh and primary neurons (Choo *et al.* 2004; Song *et al.* 2011), focused only on mitochondrial morphological changes (fragmentation, swelling and motility), whereas here to assess in detail the impact of inclusions on live mitochondrial efficiency and key function of mitochondria, respiration, we used the gold-standard approach high resolution respirometry (HRR). HRR allowed us to demonstrate significantly higher mitochondrial respiration in cells expressing Httex1 72Q compared to Httex1 16Q in the different respiration states, uncovering – to our knowledge – for the first time the surprising effect of hyper-respiration in models of polyQ Httex1 overexpression. It is, however, of interest that such effects have recently also been observed in models of alpha-synuclein aggregation.

Our extensive proteomic analyses combined with the imaging and functional respiration approach allowed us to correlate the previously reported standalone observations of morphological alterations with changes in the aggregate composition and specific effects on different respirational states. This enabled us put the reported hyper-respiration of mitochondria in the untagged Httex1 Q72 condition in context with mitochondrial fragmentation and increased levels of proteins linked to inflammation in the aggregates, but not with increased mitochondrial ROS production. In addition, in our opinion, it is particularly interesting that these processes differed in the tagged Httex1 Q72 condition, which also has not been shown before. This suggests that GFP-tagged Httex1 Q72 triggers a different mitochondrial pathology not (to the same extent as for tag-free) involving the observed fragmentation and proteomic alterations of aggregate compositions, ultimately resulting in a distinct aggregation pathology that not fully reflects the pathology induced by tag-free Httex1 Q72. These results are discussed page 29 lines 721-743 and pages 31-32 lines 780-786.

10. I noticed there is not much in Discussion about **neutral lipids**- where are they in cell and what is recruiting them- longer pieces of HTT that have incorporated? Other proteins binding to them?

Response:

Neutral lipids are predominantly found in lipid droplets that emerge from the ER, but they can be found in both the cytoplasm and the nucleus as well (see our response #2 page 14 from referee 3). In addition, our proteomic analysis revealed very few membrane-associated proteins. The low abundance of full-length HTT inside Httex1 inclusions suggests that its presence unlikely explains the recruitment of membranes and lipids (see our response #3 page 14 from referee 3). As already discussed in our response #3 page 8, we do not think, based

on the existing data, that membranes are incorporated during the early oligomerization events. We believe that the first phase of Httex1 aggregation occurs through the rapid formation of a dense fibrillar core and is driven predominantly by intermolecular interactions involving the polyQ domain via phase separation-like mechanisms, as was recently proposed (Peskett et al. 2018). That being said, further studies are needed to elucidate the role of lipids in Htt oligomerization, fibrillization and inclusion formation and maturation.

11. How do these MS findings compare to that of Hosp et al paper where mass spec was used **MS on aggregates in R6/2?**

Response:

It is difficult to directly compare our results to those of aggregates in R6/2 due to major differences in the approaches used to isolate the inclusions and the fact that in previous R6/2 proteomics studies, the authors did not differentiate between nuclear and cytoplasmic aggregates. Moreover, the detergent fractionation, mass spectrometry preparation and statistical analysis they used are different from those we used in the analysis of inclusions from primary neurons, and this could lead to different protein enrichment.

However, to answer this interesting question, we directly compared the proteins significantly enriched in Httex1 72Q of the aggregate fraction in HEK cells to proteins from the insoluble fraction reported by Hosp and colleagues (Hosp et al. 2017).

The comparison of the proteome of the Httex1 72Q inclusions formed in HEK cells with that of the five and eight weeks-old R6/2 mice showed no similarities. However, five proteins (gene names: AAK1, CHMP2B, DNAJA2, PCBP1 and PRRC2A) were found both enriched in the insoluble fraction of the Httex1 72Q inclusions and the whole-brain insoluble fraction of the twelve weeks-old R6/2 mice.

We also compared the entire set of proteins detected in the insoluble fraction of HEK cells, the insoluble fraction of R6/2 mice, and the insoluble fraction of primary neurons, using a Venn diagram, Figure III. Interestingly, only 535 proteins were similar among the three datasets. More common insoluble proteins were found between R6/2 mice and neurons (1026) than between R6/2 mice and HEK cells (592). Therefore, even if the insoluble proteomes present similarities among the datasets (Figure III), significant protein enrichment due to mutant Httex1 expression compared to internal controls (e.g., Httex1 16Q for cellular expression and WT for mice) revealed only very few common proteins between the datasets with R6/2 mice.

Figure III. Venn diagram comparison of the insoluble proteins detected in HEK cells (2433 in total, blue), R6/2 mice from Hosp *et al.* (1335 in total, green) and primary neurons (4321 in total, pink). Each number corresponds to the proteins either unique or shared among the different fractions.

12. The authors seem to have skirted the issue of whether **aggregates are toxic or not**. While Arrasate *et al.* (2004) and Saudou *et al.*, (1998) were referenced, they are only discussed in terms of their use of GFP tags. **Why study aggregates if they are not toxic?** 87. Leitman J, Ulrich Hartl F, Lederkremer GZ. Soluble forms of polyQ-expanded huntingtin rather than large aggregates cause endoplasmic reticulum stress. *Nat Commun.* 2013;4:1-10. doi:10.1038/ncomms3753

Response: We thank referee #3 for raising this point. In fact, we have recently conducted systematic studies to assess the toxicity of Httex1 inclusions formation in our cellular models using multiple Httex1 mutants. The results were explained but not shown in the original manuscript as they are fully described in a new systematic study from our group, which is currently under review and available as a preprint in BioRxiv: <https://www.biorxiv.org/content/10.1101/2021.02.15.431207v1> (Sophie Vieweg *et al.* 2021). In HEK 293 cells containing inclusions, we observed only slightly toxic effects due to Httex1 inclusion formation despite the presence of large cytoplasmic aggregates. We do believe that low or no cellular toxicity indicates strong robustness and resilience of these cells. This work is now cited in the manuscript page 7, lines 139-142; page 10 lines 223-225; and page 11 lines 245-252.

In addition, we now have conducted a similar study in primary neurons expressing Httex1 72Q and included this data in the revised manuscript. We show high toxicity levels in neurons bearing nuclear inclusions (Figure 5E and 8D). We could demonstrate that Httex1 72Q-GFP aggregates exhibited a significantly reduced toxicity compared to tag-free Httex1 72Q. Indeed, the TUNEL cell death assay revealed a dramatic increase (~60%) in DNA fragmentation in the cortical neurons bearing the intranuclear Httex1 72Q inclusions compared to only 40% in neurons expressing Httex1 72Q-GFP (Figure 8D). We also provide some insight into the

cellular factors underpinning these differences on the basis of the structural properties of the inclusions and how they interact with subcellular structures in neurons.

13. In results, Line 453, “Httex1 72Q, Httex139Q, or Httex1 16Q” described being used, but later, Q39 is referred to. Is this intentional or a mistake (Q39 should be Q139?)

Response: We thank referee #3 for pointing out this typo. One space is indeed missing, it should be Httex1 39Q, and it is now corrected in the revised manuscript (see page 18, line 419).

14. Line 458, define EV.

Response: We thank the referee for highlighting this lack of clarity. We have now spell fully EV as “empty vector” in the manuscript (see page 18, line 397).

15. **Figure 1D. Does a figure such as this exist in HD patients?** Is this type of inclusion an artifact of overexpressing mutant exon1? Although a protein the size of exon1 may occur from expression of the RNA Intron readthrough may in HD patients, especially in cells where somatic expansion has already occurred, or by proteolysis of acid peptidases, there is otherwise very low levels of small fragments in cells expressing endogenous full length mutant HTT, often detectable only by methods such as HTRF or MSD. Thus, the level of Exon1 sized protein in cells is likely only enough just for seeding. Therefore, this seems like a very artificial situation in which the entire aggregate is driven by the same size fragment. While of some esoteric interest, it not very informative in terms of actual disease manifestation.

Response: With respect to the relevance of the mutant exon1 fragment, please see our responses to Referee #3 (page 5-7).

We do not believe that this is an artifact of overexpressing this fragment. To support this statement, we have conducted a thorough review of the literature on EM studies of Htt inclusions in HD brains. Despite the variation of the techniques used and the fact that previous studies relied mainly on antibodies to detect the aggregates (see the legend for more details (Figure IV), we observed several shared morphological properties between the inclusions we observed in the cellular models and those of inclusions in HD patient brains (Figure IV). Furthermore, this comparative analysis demonstrates that our work not only complements previous studies but also provides more detailed characterization on the ultrastructural properties of Htt inclusions and systematic analysis of their proteome, which was not investigated in previous morphological studies of Htt inclusions.

Neuronal intranuclear inclusions in HD brains and *in vivo* models of HD are all detected and described as predominantly granular with the appearance of filamentous structures (Figure IVA1-5). Previous studies on neuronal intranuclear inclusions in primary neurons were only described as protein deposits at low magnification by Tagawa *et al.* (Figure IVA6) and recently as a highly fibrillar inclusion body by Bäuerlein and colleagues (Figure IVA6). Our new data on nuclear neuronal Htt inclusion revealed that tag-free mutant Httex1 expression in primary neurons leads to granulo-filamentous nuclear inclusions, similar to previous ultrastructural studies performed in HD patient brains and *in vivo* (Figure IVA).

The discussion section has been modified to highlight these points and comparisons: “Our data are in line with previous EM studies from human HD patients (Tellez-Nagel, Johnson, and Terry 1974) and HD mice models (Davies et al. 1997; Gasset-Rosa et al. 2017) showing nuclear ultrastructural changes, including altered nuclear membrane shape, nuclear invagination and increased nuclear pore density in neurons bearing Httex1 inclusions.” Pages 19- 20, lines 474-481.

Cytoplasmic inclusions from HD patients were also described by DiFiglia *et al.* as containing granules and filaments surrounded by the accumulation of mitochondria (Figure IVB1). These are similar features to what we observed for the cytoplasmic tag-free Httex1 72Q inclusions that we characterized in our HEK 293 cellular system (Figure 1). Similarly, neuritic inclusions from transgenic rats and mice are also mostly granular with the detection of filaments (Figure IVB2-3).

In primary neurons, Tagawa and colleagues observed cytoplasmic inclusions of tag-free Httex1 pushing the nucleus, similar to what we observed in HEK cells, but they did not further characterize the ultrastructural properties of these inclusions (Figure IVB4, upper panel). In contrast, Httex1 72Q-GFP or Httex1 97Q-GFP cytoplasmic inclusions were predominantly fibrillar in primary neurons, reported by Miller *et al.* by EM and by Bäuerlein *et al.* by ET. (Figure IVB4, lower panel and VB5).

Ultrastructural characterization of Htt inclusions was also previously conducted in non-neuronal mammalian cells, where they were predominantly detected in the cytoplasm (Figure IVC). As discussed in the manuscript, the core and shell organization we detected for tag-free Httex1 72Q cytoplasmic inclusions was also observed by Qin *et al.* but using a longer fragment than Httex1: FLAG-tagged Htt1-969 fragment (100Q) (Figure IVC2). (See also response #9 page 10).

Tagawa and colleagues provide EM images of cytoplasmic tag-free Httex1-111Q inclusions in HeLa cells and showed that they display a core and shell morphology at low magnification. However, they did not report high-resolution data on the ultrastructural properties of these inclusions or assessed their biochemical composition. They simply described them qualitatively as amorphous deposits (Figure IVC3).

This review (current response and Figure IV) demonstrate that many of the features of inclusions seen in HD brains can be reproduced in our cellular models, including 1) the formation of granulo-filamentous neuronal intranuclear inclusions, 2) the accumulation of mitochondria in the periphery of the cytoplasmic inclusions, and 3) the presence of Actin-F associated with the inclusions. However, many of these features were only observed with tag-free Httex1 proteins. This underscores the critical importance of using tag-free native Htt sequences to investigate the mechanisms of inclusion formation and ultrastructural properties in cells. But our work, in addition, provides extensive biochemical characterizations of the inclusions, and novel functional assessments of the consequences of these inclusions on organelle function as well as on differential effects on the proteome. Such studies are largely lacking in the literature,

In both our work and previous studies on Htt inclusions by others, the core and shell organization is only seen with tag-free Httex1 fragments or the longer FLAG-Htt969 construct. Furthermore, the substantial differences of inclusions and related cellular consequences due to tagged or untagged Httex1 overexpression had not been reported before and in our opinion is an important result for the field.

Figure IV. Comparison of huntingtin inclusions observed by electron microscopy (EM) or electron tomography (ET) in HD patient brains, *in vivo* and cellular models of HD.

A. EM or ET of huntingtin neuronal intranuclear inclusions. **1)** Electron microscopy of a human neuronal intranuclear inclusion (hNII) in the HD cortex with immunoperoxidase labeling (Ab1). Upper panel: The hNII in a cortical neuron appears as a dense aggregate with no limiting membrane separating it from the nucleoplasm. Lower panel: Higher magnification of NII shows the presence of labeled granules and filaments within the inclusion. **2)** Mutant htt was detected with EM using the immunoperoxidase method and a primary anti-body to htt1–17. The peroxidase reaction product appears as electron dense label.

EM are from a cortical neuron in human post-mortem HD brain. Upper panel: Note that the labeled inclusion occupies a large area at the center of the nucleus (nuc). The composition of the inclusion is highly heterogeneous consisting of granular/ oligomeric and fibrillar structures. Stacks of labeled fibrils are present at the upper part of the inclusion and along the right at the open arrow, which is magnified the lower panel. Lower panel: note the presence of small, labeled granular/oligomeric structures of different sizes (arrowheads). Fibrils (arrows) and structures that resemble beadson a string (ringed arrow) are also indicated. **3)** EM Localization of huntingtin within the Neuronal Nucleus of Transgenic Mice. Discrete deposition of DAB reaction product within the NII with antibodies to htt. A single intensely stained inclusion is seen. **4)** Ultrastructure of intranuclear transgenic rat expressing Htt727 with 51Q by conventional transmission electron microscopy. Top panel: A medium-sized striatal neuron shows a round intranuclear aggregate (large arrow) slightly larger than the neighbored nucleolus (N). Lower panel: At higher magnification, the membraneless aggregate reveals a granular and fibrillar (arrowheads) appearance. **5)** Nuclear inclusion in the cortex and striatum of Tet/HD94 mice: Ultrastructural transmission electron microscopy analysis of a neuropil from the cortex of a Tet/HD94 mouse. **6)** Electron microscopic analysis of cerebral cortical neurons expressing htt111: A representative neuron possessing nuclear aggregates of htt111 (arrows). The cytoplasm is disrupted. In addition, the heterochromatin has disappeared from the nucleus where a small nucleolus is observed. **7)** 1.7-nm thick tomographic slice of a nuclear inclusion body (IB) in an Htt97Q-GFP-transfected neuron.

B. EM or ET of huntingtin inclusions found in neuritic or cytoplasmic inclusions in neurons. **1)** Electron microscopy of human dystrophic neurites (hDNs) in the HD cortex with immunoperoxidase labeling: hDN contains an aggregate of immunoreactive granules and filaments, which is surrounded by a rim of cytoplasm where mitochondria are accumulated. **2)** Ultrastructure of intranuclear transgenic rat expressing Htt727 with 51Q by conventional transmission electron microscopy. Top panel: The aggregate localized in a cross-sectioned dendrite (large arrow) is surrounded by numerous mitochondria (small arrows). Lower panel: At higher magnification, the dendritic aggregate also exhibits granular and fibrillar (arrowheads) composition. **3)** Dystrophic neurite inclusion in the cortex and striatum of Tet/HD94 mice: Ultrastructural transmission electron microscopy analysis of a dystrophic neurite from the cortex of a Tet/HD94 mouse. **4)** Electron microscopic analysis of cerebral cortical neurons expressing htt111. Some neurons possess perinuclear aggregates (arrows). **5)** Primary striatal neuron model of HD. IBs in this model system have a granular composition by electron microscopy, as seen in human HD patients. Electron microscopy was performed on rat striatal neurons transfected with mutant htt. The inset is an enlarged view of the IB. **6)** Tomographic slice of an IB in an Htt97Q-GFP-transfected mouse primary neuron. The cytoplasmic electron-dense particles represent ribosomes (white arrowheads). ER, endoplasmic reticulum; IB, Htt97Q-GFP inclusion body; Vs, vesicle. Inset: high-magnification image of Htt97Q-GFP fibrils (red arrowheads) decorated by globular densities (green arrowheads).

C. EM or ET of huntingtin inclusions found in cytoplasmic inclusions in mammalian cell lines. **1)** Electron micrographs of 293 Tet-Off cells containing HDQ83 aggregates. After HDQ83 expression for 3–5 d, cells were fixed and viewed by electron microscopy. EM of a cell containing a typical perinuclear inclusion body. Top panel: At higher magnifications, HDQ83 fibrils with a diameter of ~10 nm can be observed. Lower panel: Higher magnification of the IB. **2)** Ultrastructure of htt bodies. MCF-7 cells were transiently transfected with FH969–100Q and immunostained for FLAG using the immunoperoxidase method. Top panel: cell contains one FLAG-labeled htt body (arrow) that is in close proximity to several mitochondria. Then nucleus (Nuc) has a few small patches of immunoreactivity. Lower panel: Higher magnification of a serial section through the htt body shown in the upper panel. Note the radiating fibrils in the core (C) and the less defined structure in the shell (S). Multilamellar-type autophagic bodies (a; arrows) about the shell. **3)** Electron microscopic analysis of HeLa cells expressing htt111. Top panel: A representative htt111-expressing cell harbors a large cytoplasmic aggregate (black arrows) and small nuclear aggregates (white arrows). Lower panel: Immunoelectron microscopic analysis of a representative HeLa cell expressing htt111 shows a large cytoplasmic aggregate detected by the N-18 antibody and secondary gold-labeled antibody. Magnification of the area surrounded by white dots (inset) shows numerous gold particles on the htt111 deposit. **4)** Tomographic slice from the interaction zone between an IB and cellular membranes in an Htt97Q-transfected HeLa cell. ER, endoplasmic reticulum; IB, Htt97Q inclusion body; Vs, vesicles.

Adapted from: A1) and B1) DiFiglia and colleagues (DiFiglia et al. 1997); **A2) Legleiter and colleagues** (Legleiter et al. 2010); **A3) Davies and colleagues** (Davies et al. 1997); **A4) and B3) Petrasch-Parwez and colleagues** (Petrasch-Parwez et al. 2007); **A5) and B3) Díaz-Hernández and colleagues** (Díaz-Hernández et al. 2004); **A6), B4) and C3) Tagawa and colleagues** (Tagawa et al.

2004); **A7), B6) and C4) Bäuerlein and colleagues** (Bäuerlein et al. 2017). **C1) Waelter and colleagues** (Waelter et al. 2001); **Qin and colleagues** (Qin et al. 2004).

15b. I do agree that this work has yielded some interesting, though predicable findings. The CLEM nicely presents how the ER and mitochondria interact with this aggregate. Are the aggregates observed here including mostly ER membrane because you have **focused on perinuclear aggregates?**

We thank the referee for his/her very positive feedback on the quality of our work and for highlighting some of the novel and interesting findings from our work. We agree that some of our findings were predictable, especially the comparison of nuclear and cytoplasmic inclusions, but such comparison at this level has never been done in the framework of a single study at 1) the ultrastructural, biochemical, and functional levels; and 2) using different Htt constructs, with and without GFP or Nt17.

We did not specifically focus on perinuclear aggregates as the HEK cells have a restricted cytoplasmic volume due to the presence of a large nucleus that occupies the majority of the space. Therefore, all the cytoplasmic aggregates are found in the vicinity of the nucleus, with an average distance between the nuclear membrane and the inclusions close to $\sim 1 \mu\text{m}$ (Figure S18).

References

- Angeli, Suzanne, Jieya Shao, and Marc I. Diamond. 2010. "F-Actin Binding Regions on the Androgen Receptor and Huntingtin Increase Aggregation and Alter Aggregate Characteristics." *PLoS ONE* 5(2): e9053.
- Aron, Rebecca et al. 2018. "Deubiquitinase Usp12 Functions Noncatalytically to Induce Autophagy and Confer Neuroprotection in Models of Huntington's Disease." *Nature Communications* 9(1). <http://dx.doi.org/10.1038/s41467-018-05653-z>.
- Barbaro, Brett A. et al. 2015. "Comparative Study of Naturally Occurring Huntingtin Fragments in *Drosophila* Points to Exon 1 as the Most Pathogenic Species in Huntington's Disease." *Human Molecular Genetics* 24(4): 913–25.
- Bäuerlein, Felix J.B. et al. 2017. "In Situ Architecture and Cellular Interactions of PolyQ Inclusions." *Cell* 171(1): 179-187.e10.
- Blum, Matthias et al. 2021. "The InterPro Protein Families and Domains Database: 20 Years On." *Nucleic Acids Research* 49(D1): D344–54.
- Carnemolla, Alisia, Silvia Michelazzi, and Elena Agostoni. 2017. "PIN1 Modulates Huntingtin Levels and Aggregate Accumulation: An in Vitro Model." *Frontiers in Cellular Neuroscience* 11(May): 1–12.
- Cherubini, Marta, Laura Lopez-Molina, and Silvia Gines. 2020. "Mitochondrial Fission in Huntington's Disease Mouse Striatum Disrupts ER-Mitochondria Contacts Leading to Disturbances in Ca²⁺ Efflux and Reactive Oxygen Species (ROS) Homeostasis." *Neurobiology of Disease* 136(November 2019): 104741. <https://doi.org/10.1016/j.nbd.2020.104741>.
- Choo, Yeun Su et al. 2004. "Mutant Huntingtin Directly Increases Susceptibility of Mitochondria to the Calcium-Induced Permeability Transition and Cytochrome c Release." *Human Molecular Genetics* 13(14): 1407–20.
- Cisbani, G, and F Cicchetti. 2019. "An in Vitro Perspective on the Molecular Mechanisms Underlying Mutant Huntingtin Protein Toxicity." *Cell Death and Disease* 3(8): 1–28.
- Cooper, Jillian K. et al. 1998. "Truncated N-Terminal Fragments of Huntingtin with Expanded Glutamine Repeats Form Nuclear and Cytoplasmic Aggregates in Cell Culture." *Human Molecular Genetics* 7(5): 783–90.
- Davies, Stephen W et al. 1997. "Formation of Neuronal Intranuclear Inclusions Underlies the Neurological Dysfunction in Mice Transgenic for the HD Mutation." *Cell* 90: 537–48.
- Díaz-Hernández, Miguel et al. 2004. "Biochemical, Ultrastructural, and Reversibility Studies on Huntingtin Filaments Isolated from Mouse and Human Brain." In *Journal of Neuroscience*, , 9361–71.
- DiFiglia, Marian et al. 1997. "Aggregation of Huntingtin in Neuronal Intranuclear Inclusions and Dystrophic Neurites in Brain." *Science* 277(5334): 1990–93. <http://www.sciencemag.org/cgi/doi/10.1126/science.277.5334.1990>.
- El-Daher, M.-T. et al. 2015. "Huntingtin Proteolysis Releases Non-PolyQ Fragments That Cause Toxicity through Dynamin 1 Dysregulation." *The EMBO Journal* 34(17): 2255–71. <http://emboj.embopress.org/cgi/doi/10.15252/emboj.201490808>.
- Franich, Nicholas R. et al. 2019. "Phenotype Onset in Huntington's Disease Knock-in Mice Is Correlated with the Incomplete Splicing of the Mutant Huntingtin Gene." *Journal of Neuroscience Research* 97(12): 1590–1605.
- Gasset-Rosa, Fatima et al. 2017. "Polyglutamine-Expanded Huntingtin Exacerbates Age-

- Related Disruption of Nuclear Integrity and Nucleocytoplasmic Transport.” *Neuron* 94(1): 48-57.e4.
- Gerson, Julia E. et al. 2020. “Ubiquilin-2 Differentially Regulates Polyglutamine Disease Proteins.” *Human Molecular Genetics* 29(15): 2596–2610.
- Goold, Robert et al. 2019. “FAN1 Modifies Huntington’s Disease Progression by Stabilizing the Expanded HTT CAG Repeat.” *Human Molecular Genetics* 28(4): 650–61.
- Graham, Rona K. et al. 2010. “Cleavage at the 586 Amino Acid Caspase-6 Site in Mutant Huntingtin Influences Caspase-6 Activation in Vivo.” *Journal of Neuroscience* 30(45): 15019–29.
- Grazioli, Serge, and Jérôme Pugin. 2018. “Mitochondrial Damage-Associated Molecular Patterns: From Inflammatory Signaling to Human Diseases.” *Frontiers in Immunology* 9(MAY): 1–17.
- Gu, Xiaofeng et al. 2009. “Serines 13 and 16 Are Critical Determinants of Full-Length Human Mutant Huntingtin-Induced Disease Pathogenesis in HD Mice.” *Neuron* 64(6): 828–40.
- Guo, Qiang et al. 2018. “The Cryo-Electron Microscopy Structure of Huntingtin.” *Nature*. <http://www.nature.com/doi/10.1038/nature25502>.
- Hackam, Abigail S. et al. 1998. “The Influence of Huntingtin Protein Size on Nuclear Localization and Cellular Toxicity.” *Journal of Cell Biology* 141(5): 1097–1105.
- Hegde, Ramanath Narayana et al. 2020. “TBK1 Phosphorylates Mutant Huntingtin and Suppresses Its Aggregation and Toxicity in Huntington’s Disease Models.” *The EMBO Journal* 39(17): e104671.
- Horn, Adam et al. 2020. “Mitochondrial Fragmentation Enables Localized Signaling Required for Cell Repair.” *Journal of Cell Biology* 219(5): e201909154.
- Hosp, Fabian et al. 2017. “Spatiotemporal Proteomic Profiling of Huntington’s Disease Inclusions Reveals Widespread Loss of Protein Function.” *Cell Reports* 21(8): 2291–2303. <http://linkinghub.elsevier.com/retrieve/pii/S2211124717315772>.
- Jiang, Mali et al. 2020. “Nemo-like Kinase Reduces Mutant Huntingtin Levels and Mitigates Huntington’s Disease.” *Human Molecular Genetics* 29(8): 1340–52.
- Kegel, K. B. et al. 2000. “Huntingtin Expression Stimulates Endosomal-Lysosomal Activity, Endosome Tubulation, and Autophagy.” *Journal of Neuroscience* 20(19): 7268–78.
- Kim, Manho et al. 1999. “Mutant Huntingtin Expression in Clonal Striatal Cells: Dissociation of Inclusion Formation and Neuronal Survival by Caspase Inhibition.” *Journal of Neuroscience* 19(3): 964–73.
- Kim, Yujin E. et al. 2016. “Soluble Oligomers of PolyQ-Expanded Huntingtin Target a Multiplicity of Key Cellular Factors.” *Molecular Cell* 63(6): 950–64.
- Klaips, Courtney L., Michael H.M. Gropp, Mark S. Hipp, and F. Ulrich Hartl. 2020. “Sis1 Potentiates the Stress Response to Protein Aggregation and Elevated Temperature.” *Nature Communications* 11(1): 6271. <http://dx.doi.org/10.1038/s41467-020-20000-x>.
- Kolla, Rajasekhar et al. 2021. “A New Chemoenzymatic Semisynthetic Approach Provides Insight into the Role of Phosphorylation beyond Exon1 of Huntingtin and Reveals N-Terminal Fragment Length-Dependent Distinct Mechanisms of Aggregation.” *Journal of the American Chemical Society* 143(26): 9798–9812.
- Landles, Christian et al. 2010. “Proteolysis of Mutant Huntingtin Produces an Exon 1 Fragment That Accumulates as an Aggregated Protein in Neuronal Nuclei in Huntington

- Disease." *Journal of Biological Chemistry* 285(12): 8808–23.
- Legleiter, Justin et al. 2010. "Mutant Huntingtin Fragments Form Oligomers in a Polyglutamine Length-Dependent Manner in Vitro and in Vivo." *Journal of Biological Chemistry* 285(19): 14777–90.
- Liu, Beidong et al. 2011. "Segregation of Protein Aggregates Involves Actin and the Polarity Machinery." *Cell* 147(5): 959–61.
- Lu, Meng et al. 2019. "Live-Cell Super-Resolution Microscopy Reveals a Primary Role for Diffusion in Polyglutamine-Driven Aggresome Assembly." *Journal of Biological Chemistry* 294(1): 257–68.
- Lunkes, Astrid et al. 2002. "Proteases Acting on Mutant Huntingtin Generate Cleaved Products That Differentially Build up Cytoplasmic and Nuclear Inclusions." *Molecular Cell* 10(2): 259–69.
- Lunkes, Astrid, and Jean Louis Mandel. 1998. "A Cellular Model That Recapitulates Major Pathogenic Steps of Huntington's Disease." *Human Molecular Genetics* 7(9): 1355–61.
- Luo, Huanhuan et al. 2018. "Herp Promotes Degradation of Mutant Huntingtin: Involvement of the Proteasome and Molecular Chaperones." *Molecular neurobiology* 55(10): 7652–68. <http://link.springer.com/10.1007/s12035-018-0900-8>
<http://www.ncbi.nlm.nih.gov/pubmed/29430620>.
- Manczak, Maria, and P. Hemachandra Reddy. 2015. "Mitochondrial Division Inhibitor 1 Protects against Mutant Huntingtin-Induced Abnormal Mitochondrial Dynamics and Neuronal Damage in Huntington's Disease." *Human Molecular Genetics* 24(25): 7308–25.
- Mangiarini, Laura et al. 1996. "Exon I of the HD Gene with an Expanded CAG Repeat Is Sufficient to Cause a Progressive Neurological Phenotype in Transgenic Mice." *Cell* 87(3): 493–506.
- Martindale, Diane et al. 1998. "Length of Huntingtin and Its Polyglutamine Tract Influences Localization and Frequency of Intracellular Aggregates." *Nature Genetics* 18(3): 150–54. <http://www.ncbi.nlm.nih.gov/pubmed/9500544>.
- Mistry, Jaina et al. 2021. "Pfam: The Protein Families Database in 2021." *Nucleic Acids Research* 49(D1): D412–19.
- Muchowski, Paul J., Ke Ning, Crislyn D'Souza-Schorey, and Stanley Fields. 2002. "Requirement of an Intact Microtubule Cytoskeleton for Aggregation and Inclusion Body Formation by a Mutant Huntingtin Fragment." *Proceedings of the National Academy of Sciences of the United States of America* 99(2): 727–32.
- Neueder, Andreas et al. 2017. "The Pathogenic Exon 1 HTT Protein Is Produced by Incomplete Splicing in Huntington's Disease Patients." *Scientific Reports* 7(1): 1307. <http://www.nature.com/articles/s41598-017-01510-z>.
- Neueder, Andreas, Anaëlle A. Dumas, Agnieszka C. Benjamin, and Gillian P. Bates. 2018. "Regulatory Mechanisms of Incomplete Huntingtin mRNA Splicing." *Nature Communications* 9(1): 3955. <http://www.nature.com/articles/s41467-018-06281-3>.
- Nguyen, Trent, Aaron Hamby, and Stephen M. Massa. 2005. "Clioquinol Down-Regulates Mutant Huntingtin Expression in Vitro and Mitigates Pathology in a Huntington's Disease Mouse Model." *Proceedings of the National Academy of Sciences of the United States of America* 102(33): 11840–45.
- Park, Yeonkyoung et al. 2020. "Nonsense-Mediated mRNA Decay Factor UPF1 Promotes

- Aggresome Formation." *Nature Communications* 11(1): 1–15.
<http://dx.doi.org/10.1038/s41467-020-16939-6>.
- Peskett, Thomas R. et al. 2018. "A Liquid to Solid Phase Transition Underlying Pathological Huntingtin Exon1 Aggregation." *Molecular Cell* 70(4): 588–601.
- Petrasch-Parwez, Elisabeth et al. 2007. "Cellular and Subcellular Localization of Huntingtin Aggregates in the Brain of a Rat Transgenic for Huntington Disease." *Journal of Comparative Neurology* 501(October 2007): 716–30.
- Qin, Zheng-hong et al. 2004. "Huntingtin Bodies Sequester Vesicle-Associated Proteins by a Polyproline-Dependent Interaction." *The Journal of neuroscience : the official journal of the Society for Neuroscience* 24(1): 269–81.
- Ramdzan, Yasmin M. et al. 2017. "Huntingtin Inclusions Trigger Cellular Quiescence, Deactivate Apoptosis, and Lead to Delayed Necrosis." *Cell Reports* 19(5): 919–27.
- Ratovitski, Tamara et al. 2009. "Mutant Huntingtin N-Terminal Fragments of Specific Size Mediate Aggregation and Toxicity in Neuronal Cells." *Journal of Biological Chemistry* 284(16): 10855–67.
- . 2016. "Quantitative Proteomic Analysis Reveals Similarities between Huntington's Disease (HD) and Huntington's Disease-Like 2 (HDL2) Human Brains." *J Proteome Res.* 46(5): 1247–62.
- Rockabrand, Erica et al. 2007. "The First 17 Amino Acids of Huntingtin Modulate Its Sub-Cellular Localization, Aggregation and Effects on Calcium Homeostasis." *Human Molecular Genetics* 16(1): 61–77.
- Sathasivam, K. et al. 2013. "Aberrant Splicing of HTT Generates the Pathogenic Exon 1 Protein in Huntington Disease." *Proceedings of the National Academy of Sciences* 110(6): 2366–70. <http://www.pnas.org/cgi/doi/10.1073/pnas.1221891110>.
- Saudou, Frédéric, Steven Finkbeiner, Didier Devys, and Michael E. Greenberg. 1998. "Huntingtin Acts in the Nucleus to Induce Apoptosis but Death Does Not Correlate with the Formation of Intranuclear Inclusions." *Cell* 95(1): 55–56.
- Schilling, Gabriele et al. 2007. "Characterization of Huntingtin Pathologic Fragments in Human Huntington Disease, Transgenic Mice, and Cell Models." *Journal of Neuropathology and Experimental Neurology* 66(4): 313–20.
- Scior, Annika et al. 2018. "Complete Suppression of Htt Fibrilization and Disaggregation of Htt Fibrils by a Trimeric Chaperone Complex." *The EMBO Journal* 37(2): 282–99.
- Shen, Koning et al. 2016. "Control of the Structural Landscape and Neuronal Proteotoxicity of Mutant Huntingtin by Domains Flanking the PolyQ Tract." *eLife* 5: e18065.
<http://www.ncbi.nlm.nih.gov/pubmed/27751235>
<http://www.pubmedcentral.nih.gov/articlerender.fcgi?artid=PMC5135392>.
- Singer, Elisabeth et al. 2021. "The Novel Alpha-2 Adrenoceptor Inhibitor Beditin Reduces Cytotoxicity and Huntingtin Aggregates in Cell Models of Huntington's Disease." *Pharmaceuticals* 14(3).
- Song, Wenjun et al. 2011. "Mutant Huntingtin Binds the Mitochondrial Fission GTPase Dynamin-Related Protein-1 and Increases Its Enzymatic Activity." *Nature Medicine* 17(3): 377–83. <http://dx.doi.org/10.1038/nm.2313>.
- Sophie Vieweg et al. 2021. "Towards Deciphering the Nt17 Code: How the Sequence and Conformation of the First 17 Amino Acids in Huntingtin Regulate the Aggregation, Cellular Properties and Neurotoxicity of Mutant Httex1." *bioRxiv*: 6.

- Steffan, Joan S et al. 2004. "SUMO Modification of Huntingtin and Huntington's Disease Pathology." *Science (New York, N.Y.)* 304(5667): 100–104.
<http://www.ncbi.nlm.nih.gov/pubmed/15064418>.
- Sun, Yueshan et al. 2020. "Escins Isolated from *Aesculus Chinensis* Bge. Promote the Autophagic Degradation of Mutant Huntingtin and Inhibit Its Induced Apoptosis in HT22 Cells." *Frontiers in Pharmacology* 11(February): 1–20.
- Suopanki, Jaana et al. 2006. "Interaction of Huntingtin Fragments with Brain Membranes - Clues to Early Dysfunction in Huntington's Disease." *Journal of Neurochemistry* 96(3): 870–84.
- Tagawa, Kazuhiko et al. 2004. "Distinct Aggregation and Cell Death Patterns among Different Types of Primary Neurons Induced by Mutant Huntingtin Protein." *Journal of Neurochemistry* 89(4): 974–87.
- Tam, Stephen et al. 2009. "The Chaperonin TRiC Blocks a Huntingtin Sequence Element That Promotes the Conformational Switch to Aggregation." *Nature Structural & Molecular Biology* 16(12): 1279–85.
<http://www.nature.com/doi/10.1038/nsmb.1700>.
- Taran, Aleksandra S, Lilia D Shuvalova, and Maria A Lagarkova. 2020. "Huntington's Disease—An Outlook on the Interplay of the HTT Protein, Microtubules and Actin Cytoskeletal Components." *Cells* 9(6): 1514.
- Tellez-Nagel, Isabel, Anne B. Johnson, and Robert D. Terry. 1974. "Studies on Brain Biopsies of Patients with Huntington's Chorea." *Journal of Neuropathology & Experimental Neurology* 33(2): 308–32.
- Thakur, Ashwani K et al. 2009. "Polyglutamine Disruption of the Huntingtin Exon1 N-Terminus Triggers a Complex Aggregation Mechanism Ashwani." *Nat Struct Mol Biol* 16(4): 380–89.
- Thompson, Leslie Michels et al. 2009. "IKK Phosphorylates Huntingtin and Targets It for Degradation by the Proteasome and Lysosome." *Journal of Cell Biology* 187(7): 1083–99.
- Trettel, Flavia et al. 2000. "Dominant Phenotypes Produced by the HD Mutation in STHdh(Q111) Striatal Cells." *Human Molecular Genetics* 9(19): 2799–2809.
- Ugalde, Cathryn L. et al. 2020. "Misfolded α -Synuclein Causes Hyperactive Respiration without Functional Deficit in Live Neuroblastoma Cells." *DMM Disease Models and Mechanisms* 13(1): dmm040899.
- Waelter, S et al. 2001. "Accumulation of Mutant Huntingtin Fragments in Aggresome-like Inclusion Bodies as a Result of Insufficient Protein Degradation." *Molecular biology of the cell* 12(5): 1393–1407.
- Wang, Hongmin et al. 2006. "Suppression of Polyglutamine-Induced Toxicity in Cell and Animal Models of Huntington's Disease by Ubiquilin." *Human Molecular Genetics* 15(6): 1025–41.
- Wang, Ping et al. 2017. "Acetylation-Induced TDP-43 Pathology Is Suppressed by an HSF1-Dependent Chaperone Program." *Nature Communications* 8(1): 1–15.
<http://dx.doi.org/10.1038/s41467-017-00088-4>.
- Wang, Qiuju et al. 2011. "The Common Inhaled Anesthetic Isoflurane Increases Aggregation of Huntingtin and Alters Calcium Homeostasis in a Cell Model of Huntington's Disease." *Toxicology and Applied Pharmacology* 250(3): 291–98.
<http://dx.doi.org/10.1016/j.taap.2010.10.032>.

- Warby, Simon C. et al. 2008. "Activated Caspase-6 and Caspase-6-Cleaved Fragments of Huntingtin Specifically Colocalize in the Nucleus." *Human Molecular Genetics* 17(15): 2390–2404.
- Wellington, Cheryl L et al. 2002. "Caspase Cleavage of Mutant Huntingtin Precedes Neurodegeneration in Huntington's Disease." *Journal of Cell Biology* 22(4): 749–59.
- Yin, Xiangling, Maria Manczak, and P. Hemachandra Reddy. 2016. "Mitochondria-Targeted Molecules MitoQ and SS31 Reduce Mutant Huntingtin-Induced Mitochondrial Toxicity and Synaptic Damage in Huntington's Disease." *Human Molecular Genetics* 25(9): 1739–53.
- Ylä-Anttila, Päivi, Soham Gupta, and Maria G. Masucci. 2021. "The Epstein-Barr Virus Deubiquitinase BPLF1 Targets SQSTM1/P62 to Inhibit Selective Autophagy." *Autophagy* 00(00): 1–15. <https://doi.org/10.1080/15548627.2021.1874660>.
- Zheng, Zhiqiang et al. 2013. "An N-Terminal Nuclear Export Signal Regulates Trafficking and Aggregation of Huntingtin (Htt) Protein Exon 1." *Journal of Biological Chemistry* 288(9): 6063–71.

REVIEWER COMMENTS

Reviewer #1 (Remarks to the Author):

Authors addressed my prior concerns, except literature survey - some how manuscript still gives a sense of inclusions article rather than mechanistic one.

Reviewer #3 (Remarks to the Author):

The authors have addressed my concerns especially regarding studies in neurons. It is interesting that synaptic proteins are not particularly found despite reports of synaptic aggregates. I congratulated them on a nice piece of work that clearly shows the potential problems with interpreting studies using epitope/protein tags and offers insight into aggregated pathology. For their own reference I include information on a Review of membrane association regions in HTT which may be of interest when investigating larger fragments of HTT:

J Huntingtons Dis. 2013;2(3):239-50. doi: 10.3233/JHD-130068. PMID: 25062673.

Please find below our response to the revised manuscript “***Nuclear and cytoplasmic huntingtin inclusions exhibit distinct biochemical composition, interactome and ultrastructural properties.***”

Here is a separate point-by-point response to the reviewers’ comments, reproduced verbatim.

REVIEWERS' COMMENTS

Reviewer #1 (Remarks to the Author):

Authors addressed my prior concerns, except literature survey - some how manuscript still gives a sense of inclusions article rather than mechanistic one.

We are happy that the reviewer is satisfied with our responses and the revised version of the manuscript. While it is true that the primary focus of the manuscript is to present a depth characterization of Htt cytoplasmic and nuclear inclusions at the molecular and structural levels, our work also provides a mechanistic model for inclusion formation and new insight into the role of polyQ- repeat length and the cellular milieu in regulating Htt inclusions architecture and toxicity.

Reviewer #3 (Remarks to the Author):

The authors have addressed my concerns especially regarding studies in neurons. It is interesting that synaptic proteins are not particularly found despite reports of synaptic aggregates. I congratulated them on a nice piece of work that clearly shows the potential problems with interpreting studies using epitope/protein tags and offers insight into aggregated pathology. For their own reference I include information on a Review of membrane association regions in HTT which may be of interest when investigating larger fragments of HTT:

J Huntingtons Dis. 2013;2(3):239-50. doi: 10.3233/JHD-130068. PMID: 25062673.

We thank the reviewer for the very positive feedback on our revised manuscript and reference suggestion, which we have included in the final version of the manuscript.